# Port-Hamiltonian Formulations of Some Elastodynamics Theories of Isotropic and Linearly Elastic Shells: Naghdi–Reissner's Moderately Thick Shells

**Miguel Charlotte [1], Ignacio Fernandez Núñez [2], Yves Gourinat [1,\*] and Denis Matignon [2]**

[1] Institut Clément ADER, Université de Toulouse, ISAE-SUPAERO, INSA, UTIII, IMT Mines Albi, CNRS, 31400 Toulouse, France

[2] ISAE-SUPAERO, Université de Toulouse, 31055 Toulouse, France

\* Correspondence: yves.gourinat@isae-supaero.fr

**Abstract:** The port-Hamiltonian system approach is intended to be an innovative and unifying way of modeling multiphysics systems, by expressing all of them as systems of conservation laws. Indeed, the increasing developments in recent years allow finding better control and coupling strategies. This work aimed to apply such an approach to Naghdi–Reissner's five-kinematic-field shell model in linear elasticity, while including often-neglected higher-order intrinsic geometric coupling effects, therefore preparing the theoretical background required for the coupling (or interconnection) with an acoustic fluid model and the different types of interactions that can arise among them. The model derived thusly can be used for controller design in a wide variety of applications such as inflatable space structures, launcher tank vibration damping, payload vibration protection using smart materials, and many other related applications.

**Keywords:** port-Hamiltonian systems; mixed variational formulations; Naghdi–Reissner's shell theory; elastodynamics

## 1. Introduction

**Subject matter, main purpose, and practical application value of this work.** In the last few decades, the development of control algorithms for nonlinear complex systems has been a productive field of research. To this end, the paradigm of port-Hamiltonian systems (**PHS**) currently provides an elegant and efficient (unifying mathematical) framework to model, couple, and control various complex multiphysics systems such as mechanical, electrical, and thermal systems, while taking into account the effects of energy flows, whether inside the bulk or via the boundary (or *port*) of the interconnected PHS (which, by definition, are each open). The purpose of this paper is merely to apply the PHS formalism to Naghdi–Reissner's (**NR**) shell model in linear elasticity in order to prepare the theoretical background required for the numerical coupling (or interconnection) with an acoustic fluid model and the different types of interactions that can arise among them. Our interest in applying this PHS formalism is also motivated by the fact that its possibilities and limits in structural and continuum mechanics are not yet fully elucidated. Additionally, the proper Hamiltonian/Lagrangian variational principles (or laws of Varying Action) for systematically generating the canonical PHS equations still need to be better understood from this perspective in order to adopt the most-efficient computational methods.

**Brief review of the current state-of-the-art of the PHS formalism.** A basic exposition of what port-Hamiltonian systems are can be found in [1], and the progress in the last twenty years since the pioneering work [2] is given in [3]. In essence, the underlying framework of PHS is based on the variational theory combining the Hamilton(–D'Alembert–Lagrange) principle with the Fenchel–Legendre (Legendre–Young) transformations of the Lagrangian potential. The PHS formalism highlights the (Stokes–)Dirac mathematical

structure, which guarantees a global (power) balance equation for the Hamiltonian functional [1,4–7]. This framework is currently undergoing an enormous progressive evolution for continuous (or again, distributed-parameter) systems with importance for civil, aeronautical, and astronautical engineering, though still less developed compared to the discrete (or lumped-parameter) systems.

Indeed, based on the symplectic geometry formalism [5], PHS for distributed-parameter systems were initially introduced by A. J. Van der Schaft and B. Maschke [2], with the aim to combine the advantages of network theory to model complex systems with those of the geometric Hamiltonian analysis of physical systems to derive better control algorithms based on energy considerations [8,9]. While going progressively from simple systems to more complex ones, including continuous systems or distributed-parameter systems, recent publications have proven the possibility of applying this formulation to structural models of beams [10–13] and plates [14–18] or to unidimensional acoustic fluids [19] and ideal isentropic fluids [20], among other examples. Notably, related to this current work, writing the dynamic equations of plates as a PHS was carried out, for instance, in [14] for thick plates represented by the Reissner–Mindlin model and in [15] for thin plates represented by the Kirchhoff–Love model. Regarding fluid mechanics, some light can be shed on applications involving Navier–Stokes equations, both in [21,22], just to cite a few works. Besides, as far as multiphysics coupling is concerned, the PHS framework has been proven to be very powerful and well suited to this aim: quite a few examples can be found in [23] for the Allen–Cahn equationand heat, in [24] for elasticity and heat, in [12,25] for a liquid sloshing in a moving container, in [26] for heat waves, and [27] for thermo-magneto-hydrodynamics coupling in tokamaks. It is important to point out as well that the PHS formulation for other higher-order equation systems was notably given in [17,28], thus sustaining the promise that, whenever the equations of a theory can be derived from a variational principle, symplectic geometry [4,5,29,30] can clear up and systematize the relations between the quantities entering into the theory. The readers interested in the PHS formalism can also further refer to the books and thesis dissertations [1,8,9,31,32] for a more general overview of the theory and control applications.

**The numerical motivation of this work for PHS.** Our interest in the PHS formulation greatly relies on the robustness (i.e., long-term stability) and accuracy of the Numerical Symplectic Integration (**NSI**) methods [33–38], which are concurrent with the Variational Numerical Integration (**VNI**) methods [30,35,39–48] within the Geometric Numerical Integration (**GNI**) methods (also often called structure-preserving numerical methods, as they preserve the internal geometric structure of differential equations) [35,47,49–51]. These numerical integrators or algorithms are known to be clearly superior to other methods over long times of simulation. This is generally attributed to the fact that the underlying geometric structures of the dynamic models have a certain importance that influences the nature of their solutions. More precisely, according to [29,33,52] (and the references therein), these important features can be the Hamiltonian structure, namely the Poisson bracket, the energy, and the conserved quantities in the long time computation of conservative systems. Standard numerical integration schemes (such as, for instance, the forward Euler integrator and some time-adaptive Runge–Kutta schemes, unless purposely designed for [37,53]) naïvely neglect important special features of the system dynamics (such as the energy and momentum conservation and phase-space volume preservation). These advantages partly motivate the present study, although we do not fully address the numerical aspects in this paper and postpone this task with numerical benchmarking illustrations on the NR shells to a separate communication.

**Related variational formulations for the elastodynamics of shells and plates.** To the best of the authors' knowledge, the PHS formulation of the *linear elastodynamics of the NR shell model* has not yet been given in the literature, at least in an explicit practical form. To fill this gap for the NR shell model, an idea that we promote and intend to outline below is the specific link that exists between the PHS formulation and the existing mixed (i.e., multiple primary field) variational principles.

Indeed, in some other research communities in parallel, there is an equally strong effort to reformulate the classical theory of continuum mechanics in the (generalized) frames of the Hamiltonian/Lagrangian variational formalism and/or Hamiltonian canonical equation system [29,54–68]. Many of these works were motivated by the availability of GNI [29,35,69], i.e., time-stepping methods designed to exactly satisfy conservation laws, symmetries, or symplectic properties of a system of differential equations. In elastodynamics, in particular, various analytical methods and computational approaches using these formalisms for vibration analysis of (often cylindrical or spherical) shells have been identified and highlighted more recently in excellent review papers such as [57,70,71]. However, none have formulated the systems of PHS equations proposed here.

As regards the aforementioned 2D shell theories, in general, they are based on *a priori* behavioral assumptions. These are also often seen as generalizing features of flat plate theories [72], which are viewed as special formulation cases for shells having no curvature. The historical equations of these structural theories are richly discussed in Naghdi's book [72] or Soedel's book [73]. Currently, what it is accepted in the engineering and scientific community as key starting points of the theories of thin and moderately thick shells are the equations derived by Love(–Kirchhoff) and Koiter for the thin shell category [74] (see also [73,75]) and Naghdi and Reissner (NR) for the moderately thick shell category [72,76–80]. Love's extrapolated Kirchhoff's plate assumptions to shells of a generic geometry and developed what is commonly referred as "the first order approximation shell theory", the local kinematics of which is based on three displacement fields of translation and their gradients. Currently, most mathematical models and finite elements [55] usually considered for shell structures are based on the NR kinematical assumption, which locally is kinematically based on five displacement fields of translation and rotation. Limiting our present PHS analysis to this type of shell modeling, in order to keep the essential ideas in focus, attention will be further restricted hereafter to the transversely homogeneous, and tangentially isotropic, material case. Nevertheless, we still leave the stored (kinetic and elastic) energy density dependent only on the reference positions of the material particles on the mid-surface. At the end of the aforementioned reduction process, the resulting partial differential equations of the linear elastic NR shell theory in a port-Hamiltonian setting consider as well some intrinsic higher-order geometric coupling effects that are often overlooked.

**Organization of this manuscript.** This article is composed as follows. We begin in Section 2 with some brief, but fundamental concepts, such as the shell initial geometry and the curvilinear surface description. To keep the essential ideas in focus, we restricted our attention to the formulation in the principal curvature coordinates. We remind then in Section 3 about the main geometrical and kinematic concepts required to describe the NR shell structure models. Section 3.1 introduces notably the NR-constrained displacement–strain relationship. The essential features of the classical 2D shell NR formulations are then obtained in Section 3.2 by using D'Alembert–Lagrange's virtual power formalism. The latter provides a most expedient and precise reduction route to the fundamental dynamic equations of the evolution and edge (boundary) conditions, while exhibiting the main sthenical variables. In order to obtain their expressions in terms of the displacement fields, the standard (but modified) Hooke's linear isotropic elastic law for the stress–strain relationship is adopted for the 2D shell constitutive material. For completeness, a short discussion highlights both its well-known and possibly less-well-known limitations. Throughout our development, we provide the mechanical and geometrical interpretation of the 2D shell variables such as the strain, stress, and external loads applied on the 2D shell model in correlation with their 3D counterparts. In addition to the specific features of the NR's (mixed) first-order shell model, some remarks regarding the (mixed) higher-order shell theories are also provided to briefly assess their coupling or uncoupling effects. The essential variational features of the PHS formulation of the NR shell model are then introduced in Section 4. The equations are formally re-exposed via the Hamilton(–Lagrange–D'Alembert) generalized variational principle with the intent to add physical (engineering) insights to

the fundamental formulation of the PHS of the 2D shell and surface theories. Then, we discuss the relevance of these partially or totally variational PHS formalisms, based on some works already existing in the literature. The last Section 5 concludes this communication with some remarks related to our future applications.

**Notation and conventions.** Before proceeding, we summarize some notational conventions. Hereafter, we use Einstein's summation convention for the repeated indexes (only) between two or more terms, with the lower case indexes spanning the set $\{1, 2, 3\}$ and the higher case indexes spanning the set $\{1, 2\}$. Moreover, we use bold-font characters for the n-uplets like the spatial parameters $\boldsymbol{\alpha} = (\alpha_1, \alpha_2, \alpha_3)$ or $\bar{\boldsymbol{\alpha}} = (\alpha_1, \alpha_2)$, for the vectors or first-order tensors like $\nabla$, $\overline{\nabla}$, $\boldsymbol{e}_i$ and $\boldsymbol{u} = u_i \boldsymbol{e}_i$, and for the matrices or second-order tensors like $\boldsymbol{A} = A_{ij} \boldsymbol{e}_i \otimes \boldsymbol{e}_j$. Here, "$\otimes$" denotes the tensorial (outer) product, while "$\times$" denotes the vectorial (cross) product. The dot (or inner) product of vectors and the simple contraction of tensors are denoted "$\cdot$", while the double-contraction of tensors is denoted "$:$". Thus, for instance, if $(\boldsymbol{e}_i)_{i=1,2,3}$ forms the basis of orthonormal vectors, simple and double-contractions of the aforementioned first-order tensor $\boldsymbol{u}$ or second-order tensor $\boldsymbol{A}$ with another one $\boldsymbol{B} = B_{kl} \boldsymbol{e}_k \otimes \boldsymbol{e}_l$ are, respectively, written as

$$
\begin{aligned}
\boldsymbol{u} \cdot \boldsymbol{B} &= u_i B_{kl} (\boldsymbol{e}_i \cdot \boldsymbol{e}_k) \boldsymbol{e}_l \equiv u_i B_{il} \boldsymbol{e}_l \\
\boldsymbol{B} \cdot \boldsymbol{u} &= B_{kl} u_i (\boldsymbol{e}_l \cdot \boldsymbol{e}_i) \boldsymbol{e}_k \equiv u_i B_{ki} \boldsymbol{e}_k \\
\boldsymbol{A} \cdot \boldsymbol{B} &= A_{ij} B_{kl} (\boldsymbol{e}_j \cdot \boldsymbol{e}_k) (\boldsymbol{e}_i \otimes \boldsymbol{e}_l) \equiv A_{ij} B_{jl} \boldsymbol{e}_i \otimes \boldsymbol{e}_l \\
\boldsymbol{A} : \boldsymbol{B} &= A_{ij} B_{kl} (\boldsymbol{e}_j \cdot \boldsymbol{e}_k) (\boldsymbol{e}_i \cdot \boldsymbol{e}_l) \equiv A_{ij} B_{ji} \, .
\end{aligned}
$$

Besides, the superscript T symbol "$(\ )^{\mathrm{T}}$" is used as the transposition operator for both matrices and tensors; the over-lined symbol "$\overline{(\ )}$" is related to the mid-surface whenever the same symbols, functions, or fields are used also for similar three-dimensional entities; "$(\ )'$" refers to the deformed body to clearly distinguish its rest configuration. Finally, the superposed dot symbol "$\dot{(\ )}$" and the subscripted tilde symbol "$(\ )_{\sim}$" denote, respectively, the total (material) time derivative with respect to the time parameter $t$ for the former and the virtual quantity for the latter, both arising while holding the spatial parameters $\boldsymbol{\alpha}$ fixed. Sometimes, for the sake of clarity, the dependence of the sthenical and kinematic quantities on the spatial and temporal variables will be considered implicitly. Lately, we should also consider functionals $\mathscr{F}[\boldsymbol{a}, \boldsymbol{b}, \boldsymbol{c}, \cdots]$ of field column matrices $[\boldsymbol{a}, \boldsymbol{b}, \boldsymbol{c}, \cdots]$ (of various sizes), which are defined over a variable space $\mathcal{X}$. Then, we denote the Gateaux functional (or variational) directional derivatives of $\mathscr{F}[\boldsymbol{a}, \boldsymbol{b}, \boldsymbol{c}, \cdots]$ in the directions $\underset{\sim}{\boldsymbol{a}}$ and $\underset{\sim}{\boldsymbol{c}}$ as $\delta_a \mathscr{F}[\boldsymbol{a}, \boldsymbol{b}, \boldsymbol{c}, \cdots]$ and $\delta_c \mathscr{F}[\boldsymbol{a}, \boldsymbol{b}, \boldsymbol{c}, \cdots]$ (respectively) and obtain them (simultaneously or not) from (total or partial) Dini's variations as in [61]:

$$
\lim_{\epsilon \searrow 0^+} \frac{d}{d\epsilon} \int_{\mathcal{X}} \mathscr{F}[\boldsymbol{a} + \epsilon \, \underset{\sim}{\boldsymbol{a}}, \boldsymbol{b}, \boldsymbol{c} + \epsilon \, \underset{\sim}{\boldsymbol{c}}, \cdots] \, d\mathcal{X} \overset{\text{def}}{=} \int_{\mathcal{X}} \{ \underset{\sim}{\boldsymbol{a}}^{\mathrm{T}} \delta_a \mathscr{F}[\boldsymbol{a}, \boldsymbol{b}, \boldsymbol{c}, \cdots]
$$
$$
+ \underset{\sim}{\boldsymbol{c}}^{\mathrm{T}} \delta_c \mathscr{F}[\boldsymbol{a}, \boldsymbol{b}, \boldsymbol{c}, \cdots] + \cdots \} \, d\mathcal{X} \, .
$$

We also repeatedly use the Kronecker delta symbol $\delta_{IJ}$ and the Levi-Civita 2D permutation (or the alternating) one:

$$
\epsilon_{IJ} \overset{\text{def}}{=} \det \begin{pmatrix} \delta_{I1} & \delta_{I2} \\ \delta_{J1} & \delta_{J2} \end{pmatrix} \equiv \delta_{I1} \delta_{J2} - \delta_{I2} \delta_{J1} \equiv \begin{cases} 1 & , \quad \text{if } (I, J) = (1, 2) \\ -1 & , \quad \text{if } (I, J) = (2, 1) \\ 0 & , \quad \text{if } I = J \in \{1, 2\} \end{cases} \tag{1}
$$

with which $\epsilon_{IJ} \epsilon_{KJ} \equiv \delta_{IJ} \delta_{JK} = \delta_{IK}$, $\epsilon_{IJ} \epsilon_{IJ} \equiv \delta_{IJ} \delta_{IJ} = 2$ and $|\epsilon_{IJ}| \equiv 1 - \delta_{IJ}$.

Further notations will be provided when required.

## 2. The Undeformed Reference Geometry of the Shell

**Local parametrization of the undeformed shell geometry.** The material (or Lagrangian) configuration space of the considered (shell) material body $\mathcal{V}$ is described in

Cartesian coordinates with respect to the inertial frame $(O, \boldsymbol{b}_1, \boldsymbol{b}_2, \boldsymbol{b}_3)$ (of the global coordinate system). The latter is defined with the direct orthonormal vector basis $(\boldsymbol{b}_1, \boldsymbol{b}_2, \boldsymbol{b}_3)$ (cf. Figure 1) of the three-dimensional Euclidean space $\mathbb{R}^3$. At equilibrium rest, the reference placements $\mathcal{V}$ of the shell, of its mid-surface $\overline{\mathcal{S}}$, and of its Lipschitz-continuous contour $\overline{\mathcal{C}}$ are described as embedded in $\mathbb{R}^3$ like

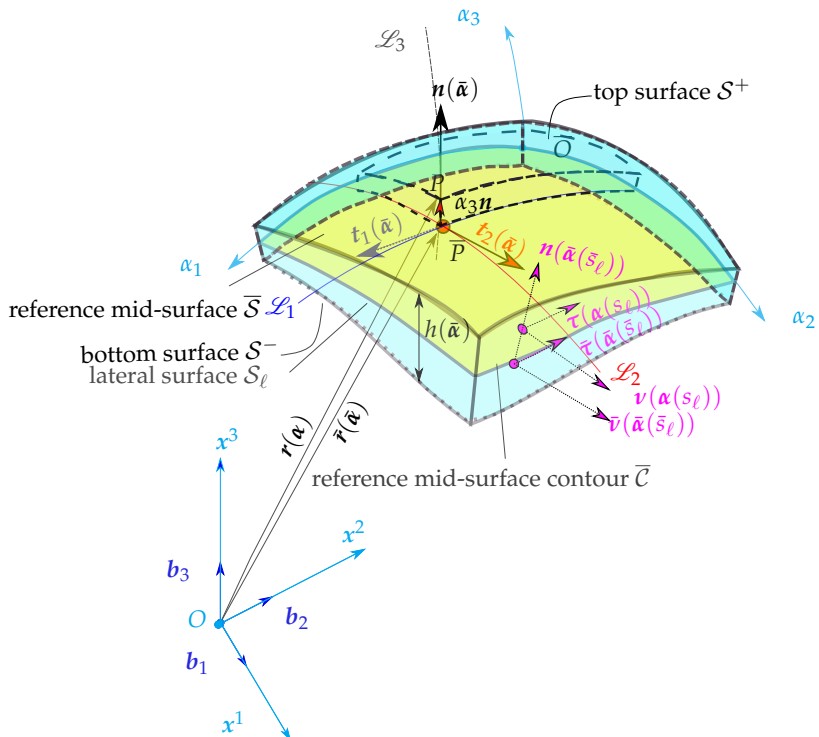

**Figure 1.** Generic doubly curved shell model and its coordinate systems. A point $P$ in the shell $\mathcal{V}$ is defined as the intersection of three curves $\mathscr{L}_i$ (with $i = 1, 2, 3$) obtained by varying only the parameter $\alpha_i$ at a time, along the principal curvature directions. The related point $\overline{P}$ in the shell mid-surface $\overline{\mathcal{S}}$ (corresponding to $\alpha_3 = 0$) is defined at the intersection of the two curves $\mathscr{L}_I$ (with $I = 1, 2$) of the mid-surface $\overline{\mathcal{S}}$. The (initial and rest) thickness $h$ along the line $\mathcal{L}_3$ is considered relatively (much) smaller than the two others ($\mathscr{L}_1$ and $\mathscr{L}_2$) or than the smaller principal radius of curvature $\min|R_n|$ (introduced hereafter) of $\overline{\mathcal{S}}$. In this work, the local (unstretchable) director passing by the material points $\overline{P}$ and $P$ is merely considered collinear to the unit-normal vector $\boldsymbol{n}$ of $\overline{\mathcal{S}}$ in the rest reference state.

$$
\begin{aligned}
\mathcal{V} &\overset{\text{def}}{=} \left\{ \boldsymbol{r}(\boldsymbol{\alpha}) = \boldsymbol{x}(\boldsymbol{\alpha}) = x^i(\boldsymbol{\alpha})\boldsymbol{b}_i \in \mathbb{R}^3 \,, \right. \\
&\qquad \left. \boldsymbol{\alpha} \overset{\text{def}}{=} (\alpha_1, \alpha_2, \alpha_3) \in \Omega \times [-h/2, h/2] \right\} \\
\overline{\mathcal{S}} &\overset{\text{def}}{=} \left\{ \overline{\boldsymbol{r}}(\bar{\boldsymbol{\alpha}}) \overset{\text{def}}{=} \boldsymbol{r}(\alpha_1, \alpha_2, 0) \in \mathbb{R}^3 \,, \bar{\boldsymbol{\alpha}} \overset{\text{def}}{=} (\alpha_1, \alpha_2) \in \Omega \right\} \subset \mathcal{V} \\
\overline{\mathcal{C}} &\overset{\text{def}}{=} \left\{ \overline{\boldsymbol{r}}(\bar{\boldsymbol{\alpha}}(\bar{s}_\ell)) \overset{\text{def}}{=} \boldsymbol{r}(\alpha_1(\bar{s}_\ell), \alpha_2(\bar{s}_\ell), 0) \in \mathbb{R}^3 \,, \right. \\
&\qquad \left. \bar{\boldsymbol{\alpha}}(\bar{s}_\ell) \overset{\text{def}}{=} (\alpha_1(\bar{s}_\ell), \alpha_2(\bar{s}_\ell)) \in \Omega \right\} \equiv \partial\overline{\mathcal{S}} \subset \partial\mathcal{V} \,.
\end{aligned} \tag{2}
$$

Here, in the first domain definition, the position of the suffix (superscript or subscript) in the coordinate $x^i(\boldsymbol{\alpha})$ is, of course, irrelevant, since the basis $(\boldsymbol{b}_1, \boldsymbol{b}_2, \boldsymbol{b}_3)$ is Cartesian; this will also be the case for the forthcoming orthogonal/principal basis $(\boldsymbol{t}_1, \boldsymbol{t}_2, \boldsymbol{n})$. Besides, $\Omega$ is a bounded parametric domain in $\mathbb{R}^3$ (respectively, $\mathbb{R}^2$, spanned by the scalar variable pair of curvilinear coordinates $\bar{\boldsymbol{\alpha}}$ parameterizing $\overline{\mathcal{S}}$, while $\alpha_3 \in [-h/2, h/2]$ denotes the shell through thickness coordinate in the shell transverse direction. The thickness of that latter is a relatively small (with respect to both the principal radius of curvature $\min|R_n|$ introduced hereafter and the shell in-plane characteristic lengths), but strictly positive—and possibly varying—function $h(\bar{\boldsymbol{\alpha}})$ of $\bar{\boldsymbol{\alpha}}$. The locations of a point $P$ in the shell $\mathcal{V}$ and a point $\overline{P}$ on the

neutral surface $\overline{S}$ are, respectively, expressed by vectors $r(\alpha)$ and $\bar{r}(\bar{\alpha})$. $\bar{s}_\ell$ is a curvilinear abscissa parameterizing and orienting the existing mid-surface boundary contour $\overline{C}$ if the shell mid-surface $\overline{S}$ is not closed (i.e., the shell does not enclose any simply connected volume domain). Besides, in that case, we shall consider as well that the shell may possibly have a lateral surface boundary denoted as

$$S_\ell \overset{\text{def}}{=} \overline{C} \times [-h/2, h/2] \subset \partial \mathcal{V} \quad \text{(and so } \overline{C} \equiv S_\ell \cap \overline{S}\text{)}, \tag{3}$$

which is assumed ruled by the unit normal vector of the mid-surface $\overline{S}$. The full boundary of the shell $\partial \mathcal{V} = S_- \cup S_+ \cup S_\ell$ is furthermore composed then of an upper (respectively, lower) bounding surface $S_+ \overset{\text{def}}{=} \overline{S} \times \{\alpha_3 = h/2\}$ (respectively, $S_- \overset{\text{def}}{=} \overline{S} \times \{\alpha_3 = -h/2\}$), which are, therefore, parallel with the mid-surface $\overline{S}$. The account of such surfaces is important as well for analyses of non-smooth shell structures containing junctions and/or self-intersections.

From some mathematical and engineering practices, though, the foregoing geometrical (slenderness) condition allows implementing simplifications in the 3D elasticity equations and transforming the problem into a 2D problem where the equations governing its mechanical behavior are solved with respect to the reference curvilinear surface $\overline{S}$, which is endowed then with certain mechanical properties. The equations required to define the shell geometry and mechanical behavior will be partial differential equations along the curvilinear surface, and hence, the derivatives along parametric curves must be properly defined. In order to do so, it is necessary to derive the surface metric relationships (or first fundamental form) and the surface curvature relationships (or second fundamental form). For self-completeness, some of these important formulas of the differential theory of curvilinear surfaces are therefore developed hereafter.

**Darboux–Ribaucour's geodesic frame on the shell mid-surface $\overline{S}$.** In order to measure features such as curvatures, deformation, etc., the shell geometry is conventionally defined on convected, covariant (and contravariant) curvilinear local basis systems induced by the parametrization $\bar{r}(\bar{\alpha})$ of the material points $\overline{P} \in \overline{S}$, instead of the former classical Cartesian coordinate system related to $(O, b_1, b_2, b_3)$. For this, we introduce the metric tensor coefficients, which define the scale factors (also known as Lamé's parameters) as $A_I \overset{\text{def}}{=} \|\frac{\partial \bar{r}}{\partial \alpha_I}\|$ and, on the other part, the pair of unit-tangent vectors $t_I \overset{\text{def}}{=} \frac{1}{A_I}\frac{\partial \bar{r}}{\partial \alpha_I} \equiv \frac{\partial \bar{r}}{\partial \bar{s}_I}$ for $I = 1, 2$. On $\overline{S}$, while $\mathscr{L}_1$ (respectively, $\mathscr{L}_2$) is defined at constant $\alpha_2$ (respectively, $\alpha_1$), as illustrated in Figure 1, $A_I$ provides the measure of length (per units of $\bar{s}_I$) along the isoparametric curve $\mathscr{L}_I$ passing by the local point $\overline{P} \in \overline{S}$. For simplicity, we assumed the parametric curves $(\mathscr{L}_1, \mathscr{L}_2)$ oriented by $(t_1, t_2)$ coincide with the principal lines of curvature of $\overline{S}$, which can be defined at points where $\bar{r}(\bar{\alpha})$ is twice continuously differentiable. We can identify then the mid-surface unit normal vector $n \equiv t_1 \times t_2$ and choose (as illustrated in Figure 2) the geodesic normal direction of each curve $\mathscr{L}_I$ as $g_I \overset{\text{def}}{=} n \times t_I \equiv \epsilon_{IJ}t_J$. The Darboux–Ribaucour formula for each principal curvature line $\mathscr{L}_I$ of $\overline{S}$ reads then

$$\frac{\partial}{\partial \bar{s}_I} \left\{ \begin{array}{c} t_I \\ g_I \\ n \end{array} \right\} = \left( \begin{array}{ccc} 0 & \dfrac{-1}{R_{g_I}} & \dfrac{-1}{R_{n_I}} \\ \dfrac{1}{R_{g_I}} & 0 & \dfrac{-1}{T_{g_I}} \\ \dfrac{1}{R_{n_I}} & \dfrac{1}{T_{g_I}} & 0 \end{array} \right) \left\{ \begin{array}{c} t_I \\ g_I \\ n \end{array} \right\} \text{ with } \frac{1}{T_{g_I}} = 0 \text{, for } I = 1, 2, \tag{4}$$

while denoting the radius of principal normal curvature as $|R_{n_I}|$, the radius of geodesic curvature as $|R_{g_I}|$, and the geodesic (or geodetic) torsion as $|T_{g_I}|$, which is infinite along each principal curvature line $\mathscr{L}_I$. For completeness, the following simplified tensor expressions

of the Weingarten's and Gauss' formulas are also mentioned with the mid-surface–covariant gradient operator $\overline{\nabla}$ introduced hereafter in Equation (18):

$$\overline{\nabla}\boldsymbol{n} = \frac{\boldsymbol{t}_I \otimes \boldsymbol{t}_I}{R_{n_I}} \qquad \text{(Weingarten)}$$

$$\overline{\nabla}\boldsymbol{t}_I = \epsilon_{JI}\left[\frac{\boldsymbol{t}_J \otimes \boldsymbol{t}_J}{R_{g_J}} + \frac{\boldsymbol{t}_I \otimes \boldsymbol{t}_J}{R_{g_I}}\right] - \frac{\boldsymbol{t}_I \otimes \boldsymbol{n}}{R_{n_I}} \text{ , for } J = 1, 2 \quad \text{(Gauss)} .$$

(5)

Naturally, those expressions are defined at regular material points $\overline{P} \in \overline{\mathcal{S}}$. Singular cases where the normal $\boldsymbol{n}$ (and the director $\boldsymbol{n}'$) can be discontinuous, as at shell junctions/intersections, have been considered in other works; see e.g., [64].

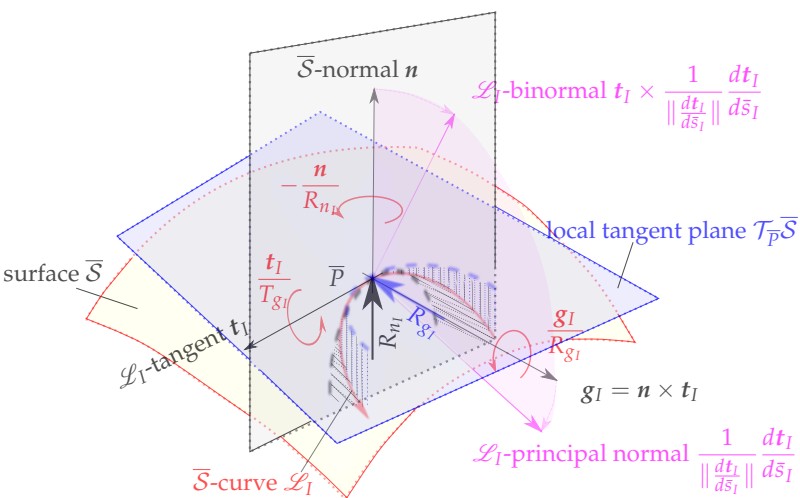

**Figure 2.** Darboux–Ribaucour's local frame $(\overline{P}, \boldsymbol{t}_I, \boldsymbol{g}_I, \boldsymbol{n})$.

With respect to the previous description and Figure 1, the three coordinates $\mathbf{r}(\boldsymbol{\alpha})$ of any material point $P$ of the shell in its reference configuration $\mathcal{V}$ can be defined then with the two curvilinear surface coordinates and the normal to the surface coordinate as

$$\mathbf{r}(\boldsymbol{\alpha}) = \overline{\mathbf{r}}(\overline{\boldsymbol{\alpha}}) + \alpha_3\,\boldsymbol{n}(\overline{\boldsymbol{\alpha}}) .$$

(6)

The following Lamé (metric) coefficients (with respect to the *principal* coordinate system) are used on the parallel surface passing by that point $P$:

$$B_I \overset{\text{def}}{=} \|\frac{\partial \boldsymbol{r}}{\partial \alpha_I}\| \equiv A_I\left(1 + \frac{\alpha_3}{R_{n_I}}\right) \text{ for } I = 1, 2 ; \quad \text{and } B_3 \overset{\text{def}}{=} \|\frac{\partial \boldsymbol{r}}{\partial \alpha_3}\| \equiv 1 .$$

(7)

**Darboux–Ribaucour's geodesic frame on the lateral surface $\mathcal{S}_\ell$.** A complete analysis of the non-closed shell dynamics also requires a detail geometric description of the contour $\overline{\mathcal{C}} = \overline{\mathcal{S}}_\ell \cap \overline{\mathcal{S}}$ of the reference mid-surface boundary (cf. Figure 1). Similar to [76] (page 39), that contour $\overline{\mathcal{C}}$ that consists of a finite set of piecewisely smooth curves will be described with another orthonormal basis $(\overline{\boldsymbol{\tau}}, \overline{\boldsymbol{\nu}}, \boldsymbol{n})$ formed with a couple $(\overline{\boldsymbol{\tau}}, \overline{\boldsymbol{\nu}})$ of unit-tangential and outward-pointing-normal basis vectors (both being tangent though the mid-surface $\overline{\mathcal{S}}$ and existing almost everywhere along the Lipschitz-continuous contour $\overline{\mathcal{C}}$). These provide the directional cosines $\overline{\tau}_I \equiv \overline{\boldsymbol{\tau}} \cdot \boldsymbol{t}_I$ and $\overline{\nu}_I \equiv \overline{\boldsymbol{\nu}} \cdot \boldsymbol{t}_I$. We introduce as well the following metric coefficients on $\mathcal{S}_\ell$:

$$B_\tau \overset{\text{def}}{=} \overline{\boldsymbol{\tau}} \cdot \frac{d\boldsymbol{r}}{d\overline{s}_\ell} \equiv \left(1 + \frac{\alpha_3}{R_{g_\tau}}\right) \quad \text{and} \quad B_\nu \overset{\text{def}}{=} \overline{\boldsymbol{\nu}} \cdot \frac{d\boldsymbol{r}}{d\overline{s}_\ell} \equiv -\frac{\alpha_3}{T_{g_\nu}}$$

(8)

and the Darboux–Ribaucour formula at any regular point of $\overline{C}$:

$$
\frac{d}{d\bar{s}_\ell}
\left\{
\begin{array}{c}
\bar{\tau} \\
n \\
\bar{v}
\end{array}
\right\}
=
\left(
\begin{array}{ccc}
0 & \dfrac{-1}{R_{g\tau}} & \dfrac{-1}{R_{n_v}} \\[2mm]
\dfrac{1}{R_{g\tau}} & 0 & \dfrac{-1}{T_{g_v}} \\[2mm]
\dfrac{1}{R_{n_v}} & \dfrac{1}{T_{g_v}} & 0
\end{array}
\right)
\left\{
\begin{array}{c}
\bar{\tau} \\
n \\
\bar{v}
\end{array}
\right\}.
\tag{9}
$$

These ones are related to the radii $|R_{g\tau}|$ and $|T_{g_v}|$ of the geodesic curvature and geodesic torsion of the contour $\overline{C} = \mathcal{S}_\ell \cap \overline{\mathcal{S}}$, which are defined, respectively, along the geodesic normal direction $\bar{v} \times \bar{\tau} \equiv n$ and the normal direction $\bar{v}$ of the Darboux–Ribaucour geodesic frame $(\overline{P}, \bar{\tau}, n, \bar{v})$ of the mid-line contour $\overline{C}$ of the lateral surface $\mathcal{S}_\ell$.

### 3. Classical NR Shell Theory for Moderately Thick, Linear Elastic, Shells

The following subsections remind about the main ingredients for the classical NR theory for *moderately thick* shells [72,76–80] with higher-order *intrinsic geometric coupling effects*, which will be relevant then for our PH formulation purposes. The nontrivial purpose of such a classical 2D shell theory is to provide simplified (surface-based) equations while accounting for the main representative kinematics of the shell material points and the corresponding effective (i.e., integrated) dual loadings [81].

*3.1. Standard Infinitesimal Deformation Kinematics (For Transversely Rigid Shells)*

From here on, all functions and quantities associated with the deformed (or again, strained) shell configuration will be distinguished with a prime symbol ($'$). In addition, we omit writing the time variable $t$ for notational convenience.

Within the classical NR theory, the admissible kinematics of the shell $\mathcal{V}$ are merely described by a displacement field $u$ that is affine in $\alpha_3$. That field relies notably on restrained infinitesimal translational motions of the shell mid-surface $\overline{\mathcal{S}}$ and restrained infinitesimal rigid motions of a unit (*unstretchable*) vector $n'(\bar{\alpha})$, called the *director*, following the literature on Cosserat (micropolar or oriented) media [72,82,83]. More explicitly, we restrict as

$$
\boldsymbol{\theta}(\bar{\alpha}) = \theta_I(\bar{\alpha})\, t_I(\bar{\alpha}) \equiv \|\boldsymbol{\theta}(\bar{\alpha})\| n(\bar{\alpha}) \times n'(\bar{\alpha}) \text{ and so that } \boldsymbol{\theta}(\bar{\alpha}) \cdot n(\bar{\alpha}) = 0 \,, \text{ for } \bar{\alpha} \in \Omega \,,
\tag{10}
$$

the infinitesimal rotational (or angular) displacement vector that allows accounting for the rotation of the director from its reference (rest) orientation $n$ to its current (displaced) orientation $n' \approx n + \boldsymbol{\theta} \times n$ in the NR model. Here, it was assumed that there is no (drilling) rotation about the normal $n$. Each $\theta_I = \boldsymbol{\theta} \cdot t_I$ represents, therefore, the *infinitesimal angle of rotation* by which the director $n'$ (that was assumed collinear at rest to the normal $n$ of the middle surface $\overline{\mathcal{S}}$) rotates about the tangent $t_I$ to the $\alpha_I$ coordinate line $\mathcal{L}_I$. Sometimes, it can also be convenient for conciseness to merely introduce these infinitesimal rotations of the fibers that were at rest orthogonal to the mid-surface with the vector of the (*sine of*) *infinitesimal angle of rotation* of the director:

$$
\boldsymbol{\beta}(\bar{\alpha}) \equiv \beta_I(\bar{\alpha}) t_I(\bar{\alpha}) \stackrel{\text{def}}{=} \boldsymbol{\theta}(\bar{\alpha}) \times n(\bar{\alpha}) \,, \text{ with } \beta_I(\bar{\alpha}) \equiv \epsilon_{IJ}\theta_J(\bar{\alpha}) \text{ for } I = 1, 2 \,.
\tag{11}
$$

Besides, the admissible translational displacements of a point $\overline{P} \in \overline{\mathcal{S}}$ from its reference (rest equilibrium) state position $\bar{r}(\bar{\alpha})$ to its current motion position $\bar{r}'(\bar{\alpha})$ (which is not necessarily located on the mid-surface of the deformed shell placement) are defined by (cf. Figure 3)

$$
\bar{r}'(\bar{\alpha}) - \bar{r}(\bar{\alpha}) \equiv \bar{u}(\bar{\alpha}) \stackrel{\text{def}}{=} \bar{u}_I(\bar{\alpha}) t_I(\bar{\alpha}) + \bar{u}_3(\bar{\alpha}) n(\bar{\alpha}) \,.
\tag{12}
$$

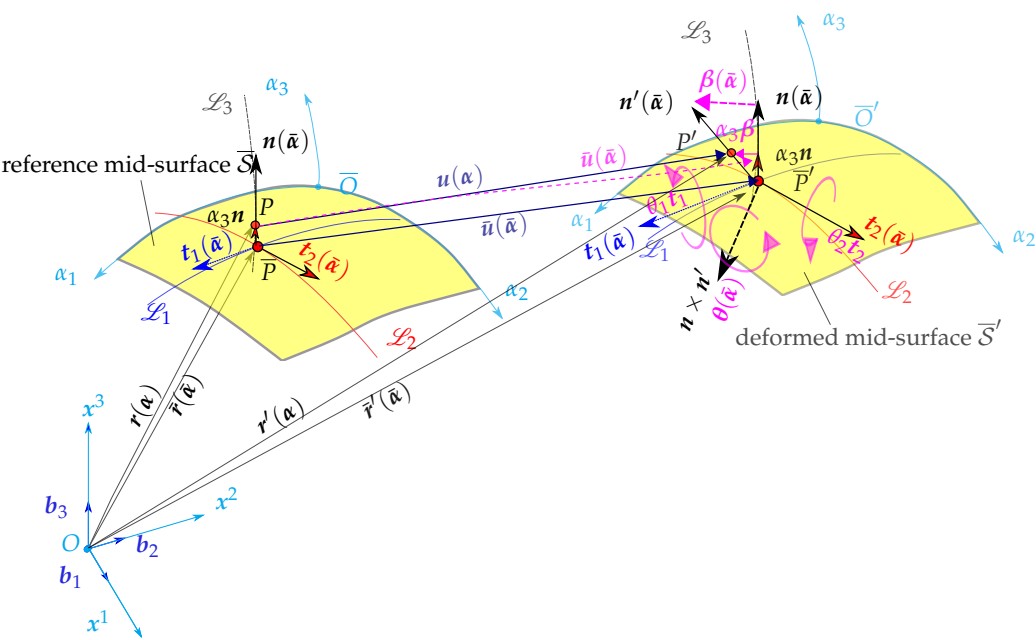

**Figure 3.** Admissible displacements $u$ of a shell material point $P \in \mathcal{S}$.

Meanwhile, the admissible translational displacements $u(\alpha)$ of a point $P \in \mathcal{V}$ from its reference state placement $r(\alpha)$ to a deformed state placement $r'(\alpha)$ take the ensuing ansatz for the expression:

$$r'(\alpha) - r(\alpha) \approx u(\alpha) \stackrel{\text{def}}{=} \bar{u}(\bar{\alpha}) + \alpha_3 \beta(\bar{\alpha}) \equiv u_I(\alpha) t_I(\bar{\alpha}) + u_3(\alpha) n(\bar{\alpha}) \,, \tag{13}$$

with its components in the local trihedron $(t_1, t_2, n)$ read explicitly as

$$u_I(\alpha) = \bar{u}_I(\bar{\alpha}) + \alpha_3 \beta_I(\bar{\alpha}) \equiv \bar{u}_I(\bar{\alpha}) + \alpha_3 \epsilon_{IJ} \theta_J(\bar{\alpha}) \,, \text{ for } I = 1, 2 \quad , \quad u_3(\alpha) = \bar{u}_3(\bar{\alpha}) \,. \tag{14}$$

Lately, as for the rest geometry configuration, we shall also introduce on the shell lateral surface $\mathcal{S}_\ell$ new natural (independent) kinematic boundary variables with respect to the Darboux–Ribaucour local geodesic frame $(\bar{r}, \bar{\tau}, n, \bar{v})$:

$$\begin{aligned} &\theta_\tau \stackrel{\text{def}}{=} \theta \cdot \bar{\tau} \quad , \quad \theta_v \stackrel{\text{def}}{=} \theta \cdot \bar{v} \quad , \quad \bar{u}_\tau \stackrel{\text{def}}{=} \bar{u} \cdot \bar{\tau} \quad \text{and} \quad \bar{u}_v \stackrel{\text{def}}{=} \bar{u} \cdot \bar{v} \\ &\text{so that } \theta = \theta_\tau \bar{\tau} + \theta_v \bar{v} \quad , \quad \bar{u} = \bar{u}_\tau \bar{\tau} + \bar{u}_v \bar{v} + \bar{u}_3 n \\ &\text{and} \quad u = \bar{u} + \alpha_3 \theta \times n \equiv \bar{u} + \alpha_3 [\theta_\tau \bar{v} - \theta_v \bar{\tau}] \,. \end{aligned} \tag{15}$$

It was assumed that some of the five boundary displacement components, if not all, are either free or constrained and imposed on some portions of $\bar{\mathcal{C}}$.

**Linearized strain–displacement relations.** In this work, the volumetric–covariant differentiation is denoted by a semi-colon subscript separator ( ; ), while the surface–covariant asymmetric gradient is denoted by a double-vertical bar subscript separator ( || ). The *volumetric(–covariant)* gradient of the displacement field $u$ in Equation (13) with respect to the undeformed system basis and coordinates over $\mathcal{S}$ can be expressed as

$$\nabla u = u_{I;J} t_J \otimes t_I + u_{I;3} n \otimes t_I + u_{3;I} t_I \otimes n + u_{3;3} n \otimes n \,. \tag{16}$$

The latter is composed of the following: *covariant derivatives*

$$
\begin{aligned}
u_{I;I} &\equiv \frac{A_I}{B_I} u_{I||I} = \frac{A_I}{B_I}\left(\frac{1}{A_I}\frac{\partial u_I}{\partial \alpha_I} + \frac{\epsilon_{IK} u_K}{R_{g_I}} + \frac{u_3}{R_{n_I}}\right)\\
u_{J;I} &\equiv \frac{A_I}{B_I} u_{J||I} = \frac{A_I}{B_I}\left(\frac{1}{A_I}\frac{\partial u_J}{\partial \alpha_I} - \frac{\epsilon_{IJ} u_I}{R_{g_I}}\right)\text{, for } I \neq J\\
u_{3;I} &\equiv \frac{A_I}{B_I} u_{3||I} = \frac{A_I}{B_I}\left(\frac{1}{A_I}\frac{\partial u_3}{\partial \alpha_I} - \frac{u_I}{R_{n_I}}\right) \equiv \frac{A_I}{B_I}\left(\frac{1}{A_I}\frac{\partial \bar{u}_3}{\partial \alpha_I} - \frac{\bar{u}_I + \alpha_3 \beta_I}{R_{n_I}}\right)\\
u_{I;3} &= \frac{\partial u_I}{\partial \alpha_3} \equiv \beta_I \equiv \epsilon_{IK}\theta_K\\
u_{3;3} &= \frac{\partial u_3}{\partial \alpha_3} \equiv \frac{\partial \bar{u}_3}{\partial \alpha_3} \equiv 0\,.
\end{aligned}
\tag{17}
$$

The expressions in Equation (17) exhibit as well for later use some existing relations with the components $\{u_{i||J}\}_{(i,J)\in\{1,2,3\}\times\{1,2\}}$ of the *surface–covariant derivatives* (over a surface parallel $\mathcal{S}$ with the mid-surface $\overline{\mathcal{S}}$ at a local signed distance $\alpha_3 \boldsymbol{n}$) and those forming the gradient evaluated on $\overline{\mathcal{S}}$:

$$
\overline{\nabla}\boldsymbol{u} = u_{I||J}\boldsymbol{t}_J \otimes \boldsymbol{t}_I + u_{3||I}\boldsymbol{t}_I \otimes \boldsymbol{n}\,.
\tag{18}
$$

Given the previous gradient expressions, the *linearized strain* tensor comes then as

$$
\boldsymbol{\varepsilon}^s[\boldsymbol{u}] \stackrel{\text{def}}{=} \frac{1}{2}\left[\nabla\boldsymbol{u} + (\nabla\boldsymbol{u})^{\mathrm{T}}\right] = \varepsilon^s_{IJ}[\boldsymbol{u}]\boldsymbol{t}_I \otimes \boldsymbol{t}_J + \varepsilon^s_{I3}[\boldsymbol{u}](\boldsymbol{t}_I \otimes \boldsymbol{n} + \boldsymbol{n} \otimes \boldsymbol{t}_I) + \varepsilon^s_{33}[\boldsymbol{u}]\boldsymbol{n} \otimes \boldsymbol{n}
$$
$$
\text{with } \varepsilon^s_{IJ}[\boldsymbol{u}] = \varepsilon^s_{IJ}[\bar{\boldsymbol{u}}] + \alpha_3 \kappa^s_{IJ}[\boldsymbol{\theta}]\quad,\quad 2\varepsilon^s_{I3}[\boldsymbol{u}] = \gamma_I[\boldsymbol{u}]\quad,\quad \varepsilon^s_{33}[\boldsymbol{u}] = 0\,.
\tag{19}
$$

This decomposition emphasizes separately the *membrane* (or *in-plane*, *tangent* in reference to the local tangent plane $\mathcal{T}_{\bar{P}}\overline{\mathcal{S}}$) strains $\bar{\varepsilon}^s_{IJ}$ (with $I = J$ for the *in-plane normal strains* and $I \neq J$ for the *tangential shear* ones, while $I, J \in \{1,2\}$):

$$
\varepsilon^s_{II}[\bar{\boldsymbol{u}}] \stackrel{\text{def}}{=} \bar{u}_{I;I} \quad\equiv\quad \frac{1}{B_I}\left[\frac{\partial \bar{u}_I}{\partial \alpha_I} + \bar{u}_K \frac{\epsilon_{IK}A_I}{R_{g_I}} + \bar{u}_3 \frac{A_I}{R_{n_I}}\right],
\tag{20a}
$$

$$
2\varepsilon^s_{IJ}[\bar{\boldsymbol{u}}] \stackrel{\text{def}}{=} \bar{u}_{I;J} + \bar{u}_{J;I} \quad\equiv\quad \frac{1}{B_J}\left[\frac{\partial \bar{u}_I}{\partial \alpha_J} + \bar{u}_J \frac{\epsilon_{IJ}A_J}{R_{g_J}}\right] + \frac{1}{B_I}\left[\frac{\partial \bar{u}_J}{\partial \alpha_I} + \bar{u}_I \frac{\epsilon_{JI}A_I}{R_{g_I}}\right],
\tag{20b}
$$

the *change-in-curvature strains* $\kappa^s_{IJ}$ (with $I = J$ for *bending* and $I \neq J$ for *torsion* or *twist*, while $I, J \in \{1,2\}$):

$$
\kappa^s_{II}[\boldsymbol{\theta}] \stackrel{\text{def}}{=} \beta_{I;I} \quad\equiv\quad \frac{1}{B_I}\left[\frac{\partial \beta_I}{\partial \alpha_I} + \beta_K \frac{\epsilon_{IK}A_I}{R_{g_I}}\right],
\tag{20c}
$$

$$
2\kappa^s_{IJ}[\boldsymbol{\theta}] \stackrel{\text{def}}{=} \beta_{I;J} + \beta_{J;I} \quad\equiv\quad \frac{1}{B_J}\left[\frac{\partial \beta_I}{\partial \alpha_J} + \beta_J \frac{\epsilon_{IJ}A_J}{R_{g_J}}\right] + \frac{1}{B_I}\left[\frac{\partial \beta_J}{\partial \alpha_I} + \beta_I \frac{\epsilon_{JI}A_I}{R_{g_I}}\right],
\tag{20d}
$$

and finally, the *transverse* (or *out-of-plane*) *shear strains* $\gamma_I$ (while $I \in \{1,2\}$):

$$
2\varepsilon^s_{I3}[\boldsymbol{u}] \equiv \gamma_I[\boldsymbol{u}] \stackrel{\text{def}}{=} u_{I;3} + u_{3;I} \quad\equiv\quad \frac{1}{B_I}\left[\frac{\partial \bar{u}_3}{\partial \alpha_I} + A_I\left(\beta_I - \frac{\bar{u}_I}{R_{n_I}}\right)\right].
\tag{20e}
$$

When evaluating these expressions on $\overline{\mathcal{S}}$ (where, therefore, $\alpha_3 = 0$ and $B_I = A_I$), we shall denote them as (while $I, J \in \{1,2\}$)

$$
\begin{aligned}
2\bar{\varepsilon}^s_{IJ}[\bar{\boldsymbol{u}}] &\stackrel{\text{def}}{=} \bar{u}_{I||J} + \bar{u}_{J||I}\,,\\
2\bar{\kappa}^s_{IJ}[\boldsymbol{\theta}] &\stackrel{\text{def}}{=} \beta_{I||J} + \beta_{J||I} \equiv \epsilon_{IK}\theta_{K||J} + \epsilon_{JK}\theta_{K||I}\,,\\
2\bar{\varepsilon}^s_{I3}[\mathbf{q}] \equiv \bar{\gamma}_I[\mathbf{q}] &\stackrel{\text{def}}{=} u_{I;3} + \bar{u}_{3||I} \equiv \beta_I + \bar{u}_{3||I} \equiv \epsilon_{IK}\theta_K + \bar{u}_{3||I}\,,
\end{aligned}
\tag{21}
$$

with therefore

$$\theta_{J||I} = \frac{1}{A_I}\frac{\partial \theta_J}{\partial \alpha_I} - \theta_I \frac{\epsilon_{IJ}}{R_{g_I}} + \theta_K \frac{\delta_{IJ}\epsilon_{IK}}{R_{g_I}} \ . \tag{22}$$

The transverse shear strain $\bar{\gamma}_I$ in Equation (21) is merely expressed here as a function of a column matrix q formed from the couple $(\bar{\boldsymbol{u}}, \boldsymbol{\theta})$ as, hereafter, in Equation (23).

**Matrix formulation of the linear strain–displacement relations of the NR shell model.** Of great interest for the PHS formulation is that the following "surface strain-based" matrix:

$$\overline{\mathbf{E}}[q] \overset{\text{def}}{=} \overline{\mathbb{B}}\, q \ , \quad \text{over } \overline{\mathcal{S}} \tag{23a}$$

which is locally related, by means of the $(10 \times 5)$ differential operator matrix

$$\overline{\mathbb{B}} \overset{\text{def}}{=} \begin{pmatrix} \frac{1}{A_2}\frac{\partial}{\partial \bar{s}_1} & 0 & \frac{1}{A_2 R_{g_1}} & 0 & \frac{1}{A_2 R_{n_1}} \\ \frac{-1}{A_1 R_{g_2}} & 0 & \frac{1}{A_1}\frac{\partial}{\partial \bar{s}_2} & 0 & \frac{1}{A_1 R_{n_2}} \\ \frac{1}{A_1}\frac{\partial}{\partial \bar{s}_2} & 0 & \frac{1}{A_1 R_{g_2}} & 0 & 0 \\ \frac{-1}{A_2 R_{g_1}} & 0 & \frac{1}{A_2}\frac{\partial}{\partial \bar{s}_1} & 0 & 0 \\ \frac{1}{A_2 R_{n_1}} & 0 & 0 & \frac{1}{A_2} & \frac{1}{A_2}\frac{\partial}{\partial \bar{s}_1} \\ 0 & \frac{-1}{A_1} & \frac{1}{A_1 R_{n_2}} & 0 & \frac{1}{A_1}\frac{\partial}{\partial \bar{s}_2} \\ 0 & \frac{1}{A_2}\frac{\partial}{\partial \bar{s}_1} & 0 & \frac{1}{A_2 R_{g_1}} & 0 \\ 0 & \frac{-1}{A_1 R_{g_2}} & 0 & \frac{1}{A_1}\frac{\partial}{\partial \bar{s}_2} & 0 \\ 0 & \frac{1}{A_1}\frac{\partial}{\partial \bar{s}_2} & 0 & \frac{1}{A_1 R_{g_2}} & 0 \\ 0 & \frac{-1}{A_2 R_{g_1}} & 0 & \frac{1}{A_2}\frac{\partial}{\partial \bar{s}_1} & 0 \end{pmatrix} \begin{matrix} \frac{\bar{u}_{1||1}}{A_2} \\ \frac{\bar{u}_{2||2}}{A_1} \\ \frac{\bar{u}_{1||2}}{A_1} \\ \frac{\bar{u}_{2||1}}{A_2} \\ \frac{\bar{\gamma}_1}{A_2} \\ \frac{\bar{\gamma}_2}{A_1} \\ \frac{\theta_{1||1}}{A_2} \\ \frac{\theta_{2||2}}{A_1} \\ \frac{\theta_{1||2}}{A_1} \\ \frac{\theta_{2||1}}{A_2} \end{matrix} \tag{23b}$$

with column headers $\bar{u}_1$, $\theta_1$, $\bar{u}_2$, $\theta_2$, $\bar{u}_3$.

the $(5 \times 1)$ (translation and rotation) displacement matrix to the $(10 \times 1)$ one composed of membrane and transverse gradients or strains and curvature changes:

$$q(\bar{\boldsymbol{\alpha}}) \overset{\text{def}}{=} \begin{Bmatrix} \bar{u}_1 \\ \theta_1 \\ \bar{u}_2 \\ \theta_2 \\ \bar{u}_3 \end{Bmatrix} , \quad \overline{\mathbf{E}}[q] \overset{\text{def}}{=} \begin{Bmatrix} \frac{\bar{u}_{1||1}}{A_2} \\ \frac{\bar{u}_{2||2}}{A_1} \\ \frac{\bar{u}_{1||2}}{A_1} \\ \frac{\bar{u}_{2||1}}{A_2} \\ \frac{\bar{\gamma}_1}{A_2} = \frac{\bar{u}_{3||1}+\theta_2}{A_2} \\ \frac{\bar{\gamma}_2}{A_1} = \frac{\bar{u}_{3||2}-\theta_1}{A_1} \\ \frac{\theta_{1||1}}{A_2} \equiv -\frac{\beta_{2||1}}{A_2} \\ \frac{\theta_{2||2}}{A_1} \equiv \frac{\beta_{1||2}}{A_1} \\ \frac{\theta_{1||2}}{A_1} \equiv -\frac{\beta_{2||2}}{A_1} \\ \frac{\theta_{2||1}}{A_2} \equiv \frac{\beta_{1||1}}{A_2} \end{Bmatrix} . \tag{23c}$$

(For convenience, the matrix $\overline{\mathbb{B}}$ in Equation (23) hereabove and some other matrices hereafter are described with the following information: on each line, each "red" variable on the right-hand side of the matrix is equal to the sum of the matrix-line coefficients applied (or multiplied) by the matrix-column top "blue" variables. In the energy expressions, each "blue" variable on the right-hand side of the matrix multiplies the coefficients of the matrix-line coefficients applied (or multiplied) by the column matrix top "blue" variables.)

For completeness and later use, the following kinematic column matrix is also introduced on the contour $\overline{\mathcal{C}}$:

$$\mathfrak{q}_{/\overline{\mathcal{C}}}\big(\bar{\boldsymbol{\alpha}}(\bar{s}_\ell)\big) \overset{\text{def}}{=} \left\{ \begin{array}{c} \bar{u}_\tau \\ \theta_\tau \\ \bar{u}_\nu \\ \theta_\nu \\ \bar{u}_3 \end{array} \right\}. \tag{24}$$

### 3.2. Reduction from 3D to 2D Shell Elasticity by Means of the Principle of Virtual Power

We intend now to remind about the weak and strong formal formulations of the 2D shell governing equations. As is well known, our simplification task can formally be achieved then by considering, for instance, the *variational principle* linking the *acceleration*, *"external"*, and *"internal"* *virtual powers* [84,85] (referred to hereafter, respectively, as **AVP**, **EVP**, and **IVP**) as

$$\mathscr{P}^{\text{acc}}[\boldsymbol{u},\underset{\sim}{\boldsymbol{u}}] = \mathscr{P}^{\text{ext}}[\boldsymbol{u},\underset{\sim}{\boldsymbol{u}}] + \mathscr{P}^{\text{int}}[\boldsymbol{u},\underset{\sim}{\boldsymbol{u}}] \ \text{ for all k.a. } \underset{\sim}{\boldsymbol{u}} \tag{25}$$

with the *kinematic admissible* (*k.a.*) virtual displacements following, in our case, Equations (13) and (14) as

$$\underset{\sim}{\boldsymbol{u}}(\boldsymbol{\alpha}) \overset{\text{def}}{=} \underset{\sim}{\bar{\boldsymbol{u}}}(\bar{\boldsymbol{\alpha}}) + \alpha_3 \underset{\sim}{\boldsymbol{\beta}}(\bar{\boldsymbol{\alpha}}) \text{ where } \underset{\sim}{\boldsymbol{\beta}}(\bar{\boldsymbol{\alpha}}) \overset{\text{def}}{=} \underset{\sim}{\boldsymbol{\theta}}(\bar{\boldsymbol{\alpha}}) \times \boldsymbol{n}(\bar{\boldsymbol{\alpha}}) , \tag{26}$$

the components of which are, of course, analogous to those in Equations (11)–(15), and then, finally, by enforcing

$$\mathscr{P}^*[\boldsymbol{u},\underset{\sim}{\boldsymbol{u}}] \equiv \overline{\mathscr{P}}^*[\mathfrak{q},\underset{\sim}{\mathfrak{q}}] \text{ while } \overline{\mathscr{P}}^*[\mathfrak{q},\underset{\sim}{\mathfrak{q}}] \equiv \overline{\mathscr{P}}^*_{/\overline{\mathcal{S}}}[\mathfrak{q},\underset{\sim}{\mathfrak{q}}] + \overline{\mathscr{P}}^*_{/\overline{\mathcal{C}}}[\mathfrak{q},\underset{\sim}{\mathfrak{q}}] . \tag{27}$$

Here, the substituted superscript symbol ($*$) underlies either *"acc"*, *"int"* or *"ext"*; moreover, while assuming no inertia for the shell surface $\partial\mathcal{V}$, we anticipated the fact that the related (virtual) powers can be split into bulk and boundary contributions with

$$
\begin{aligned}
\overline{\mathscr{P}}^*_{/\overline{\mathcal{S}}}[\mathfrak{q},\underset{\sim}{\mathfrak{q}}] & \overset{\text{def}}{=} & \int_{\overline{\mathcal{S}}} \Big[ \mathcal{Q}^*_{\bar{u}_i} \underset{\sim}{\bar{u}}_i + \mathcal{Q}^*_{\theta_I} \underset{\sim}{\theta}_I \Big] d\overline{\mathcal{S}} \equiv \int_{\overline{\mathcal{S}}} \underset{\sim}{\mathfrak{q}}^{\text{T}} \mathcal{Q}^*_{\mathfrak{q}} d\overline{\mathcal{S}} \\
\text{where} & & d\overline{\mathcal{S}} \overset{\text{def}}{=} A_1 A_2 \, d\alpha_1 \, d\alpha_2 \equiv d\bar{s}_1 \, d\bar{s}_2 \\
\overline{\mathscr{P}}^*_{/\overline{\mathcal{C}}}[\mathfrak{q},\underset{\sim}{\mathfrak{q}}] & \overset{\text{def}}{=} & \oint_{\overline{\mathcal{C}}} \Big[ \mathcal{Q}^*_{\bar{u}_\tau} \underset{\sim}{\bar{u}}_\tau + \mathcal{Q}^*_{\bar{u}_\nu} \underset{\sim}{\bar{u}}_\nu + \mathcal{Q}^*_{\bar{u}_3} \underset{\sim}{\bar{u}}_3 + \mathcal{Q}^*_{\theta_\tau} \underset{\sim}{\theta}_\tau + \mathcal{Q}^*_{\theta_\nu} \underset{\sim}{\theta}_\nu \Big] d\bar{s}_\ell \equiv \oint_{\overline{\mathcal{C}}} \underset{\sim}{\mathfrak{q}}^{\text{T}} \mathcal{Q}^*_{\mathfrak{q}} d\bar{s}_\ell .
\end{aligned} \tag{28}
$$

Their expressions involve the energetic dual entities to specify

$$\mathcal{Q}^*_{\mathfrak{q}}[\mathfrak{q}] \overset{\text{def}}{=} \left\{ \begin{array}{c} \mathcal{Q}^*_{\bar{u}_1} \\ \mathcal{Q}^*_{\theta_1} \\ \mathcal{Q}^*_{\bar{u}_2} \\ \mathcal{Q}^*_{\theta_2} \\ \mathcal{Q}^*_{\bar{u}_3} \end{array} \right\} \text{ over } \overline{\mathcal{S}} \ \text{ and } \ \mathcal{Q}^*_{\mathfrak{q}}[\mathfrak{q}] \overset{\text{def}}{=} \left\{ \begin{array}{c} \mathcal{Q}^*_{\bar{u}_\tau} \\ \mathcal{Q}^*_{\theta_\tau} \\ \mathcal{Q}^*_{\bar{u}_\nu} \\ \mathcal{Q}^*_{\theta_\nu} \\ \mathcal{Q}^*_{\bar{u}_3} \end{array} \right\} \text{ over } \overline{\mathcal{C}}. \tag{29}$$

These represent generalized macro-forces and micro-couples, or again, inertial, internal, and external forces and couples that are associated with the generalized local *"macro-scale translations"* $\bar{\boldsymbol{u}}$ and the local *"micro-scale rotations"* $\boldsymbol{\theta}$.

In agreement with the generic 3D formulation in Equation (25), the virtual power principle (**VPP**) for the 2D shell model can be formulated as follows:

$$\overline{\mathscr{P}}^{\text{acc}}[\mathfrak{q},\underset{\sim}{\mathfrak{q}}] = \overline{\mathscr{P}}^{\text{ext}}[\mathfrak{q},\underset{\sim}{\mathfrak{q}}] + \overline{\mathscr{P}}^{\text{int}}[\mathfrak{q},\underset{\sim}{\mathfrak{q}}] , \text{ for all k.a. } \underset{\sim}{\mathfrak{q}} \tag{30}$$

and yields, then, by applying the Gateaux *variational derivation* of the former functionals with respect to the *virtual kinematic variable* $\underset{\sim}{q}$, a system of bulk and boundary differential

equations where the actors (D'Alembert–Lagrange's (**DL**) generalized inertial, internal, and external forces) are explicit hereafter:

$$\mathcal{Q}^{\text{acc}}_{\mathfrak{q}/\overline{\mathcal{S}}}[\mathfrak{q}] = \mathcal{Q}^{\text{ext}}_{\mathfrak{q}/\overline{\mathcal{S}}}[\mathfrak{q}] + \mathcal{Q}^{\text{int}}_{\mathfrak{q}/\overline{\mathcal{S}}}[\mathfrak{q}] \, , \text{ for } \mathfrak{q} \in \{\bar{u}_1, \theta_1, \bar{u}_2, \theta_2, \bar{u}_3\} \text{ on } \overline{\mathcal{S}}$$
$$\mathcal{Q}^{\text{acc}}_{\mathfrak{q}/\overline{\mathcal{C}}}[\mathfrak{q}] = \mathcal{Q}^{\text{ext}}_{\mathfrak{q}/\overline{\mathcal{C}}}[\mathfrak{q}] + \mathcal{Q}^{\text{int}}_{\mathfrak{q}/\overline{\mathcal{C}}}[\mathfrak{q}] \, , \text{ for } \mathfrak{q} \in \{\bar{u}_\tau, \theta_\tau, \bar{u}_\nu, \theta_\nu, \bar{u}_3\} \text{ on } \overline{\mathcal{C}} \, . \tag{31}$$

The complete formulation of the boundary initial-value problem (BIVP) for the elastic shells requires adjoining the initial conditions for $\mathfrak{q}$ (i.e., $(\bar{\boldsymbol{u}}, \boldsymbol{\theta})$) and their velocities $\dot{\mathfrak{q}}$ (i.e., $(\dot{\bar{\boldsymbol{u}}}, \dot{\boldsymbol{\theta}})$). Below, we describe explicitly each of the foregoing inertial, internal, and external contributions. For simplicity, in the material description of our elasto-dynamic problems, in order for the *geometrical* mid-surface to coincide with the *material* mid-surface formed by the mass centroids, the shell mass density, as well as all the elastic parameters introduced hereafter were assumed to be (*transversely*) *homogeneous* through the shell thickness.

### 3.2.1. Effective Internal Stress Force and Stress Couple Resultants

The IVP (see [84,85]) due to the stresses within the moderately thick shell:

$$\mathscr{P}^{\text{int}}[\boldsymbol{u}, \underline{\boldsymbol{u}}](t) \stackrel{\text{def}}{=} - \int_{\mathcal{V}} \sigma(\boldsymbol{\alpha}, t) : \varepsilon^s[\underline{\boldsymbol{u}}(\boldsymbol{\alpha})] d\mathcal{V} \, ,$$
$$\text{where } \varepsilon^s[\underline{\boldsymbol{u}}] \stackrel{\text{def}}{=} \frac{1}{2}\Big[\nabla \underline{\boldsymbol{u}} + (\nabla \underline{\boldsymbol{u}})^{\mathrm{T}}\Big] \text{ and } \sigma^{\mathrm{T}} = \sigma \tag{32}$$

relies on the ensuing (Cauchy) classical (non-polar) continuum stress tensor (in absence of bulk concentrated loading couples):

$$\sigma = \sigma_{IJ} \boldsymbol{t}_I \otimes \boldsymbol{t}_J + \sigma_{I3} \boldsymbol{t}_I \otimes \boldsymbol{n} + \sigma_{3I} \boldsymbol{n} \otimes \boldsymbol{t}_I + \sigma_{33} \boldsymbol{n} \otimes \boldsymbol{n} \, . \tag{33}$$

Here, $\sigma_{ii}$ (for $i \in \{1, 2, 3\}$) represent (respectively, tangent and transverse) normal stress components (respectively, for $i \in \{1, 2\}$ and $i = 3$); $\sigma_{IJ} \equiv \sigma_{JI}$ (for $I \neq J$, $I, J \in \{1, 2\}$) are the membrane (or tangent) shear stresses; $\sigma_{3I} \equiv \sigma_{I3}$ are the transverse shear stresses. Taking the symmetries $\sigma_{ij} = \sigma_{ji}$ into account, which again is valid for non-polar media and in absence of bulk concentrated couples, as well as the symmetries of the linear virtual strain fields $\varepsilon^s_{ij}[\underline{\boldsymbol{u}}] \equiv \varepsilon^s_{ji}[\underline{\boldsymbol{u}}]$, the well-known IVP due to the stresses generated within a thin shell can be expressed and provides then the requested IVP expression for the 2D shell modeling:

$$\overline{\mathscr{P}}^{\text{int}}[\mathfrak{q}, \underline{\mathfrak{q}}] \stackrel{\text{def}}{=} - \int_{\overline{\mathcal{S}}}\Big[N_{IJ}\bar{u}_{J||I} + |\epsilon_{JK}|M_{IJK}\underline{\theta}_{K||I} + Q_I\big(\underline{\bar{u}}_{3||I} + \epsilon_{IK}\underline{\theta}_K\big)\Big]d\overline{\mathcal{S}} \, ,$$
$$\text{with} \qquad |\epsilon_{JK}|M_{IJK}\underline{\theta}_{K||I} \equiv |\epsilon_{IJ}|\big(M_{IJI}\underline{\theta}_{I||I} + M_{IIJ}\underline{\theta}_{J||I}\big) \, . \tag{34}$$

Accordingly, with Love's shell modeling expositions and terminology (cf. the generalization in [72], Section 11, pages 512–515), this functional—wherein the variations with respect to $\alpha_3$ are completely eliminated (by integration) to give a 2D theory of thin shells—involves all the possible, cross-sectional, and internal force and moment resultants (per infinitesimal unit of length $|\epsilon_{IL}|d\bar{s}_L = |\epsilon_{IL}|A_L d\alpha_L$ of the neutral surface $\overline{\mathcal{S}}$) for the shell, with, for $I, J \in \{1, 2\}$,

$$N_{IJ} \stackrel{\text{def}}{=} \int_{-h/2}^{h/2} \sigma_{IJ}|\epsilon_{IL}|\Big(1 + \frac{\alpha_3}{R_{n_L}}\Big)d\alpha_3 \approx \int_{-h/2}^{h/2} \sigma_{IJ}d\alpha_3 + \mathcal{O}\Big(\max_{L=1,2}\Big\{\Big|\frac{h}{R_{n_L}}\Big|\Big\}\Big)$$
$$Q_I \stackrel{\text{def}}{=} \int_{-h/2}^{h/2} \sigma_{I3}|\epsilon_{IL}|\Big(1 + \frac{\alpha_3}{R_{n_L}}\Big)d\alpha_3 \approx \int_{-h/2}^{h/2} \sigma_{I3}d\alpha_3 + \mathcal{O}\Big(\max_{L=1,2}\Big\{\Big|\frac{h}{R_{n_L}}\Big|\Big\}\Big) \tag{35}$$
$$\epsilon_{JK}M_{IJK} \stackrel{\text{def}}{=} \int_{-h/2}^{h/2} \sigma_{IJ}\alpha_3|\epsilon_{IL}|\Big(1 + \frac{\alpha_3}{R_{n_L}}\Big)d\alpha_3 \approx \int_{-h/2}^{h/2} \sigma_{IJ}\alpha_3 d\alpha_3 + \mathcal{O}\Big(\max_{L=1,2}\Big\{\Big|\frac{h^2}{R_{n_L}}\Big|\Big\}\Big) .$$

Equation (35) allows introducing the following effective internal force tensor for later use:

$$\boldsymbol{\Sigma} \stackrel{\text{def}}{=} N_{IJ}\boldsymbol{t}_I \otimes \boldsymbol{t}_J + Q_I\boldsymbol{t}_I \otimes \boldsymbol{n} \, . \tag{36}$$

The latter contains no contribution, like $N_{33}\boldsymbol{n} \otimes \boldsymbol{n}$, that would be related to the energetically work-less or negligible normal stress $\sigma_{33}\boldsymbol{n} \otimes \boldsymbol{n}$. The tensor $\boldsymbol{\Sigma}$ includes, however, other contributions providing various stress resultants that act as illustrated in Figure 4. More explicitly, $N_{II}\boldsymbol{t}_I$ and $|\epsilon_{IJ}|N_{IJ}\boldsymbol{t}_J$ for $I = 1,2$ are, respectively, the "*membrane*" (or "*in-plane*") *normal* and *shear* stress resultants (or force densities) per unit length $|\epsilon_{IL}|d\bar{s}_L$ of neutral surface $\overline{\mathcal{S}}$, acting on a surface strip element of area $d\mathcal{S}_I = |\epsilon_{IL}|B_L d\alpha_L d\alpha_3 \equiv |\epsilon_{IL}|\left(1 + \dfrac{\alpha_3}{R_{n_L}}\right)d\bar{s}_L d\alpha_3$ and oriented as $\epsilon_{IL}\boldsymbol{t}_L = \boldsymbol{n} \times \boldsymbol{t}_I$. Besides, $Q_I\boldsymbol{n}$ for $I = 1,2$ in Equation (36) represent the *transverse shear* stress resultants per unit length $|\epsilon_{IL}|d\bar{s}_L$ of $\overline{\mathcal{S}}$, due to possibly nonzero *transverse shear* stresses $\sigma_{I3}$ acting on the prior surface strip element.

Equation (35) provides as well a stress couple (pseudo) tensor made of different contributions as

$$\mathcal{M} \overset{\text{def}}{=} |\epsilon_{JK}|M_{IJK}\boldsymbol{t}_I \otimes \boldsymbol{t}_K \equiv |\epsilon_{IJ}|\big[M_{IJI}\boldsymbol{t}_I \otimes \boldsymbol{t}_I + M_{IIJ}\boldsymbol{t}_I \otimes \boldsymbol{t}_J\big] \tag{37}$$

but without the component on $\boldsymbol{t}_I \otimes \boldsymbol{n}$ (for $I = 1,2$). This is due to the fact that the transverse differential element of the shell along $\boldsymbol{n}$ has negligible side dimensions with respect to $h$ and can sustain then no moment action about $\boldsymbol{n}$. In the "membrane" tensor $\mathcal{M}$, the terms $|\epsilon_{IJ}|M_{IIJ}\boldsymbol{t}_J$ and $|\epsilon_{IJ}|M_{IJI}\boldsymbol{t}_I$ for $I = 1,2$ represent, respectively, (as illustrated in Figure 4) the *bending* and *twisting* stress couples per unit length $|\epsilon_{IL}|d\bar{s}_L$ of $\overline{\mathcal{S}}$, and that results from the stress acting about the neutral surface $\overline{\mathcal{S}}$ on the aforementioned surface strip element at the local material point $\overline{P} \in \overline{\mathcal{S}}$.

**Remark on the geometric shape parameter of intrinsic coupling modes.** The approximations of the foregoing effective stress forces and stress couples at first order in $\alpha_3$ in Equation (35) correspond to the standard NR first-order shell case; they notably imply the following standard relations of symmetry:

$$N_{12} = N_{21} \text{ and } M_{121} = -M_{212}\,, \text{ as } \sigma_{12} = \sigma_{21}\,,$$

which obviously hold only when the second term $\dfrac{\alpha_3}{R_{n_L}}$ in parentheses can be neglected (as for the shallow shell and flat plate cases) or $R_{n1} \equiv R_{n2}$ (as for the spherical shell case) [76]. Nevertheless, these symmetries do not necessarily arise otherwise if higher orders of approximation are considered; instead, we have, in general,

$$N_{12} - N_{21} \equiv \dfrac{M_{121}}{R_{n_1}} + \dfrac{M_{212}}{R_{n_2}} \equiv \varrho \int_{-h/2}^{h/2} \sigma_{12}\dfrac{12\alpha_3}{h}\,d\alpha_3\,, \text{ as } \sigma_{12} = \sigma_{21}\,, \tag{38}$$

which must be viewed as stress field constraints when the shell modeling must be interpreted as resulting from an asymptotic dimensional reduction process. Besides the particular circumstance where $\sigma_{12} = \sigma_{21}$ is symmetric through the shell thickness, these equivalent combinations vanish whenever the following does:

$$\varrho \overset{\text{def}}{=} \dfrac{h}{12R_{n_2}} - \dfrac{h}{12R_{n_1}}\,, \text{ for } \bar{\boldsymbol{\alpha}} \in \Omega\,. \tag{39}$$

This dimensionless function $\varrho(\bar{\boldsymbol{\alpha}})$ measures (in a sense) some local features of "non-flatness" and "non-sphericity" in the principal coordinate system of the mid-surface generic rest shape and of its diagonalized extrinsic curvature tensor $\dfrac{h}{12}\overline{\nabla}\boldsymbol{n}$. Moreover, when $\varrho(\bar{\boldsymbol{\alpha}}) \neq 0$, $\boldsymbol{\Sigma}$ is not generally symmetric, and the trace $M_{121} + M_{212}$ of $\mathcal{M}$ is generally not zero, so that one can say accordingly that these effective tensors given by model reduction (as in homogenization theory) have, in general, no inherent symmetry property. As seen hereafter, nonzero values $\varrho(\bar{\boldsymbol{\alpha}})$ are also associated with the more complicated constitutive relations than those typically known for flat plates.

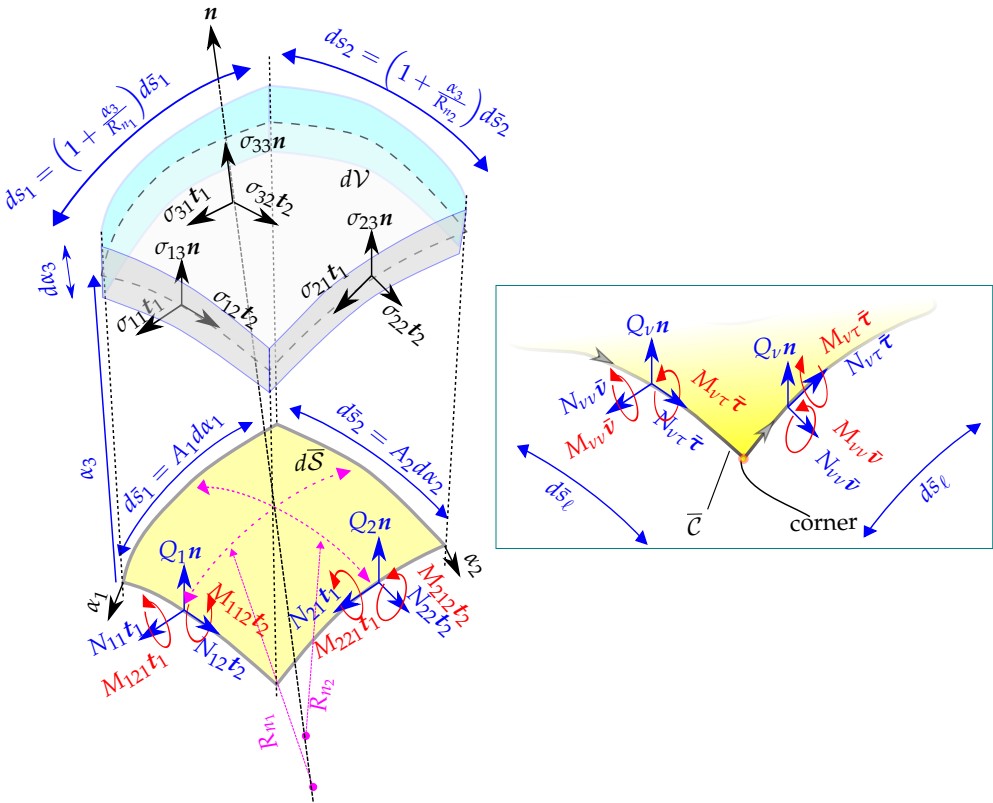

**Figure 4.** Stress traction components on an infinitesimal volume $d\mathcal{V}$, as well as the orientation convention for the effective stress resultant and couple vectors on the related infinitesimal mid-surface element $d\overline{\mathcal{S}}$ and on related infinitesimal contiguous elements $d\overline{s}_\ell$ of $\overline{\mathcal{C}}$ in the medallion.

**D'Alembert–Lagrange's generalized internal forces** $\mathcal{Q}_\mathfrak{q}^{\text{int}}$**.** In order to apply the VPP, the virtual power of the internal forces and couples in Equation (34) must be split into bulk and boundary contributions as in Equations (28) by integration by parts. This then yields

$$\overline{\mathscr{P}}_{/\overline{\mathcal{S}}}^{\text{int}}[\mathfrak{q},\mathfrak{g}] \equiv \int_{\overline{\mathcal{S}}}\left\{\left[\frac{|\epsilon_{IJ}|}{A_J}\frac{\partial(A_J N_{II})}{\partial\overline{s}_I} + \frac{|\epsilon_{IJ}|}{A_I}\frac{\partial(A_I N_{JI})}{\partial\overline{s}_J} + \epsilon_{IJ}\left(\frac{N_{JJ}}{R_{g_J}} + \frac{N_{IJ}}{R_{g_J}}\right) + \frac{Q_I}{R_{n_I}}\right]\underline{\bar{u}}_I\right.$$
$$+ \left[\frac{|\epsilon_{IJ}|}{A_J}\frac{\partial(A_J M_{IJI})}{\partial\overline{s}_I} + \frac{|\epsilon_{IJ}|}{A_I}\frac{\partial(A_I M_{JJI})}{\partial\overline{s}_J} + \epsilon_{IJ}\left(\frac{M_{JIJ}}{R_{g_J}} + \frac{M_{IIJ}}{R_{g_I}} + Q_J\right)\right]\underline{\theta}_I$$
$$\left. + \left[\frac{|\epsilon_{IJ}|}{A_J}\frac{\partial(A_J Q_I)}{\partial\overline{s}_I} - \frac{N_{II}}{R_{n_I}}\right]\underline{\bar{u}}_3\right\}d\overline{\mathcal{S}}$$
$$\overline{\mathscr{P}}_{/\overline{\mathcal{C}}}^{\text{int}}[\mathfrak{q},\mathfrak{g}] \equiv -\oint_{\overline{\mathcal{C}}}[N_{\nu\tau}\underline{\bar{u}}_\tau + M_{\nu\tau}\underline{\theta}_\tau + N_{\nu\nu}\underline{\bar{u}}_\nu + M_{\nu\nu}\underline{\theta}_\nu + Q_\nu\underline{\bar{u}}_3]d\overline{s}_\ell .$$

Accordingly, the D'Alembert–Lagrange generalized internal forces in Equation (29) related to the *virtual kinematic variable* $\mathfrak{q}$ come up then as

$$\begin{aligned}
\mathcal{Q}_{\bar{u}_I}^{\text{int}} &= \frac{|\epsilon_{IJ}|}{A_J}\frac{\partial(A_J N_{II})}{\partial\overline{s}_I} + \frac{|\epsilon_{IJ}|}{A_I}\frac{\partial(A_I N_{JI})}{\partial\overline{s}_J} + \epsilon_{IJ}\left(\frac{N_{JJ}}{R_{g_J}} + \frac{N_{IJ}}{R_{g_I}}\right) + \frac{Q_I}{R_{n_I}}\\
\mathcal{Q}_{\bar{u}_3}^{\text{int}} &= \frac{|\epsilon_{IJ}|}{A_J}\frac{\partial(A_J Q_I)}{\partial\overline{s}_I} - \frac{N_{II}}{R_{n_I}}\\
\mathcal{Q}_{\theta_I}^{\text{int}} &= \frac{|\epsilon_{IJ}|}{A_J}\frac{\partial(A_J M_{IJI})}{\partial\overline{s}_I} + \frac{|\epsilon_{IJ}|}{A_I}\frac{\partial(A_I M_{JJI})}{\partial\overline{s}_J} + \epsilon_{IJ}\left(\frac{M_{JIJ}}{R_{g_J}} + \frac{M_{IIJ}}{R_{g_I}} + Q_J\right)\\
\mathcal{Q}_{\bar{u}_\tau}^{\text{int}} &= -N_{\nu\tau} , \quad \mathcal{Q}_{\bar{u}_\nu}^{\text{int}} = -N_{\nu\nu} , \quad \mathcal{Q}_{\bar{u}_3}^{\text{int}} = -Q_\nu , \quad \mathcal{Q}_{\theta_\tau}^{\text{int}} = -M_{\nu\tau} , \quad \mathcal{Q}_{\theta_\nu}^{\text{int}} = -M_{\nu\nu} .
\end{aligned} \tag{40}$$

These involve on the boundary $\overline{\mathcal{C}}$ the components of the resultant stress force and couple tensors $\boldsymbol{\Sigma}$ and $\mathcal{M}$:

$$N_{\nu\tau} \overset{\text{def}}{=} N_{IJ}\bar{\nu}_I\bar{\tau}_J \equiv \bar{\boldsymbol{\nu}}\cdot\boldsymbol{\Sigma}\cdot\bar{\boldsymbol{\tau}} , \ N_{\nu\nu} \overset{\text{def}}{=} N_{IJ}\bar{\nu}_I\bar{\nu}_J \equiv \bar{\boldsymbol{\nu}}\cdot\boldsymbol{\Sigma}\cdot\bar{\boldsymbol{\nu}} \text{ and } Q_\nu \overset{\text{def}}{=} Q_I\bar{\nu}_I \equiv \bar{\boldsymbol{\nu}}\cdot\boldsymbol{\Sigma}\cdot\boldsymbol{n}$$
$$M_{\nu\tau} \overset{\text{def}}{=} |\epsilon_{JK}|M_{IJK}\bar{\nu}_I\bar{\tau}_K \equiv \bar{\boldsymbol{\nu}}\cdot\mathcal{M}\cdot\bar{\boldsymbol{\tau}} \text{ and } M_{\nu\nu} \overset{\text{def}}{=} |\epsilon_{JK}|M_{IJK}\bar{\nu}_I\bar{\nu}_K \equiv \bar{\boldsymbol{\nu}}\cdot\mathcal{M}\cdot\bar{\boldsymbol{\nu}} . \tag{41}$$

Similar to the previous ones on $\overline{\mathcal{S}}$, these correspond to the effective (work) mechanical internal reactions per unit reference length $d\bar{s}_\ell$ at the boundary $\overline{\mathcal{C}}$ with, in the edge tangent, co-normal and normal directions $\bar{\boldsymbol{\tau}}$, $\boldsymbol{n}$, and $\bar{\boldsymbol{\nu}}$: the in-plane shear stress resultant $N_{\nu\tau}\bar{\boldsymbol{\tau}}$; the in-plane normal stress resultant $N_{\nu\nu}\bar{\boldsymbol{\nu}}$; the transverse stress resultant $Q_\nu \boldsymbol{n}$; the *bending* stress couple $M_{\nu\tau}\bar{\boldsymbol{\tau}}$; the *twisting* stress couple $M_{\nu\nu}\bar{\boldsymbol{\nu}}$ (while, of course, there is no drilling moment as $M_{\nu 3}\boldsymbol{n} = \boldsymbol{0}$). Those effective efforts are linked in principle, in a less trivial way not described here, to the resultants and moments of the edge stresses acting on an surface strip element of area $d\mathcal{S}_\ell = B_\ell d\bar{s}_\ell d\alpha_3$ of the lateral surface $\mathcal{S}_\ell$:

$$\sigma_{\nu\nu} \stackrel{\text{def}}{=} \boldsymbol{\nu} \cdot \boldsymbol{\sigma} \cdot \boldsymbol{\nu} \quad , \quad \sigma_{\nu\tau} \equiv \sigma_{\tau\nu} \stackrel{\text{def}}{=} \boldsymbol{\nu} \cdot \boldsymbol{\sigma} \cdot \boldsymbol{\tau} \quad \text{and} \quad \sigma_{\nu 3} \stackrel{\text{def}}{=} \boldsymbol{\nu} \cdot \boldsymbol{\sigma} \cdot \boldsymbol{n} \quad , \quad \text{for } \boldsymbol{r}(\boldsymbol{\alpha}) \in \mathcal{S}_\ell \quad (42)$$

while performing the calculus with the differential length $B_\ell d\bar{s}_\ell$ (defined with the *total geodetic curvature* $B_\ell \stackrel{\text{def}}{=} \sqrt{B_\tau^2 + B_\nu^2}$ of the curve $\mathcal{C}$ locally located at a signed distance $\alpha_3 \boldsymbol{n}$ from $\overline{\mathcal{C}}$) in place of $|\epsilon_{IL}| \left(1 + \dfrac{\alpha_3}{R_{n_L}}\right) d\bar{s}_L$.

Note that the previous formulas apply to shells made of any "*simple*", but not necessarily elastic, material. The class of simple materials contains all elastic materials, the linearly viscous fluid, and many of the other materials commonly studied in the applications. Moreover, those equations reduce to the Mindlin–Reissner moderately thick plate theory when the curvatures $(\min_{I=1,2}\{R_{n_I}\} = \infty)$ are set to zero.

**Matrix formulation of the generalized internal force relations of the NR shell model.** For our PHS variational purposes, the bulk Equation (40) over $\overline{\mathcal{S}}$ can be rewritten in matrix form as

$$\mathcal{Q}^{\text{int}}_{\mathfrak{q}/\overline{\mathcal{S}}}[\mathfrak{q}] = \overline{\mathbb{D}}\, \overline{\mathbf{F}}^{\text{int}}_{/\overline{\mathcal{S}}}[\mathfrak{q}] \ , \tag{43a}$$

with the following $(5 \times 10)$ local differential operator matrix (with possibly non-constant coefficients):

$$\overline{\mathbb{D}} \stackrel{\text{def}}{=} \begin{pmatrix} \frac{1}{A_2}\frac{\partial}{\partial \bar{s}_1} & \frac{1}{A_1 R_{g_2}} & \frac{1}{A_1}\frac{\partial}{\partial \bar{s}_2} & \frac{1}{A_2 R_{g_1}} & \frac{1}{A_2 R_{n_1}} & 0 & 0 & 0 & 0 & 0 \\ 0 & 0 & 0 & 0 & 0 & \frac{1}{A_1} & \frac{1}{A_2}\frac{\partial}{\partial \bar{s}_1} & \frac{1}{A_1 R_{g_2}} & \frac{1}{A_1}\frac{\partial}{\partial \bar{s}_2} & \frac{1}{A_2 R_{g_1}} \\ \frac{-1}{A_2 R_{g_1}} & \frac{1}{A_1}\frac{\partial}{\partial \bar{s}_2} & \frac{-1}{A_1 R_{g_2}} & \frac{1}{A_2}\frac{\partial}{\partial \bar{s}_1} & 0 & \frac{1}{A_1 R_{n_2}} & 0 & 0 & 0 & 0 \\ 0 & 0 & 0 & 0 & \frac{-1}{A_2} & 0 & \frac{-1}{A_2 R_{g_1}} & \frac{1}{A_1}\frac{\partial}{\partial \bar{s}_2} & \frac{-1}{A_1 R_{g_2}} & \frac{1}{A_2}\frac{\partial}{\partial \bar{s}_1} \\ \frac{-1}{A_2 R_{n_1}} & \frac{-1}{A_1 R_{n_2}} & 0 & 0 & \frac{1}{A_2}\frac{\partial}{\partial \bar{s}_1} & \frac{1}{A_1}\frac{\partial}{\partial \bar{s}_2} & 0 & 0 & 0 & 0 \end{pmatrix} \tag{43b}$$

with column headers $N_{11}A_2$, $N_{22}A_1$, $N_{21}A_1$, $N_{12}A_2$, $Q_1 A_2$, $Q_2 A_1$, $M_{121}A_2$, $M_{212}A_1$, $M_{221}A_1$, $M_{112}A_2$ and row labels $\mathcal{Q}^{\text{int}}_{\bar{u}_1}$, $\mathcal{Q}^{\text{int}}_{\theta_1}$, $\mathcal{Q}^{\text{int}}_{\bar{u}_2}$, $\mathcal{Q}^{\text{int}}_{\theta_2}$, $\mathcal{Q}^{\text{int}}_{\bar{u}_3}$

linking the column matrices of internal forces and couples to their resultants on a surface portion of area $d\overline{\mathcal{S}}$:

$$\overline{\mathbf{F}}^{\text{int}}_{/\overline{\mathcal{S}}} \stackrel{\text{def}}{=} \left\{ \begin{array}{c} A_2 N_{11} \\ A_1 N_{22} \\ A_1 N_{21} \\ A_2 N_{12} \\ A_2 Q_1 \\ A_1 Q_2 \\ A_2 M_{121} \\ A_1 M_{212} \\ A_1 M_{221} \\ A_2 M_{112} \end{array} \right\} \text{ and (as a reminder) } \mathcal{Q}^{\text{int}}_{\mathfrak{q}/\overline{\mathcal{S}}} \stackrel{\text{def}}{=} \left\{ \begin{array}{c} \mathcal{Q}^{\text{int}}_{\bar{u}_1} \\ \mathcal{Q}^{\text{int}}_{\theta_1} \\ \mathcal{Q}^{\text{int}}_{\bar{u}_2} \\ \mathcal{Q}^{\text{int}}_{\theta_2} \\ \mathcal{Q}^{\text{int}}_{\bar{u}_3} \end{array} \right\}. \tag{43c}$$

Now, it is also natural (but it may not always be) to additionally introduce the energy dual-column matrix for $\mathfrak{q}_{/\overline{\mathcal{C}}}$ in Equation (24) on any curvilinear portion of contour length $d\overline{s}_\ell$:

$$\overline{\mathbf{F}}_{/\overline{\mathcal{C}}}^{\text{int}} \stackrel{\text{def}}{=} \left\{ \begin{array}{c} N_{\nu\tau} \\ M_{\nu\tau} \\ N_{\nu\nu} \\ M_{\nu\nu} \\ Q_\nu \end{array} \right\} \equiv -\mathcal{Q}_{\mathfrak{q}_{/\overline{\mathcal{C}}}}^{\text{int}} \stackrel{\text{def}}{=} -\left\{ \begin{array}{c} \mathcal{Q}_{\overline{u}_\tau}^{\text{int}} \\ \mathcal{Q}_{\theta_\tau}^{\text{int}} \\ \mathcal{Q}_{\overline{u}_\nu}^{\text{int}} \\ \mathcal{Q}_{\theta_\nu}^{\text{int}} \\ \mathcal{Q}_{\overline{u}_3}^{\text{int}} \end{array} \right\}. \tag{44}$$

**Approximate effective stress–strain constitutive laws.** Regarding the previous (generalized) internal forces and moments in Equation (35), the final non-trivial task would merely be to compute them if the stress tensor distribution $\sigma$ was already sthenically known through the shell thickness. As usual, this is performed by default by using some constitutive equations, which, definitive for the shell 2D-formulation, relate the prior stress resultants and stress couples (which have a more complex form than in the plate theory), instead of merely stresses, to the corresponding strains and curvatures (and therefore, express the effective internal forces and moments in terms of the displacements $(\overline{u}, \theta)$). In this work, we were interested only in isotropic linear elastic shells (i.e., whose material properties have, therefore, no directional preference in the tangent plane of the middle surface) in small displacements and transversely homogeneous through the shell thickness. In principle, Hooke's generalized law must apply for a 3D linear isotropic elasticity body with Coulomb–Lamé's shear modulus $G(\overline{\alpha}) \stackrel{\text{def}}{=} \dfrac{E(\overline{\alpha})}{2(1+\nu(\overline{\alpha}))} > 0$ being related to Young's modulus $E(\overline{\alpha}) > 0$ and Poisson's ratio $\nu(\overline{\alpha}) > 1/2$, in the local principal basis $(e_1, e_2, e_3) \equiv (t_1, t_2, n)$. Nevertheless, in view of the shell thinness and owing to the linear framework, we adopted, as most classical shell models [74] (see also, e.g., the discussions in [76] (page 19)), the ensuing (seemingly ad hoc, but somehow common) approximate stress–strain constitutive relation:

$$\sigma_{II} = 2G\left(\varepsilon_{II}^s + \tfrac{\nu}{1-\nu}|\epsilon_{IJ}|\varepsilon_{JJ}^s\right), \text{ for } I = 1,2 \qquad \text{(plane stress)}$$

$$\sigma_{12} = G(\varepsilon_{12}^s + \varepsilon_{21}^s) \equiv 2G\varepsilon_{12}^s \equiv \sigma_{21} \qquad \text{(general case)} \tag{45}$$

$$\sigma_{I3} = \zeta G(\varepsilon_{I3}^s + \varepsilon_{3I}^s) \equiv 2\zeta G\varepsilon_{I3}^s \equiv \sigma_{3I}, \text{ for } I = 1,2 \qquad \text{(general case)}$$

while letting $(G, E, \nu)$ take possibly values different from the 3D elasticity ones. Here, a "*quasi-plane stress state*" hypothesis is enforced only for the "in-plane" (tangent) normal stress components $\sigma_{II}$ (in anticipation of static membrane problems), but not necessary for the surface "in-plane" (tangent) and "out-plane" (transverse) shear stress components $\sigma_{12} = \sigma_{21}$ and $\sigma_{3I} = \sigma_{I3}$, nor the *indeterminable* "out-plane" normal stress one $\sigma_{33}$. Note as well that these expressions assume $\varepsilon_{33}^s = 0$ (i.e., the shell is transversely undeformable), neglecting, therefore, Poisson's effects in the transverse direction. Moreover, in order to treat the moderately thick shell case, a classical "*transverse shear correction factor*" (or "*shear reduced coefficient*") $\zeta(\overline{\alpha}) > 0$ (see the third line in Equation (45)) is introduced in the transverse shear stress–strain constitutive relations to make the theory (energetically or first vibrational mode) compatible with the complete 3D formulation, like for plates [86].

**Geometric in-plane extension–bending and in-plane twisting–shearing interaction effects.** The integration over the shell thickness of the local constitutive equations for the 3D thin shell stress fields in Equation (45) provides then effective constitutive ones for the 2D thin shell stress resultants and couples in Equation (35) (for $I = 1,2$) as

$$N_{II} = 2Gh|\epsilon_{IJ}|\left[a_{IJ}\overline{u}_{I||I} + \frac{\nu}{1-\nu}\overline{u}_{J||J} + \frac{hb_{IJ}}{4}\beta_{I||I}\right] \quad \text{(in-plane normal stress resultants)} \tag{47a}$$

$$|\epsilon_{IJ}|N_{IJ} = Gh|\epsilon_{IJ}|\left[\overline{u}_{I||J} + a_{IJ}\overline{u}_{J||I} + \frac{hb_{IJ}}{4}\beta_{J||I}\right] \quad \text{(in-plane shear stress resultants)} \tag{47b}$$

$$Q_I \quad = \quad \zeta G h |\epsilon_{IJ}| a_{IJ} \left[ \bar{u}_{3||I} + \beta_I \right] \quad \text{(transverse shear stress resultants)} \tag{47c}$$

$$|\epsilon_{IJ}| M_{IIJ} \quad = \quad \frac{G h^2}{6} \epsilon_{IJ} \left[ 3 b_{IJ} \bar{u}_{I||I} + h c_{IJ} \beta_{I||I} + \frac{h\nu}{1-\nu} \beta_{J||J} \right] \quad \text{(bending stress moments)} \tag{47d}$$

$$|\epsilon_{IJ}| M_{IJI} \quad = \quad \frac{G h^2}{12} \epsilon_{JI} \left[ 3 b_{IJ} \bar{u}_{J||I} + h \beta_{I||J} + h c_{IJ} \beta_{J||I} \right] \quad \text{(twisting stress moments)} . \tag{47e}$$

These expressions rely on the following (zeroth, first, and second) statistical moments:

$$
\begin{aligned}
a_{IJ}(\varrho, \bar{\boldsymbol{\alpha}}) \quad &\overset{\text{def}}{=} \quad \int_{-h/2}^{h/2} \frac{A_I B_J}{A_J B_I} \frac{d\alpha_3}{h} \equiv \int_{-h/2}^{h/2} \frac{\frac{\alpha_3}{R_{n_J}} + 1}{\frac{\alpha_3}{R_{n_I}} + 1} \frac{d\alpha_3}{h} \\
&\equiv \quad 1 - \frac{R_{n_I}^2}{h^2} \left( \frac{h}{R_{n_I}} - \frac{h}{R_{n_J}} \right) \left[ \frac{h}{R_{n_I}} - \ln\left( \frac{1 + \frac{h}{2R_{n_I}}}{1 - \frac{h}{2R_{n_I}}} \right) \right] \\
&\approx \quad 1 - \frac{h \epsilon_{IJ} \varrho}{R_{n_I}} + \mathcal{O}\left( \max_{L=1,2} \left\{ \frac{|\varrho| h^2}{R_{n_L}^2} \right\} \right)
\end{aligned}
\tag{48a}
$$

$$
\begin{aligned}
b_{IJ}(\varrho, \bar{\boldsymbol{\alpha}}) \quad &\overset{\text{def}}{=} \quad 4 \int_{-h/2}^{h/2} \frac{A_I B_J}{B_I A_J} \frac{\alpha_3 d\alpha_3}{h^2} \equiv 4 \frac{R_{n_I}^3}{h^3} \left( \frac{h}{R_{n_I}} - \frac{h}{R_{n_J}} \right) \left[ \frac{h}{R_{n_I}} - \ln\left( \frac{1 + \frac{h}{2R_{n_I}}}{1 - \frac{h}{2R_{n_I}}} \right) \right] \\
&\approx \quad 4 \epsilon_{IJ} \varrho + \mathcal{O}\left( \max_{L=1,2} \left\{ \frac{|\varrho| h^2}{R_{n_L}^2} \right\} \right)
\end{aligned}
\tag{48b}
$$

$$
\begin{aligned}
c_{IJ}(\varrho, \bar{\boldsymbol{\alpha}}) \quad &\overset{\text{def}}{=} \quad 12 \int_{-h/2}^{h/2} \frac{A_I B_J}{B_I A_J} \frac{(\alpha_3)^2 d\alpha_3}{h^3} \\
&\equiv \quad 1 - \frac{R_{n_I}}{h} \left( \frac{h}{R_{n_I}} - \frac{h}{R_{n_J}} \right) \left\{ 1 + 12 \frac{R_{n_I}^3}{h^3} \left[ \frac{h}{R_{n_I}} - \ln\left( \frac{1 + \frac{h}{2R_{n_I}}}{1 - \frac{h}{2R_{n_I}}} \right) \right] \right\} \\
&\approx \quad 1 - \frac{9 h \epsilon_{IJ} \varrho}{5 R_{n_I}} + \mathcal{O}\left( \max_{L=1,2} \left\{ \frac{|\varrho| h^2}{R_{n_L}^2} \right\} \right) .
\end{aligned}
\tag{48c}
$$

As observed now in Equation (47), these functions of $\bar{\boldsymbol{\alpha}}$ and $\varrho(\bar{\boldsymbol{\alpha}})$ introduced in Equation (39) produce various coupling effects between all the stress resultants and stress couples due as well to the possible local "non-flatness" and "non-sphericity" of the shell principal coordinate system on its generic rest shape $\overline{\mathcal{S}}$. Their approximations in Equation (48) constitute in a sense Byrne–Flügge-Lur'ye's approximations when used notably with the Love–Kirchhoff–Koiter kinematic constraints (see [76,80]; [73], Section 2.10, pages 43–46). The account of at least such a second approximation order in $\max\limits_{J=1,2} \left\{ \left| \frac{h}{R_{n_J}} \right| \right\} \ll$ 1 (i.e., all the terms linear in $\left\{ \frac{\alpha_3}{R_{n_I}}, \left( \frac{\alpha_3}{R_{n_I}} \right)^2, \frac{|\epsilon_{IJ}| \alpha_3^2}{R_{n_J} R_{n_I}} \right\}_{I=1,2}$ ) usually allows not restricting the analysis to *thin, shallow shells* and explaining then the appearance of *bending moments under extension* and, reciprocally, *extension forces under bending*—as well as the in-plane analogous *twisting–shearing* effects induced (without adding new kinematical variables) by geometrical corrections. One can mention as well that such a coupling between the membrane force resultants and bending strains or between the bending moment resultants and membrane strains occurs if the reference surface is not halfway between the inner and outer surfaces of the shell; see e.g., [73] (Chapter 2.11, page 47).

**Effective potential energy of the elastic deformation of a strained shell.** Now, as is well known for elastic materials, the IVP represents also some expression of the first variation of the potential (or stored) energy of elastic deformation (or strain energy). In particular, the shell IVP in Equation (34) can be rewritten indeed as

$$
\begin{aligned}
\overline{\mathscr{P}}^{\mathrm{int}}[\mathfrak{q},\mathfrak{q}] \quad\equiv\quad & -\int_{\overline{\mathcal{S}}} Gh|\epsilon_{IJ}|\left\{2\left[a_{IJ}\bar{u}_{I||I}+\frac{\nu}{1-\nu}\bar{u}_{J||J}+\frac{hb_{IJ}}{4}\beta_{I||I}\right]\underline{\bar{u}}_{I||I}\right.\\
& +\left[\bar{u}_{I||J}+a_{IJ}\bar{u}_{J||I}+\frac{hb_{IJ}}{4}\beta_{J||I}\right]\underline{\bar{u}}_{J||I}\\
& +\zeta a_{IJ}\left[\bar{u}_{3||I}+\beta_I\right]\left(\underline{\bar{u}}_{3||I}+\underline{\beta}_I\right)\\
& +\frac{h}{6}\left[3b_{IJ}\bar{u}_{I||I}+hc_{IJ}\beta_{I||I}+\frac{h\nu}{1-\nu}\beta_{J||J}\right]\underline{\beta}_{I||I}\\
& \left.+\frac{h}{12}\left[3b_{IJ}\bar{u}_{J||I}+h\beta_{I||J}+hc_{IJ}\beta_{J||I}\right]\underline{\beta}_{J||I}\right\}d\overline{\mathcal{S}}
\end{aligned}
\tag{49}
$$

since $|\epsilon_{JK}|M_{IJK}\underline{\theta}_{K||I}\equiv|\epsilon_{IJ}|\left(M_{IIJ}\underline{\theta}_{J||I}+M_{IJI}\underline{\theta}_{I||I}\right)\equiv\epsilon_{IJ}M_{IIJ}\underline{\beta}_{I||I}+\epsilon_{JI}M_{IJI}\underline{\beta}_{J||I}$, while re-minding that $\underline{\theta}_K\equiv\epsilon_{LK}\underline{\beta}_L\Leftrightarrow\underline{\beta}_K\equiv\epsilon_{KL}\underline{\theta}_L$ for $K=1,2$. Subsequently, we inferred that the stress forces and stress couples in Equation (47) of the internal virtual power $\overline{\mathscr{P}}^{\mathrm{int}}[\mathfrak{q},\mathfrak{q}]$ in Equation (34) for the 2D shell also "derive" from an effective elastic potential energy of deformation as

$$
\begin{aligned}
\overline{\mathscr{E}}^{\mathrm{elas}}[\mathfrak{q},\overline{\mathbf{E}}] \quad\overset{\mathrm{def}}{=}\quad & \int_{\overline{\mathcal{S}}}\frac{Gh}{2}|\epsilon_{IJ}|\left\{a_{IJ}\left[2\bar{u}_{I||I}^2+\bar{u}_{J||I}^2+\zeta\left(\bar{u}_{3||I}+\beta_I\right)^2\right]\right.\\
& +\frac{b_{IJ}}{2}h\left(2\beta_{I||I}\bar{u}_{I||I}+\beta_{J||I}\bar{u}_{J||I}\right)\\
& +\bar{u}_{I||J}\bar{u}_{J||I}+\frac{2\nu}{1-\nu}\left(\bar{u}_{J||J}\bar{u}_{I||I}+\frac{h^2}{12}\beta_{J||J}\beta_{I||I}\right)\\
& \left.+\frac{h^2}{12}\beta_{I||J}\beta_{J||I}+\frac{c_{IJ}h^2}{12}\left(2\beta_{I||I}^2+\beta_{J||I}^2\right)\right\}d\overline{\mathcal{S}}\,.
\end{aligned}
\tag{50}
$$

Equivalently, the following matrix-based forms (inspired by Voigt's) will be convenient for our purposes:

$$
\begin{aligned}
-\overline{\mathscr{P}}^{\mathrm{int}}[\mathfrak{q},\mathfrak{q}] \quad\equiv\quad & \int_{\overline{\mathcal{S}}}\underline{\overline{\mathbf{E}}}^{\mathrm{T}}\overline{\mathbf{F}}^{\mathrm{int}}\,d\overline{\mathcal{S}}\equiv-\int_{\overline{\mathcal{S}}}\mathfrak{q}^{\mathrm{T}}\overline{\mathbb{D}}\,\overline{\mathbf{F}}^{\mathrm{int}}\,d\overline{\mathcal{S}}+\oint_{\overline{\mathcal{C}}}\mathfrak{q}^{\mathrm{T}}\overline{\mathbf{F}}^{\mathrm{int}}\,d\bar{s}_\ell\,,\\
& \text{with }\underline{\overline{\mathbf{E}}}\overset{\mathrm{def}}{=}\overline{\mathbf{E}}[\underline{\mathfrak{q}}]\equiv\overline{\mathbb{B}}\,\underline{\mathfrak{q}}\,;\\
\overline{\mathscr{E}}^{\mathrm{elas}}[\mathfrak{q},\overline{\mathbf{E}}] \quad=\quad & \frac{1}{2}\int_{\overline{\mathcal{S}}}\overline{\mathbf{E}}^{\mathrm{T}}\overline{\mathbb{C}}\,\overline{\mathbf{E}}\,d\overline{\mathcal{S}}\equiv\frac{1}{2}\int_{\overline{\mathcal{S}}}\overline{\mathbf{E}}^{\mathrm{T}}\overline{\mathbf{F}}^{\mathrm{int}}\,d\overline{\mathcal{S}}\,.
\end{aligned}
\tag{51}
$$

The expression for $\overline{\mathscr{E}}^{\mathrm{elas}}[\mathfrak{q},\overline{\mathbf{E}}]$ notably involves the ensuing $(10\times10)$, symmetric and positive definite, sparse (due to the material isotropy and transverse homogeneity) stiffness matrix:

$$
\overline{\mathbb{C}}\overset{\mathrm{def}}{=}GhA_1A_2
\begin{pmatrix}
\frac{2a_{12}A_2}{A_1} & \frac{2\nu}{1-\nu} & 0 & 0 & 0 & 0 & 0 & 0 & 0 & \frac{hb_{12}A_2}{2A_1}\\
\frac{2\nu}{1-\nu} & \frac{2a_{21}A_1}{A_2} & 0 & 0 & 0 & 0 & 0 & 0 & \frac{-hb_{21}A_1}{2A_2} & 0\\
0 & 0 & \frac{a_{21}A_1}{A_2} & 1 & 0 & 0 & 0 & \frac{hb_{21}A_1}{4A_2} & 0 & 0\\
0 & 0 & 1 & \frac{a_{12}A_2}{A_1} & 0 & 0 & \frac{-hb_{12}A_2}{4A_1} & 0 & 0 & 0\\
0 & 0 & 0 & 0 & \frac{\zeta a_{12}A_2}{A_1} & 0 & 0 & 0 & 0 & 0\\
0 & 0 & 0 & 0 & 0 & \frac{\zeta a_{21}A_1}{A_2} & 0 & 0 & 0 & 0\\
0 & 0 & 0 & \frac{-hb_{12}A_2}{4A_1} & 0 & 0 & \frac{h^2c_{21}A_1}{12A_2} & \frac{-h^2}{12} & 0 & 0\\
0 & 0 & \frac{hb_{21}A_1}{4A_2} & 0 & 0 & 0 & \frac{-h^2}{12} & \frac{h^2c_{21}A_1}{12A_2} & 0 & 0\\
0 & \frac{-hb_{21}A_1}{2A_2} & 0 & 0 & 0 & 0 & 0 & 0 & \frac{h^2c_{21}A_1}{6A_2} & \frac{-h^2\nu}{6(1-\nu)}\\
\frac{hb_{12}A_2}{2A_1} & 0 & 0 & 0 & 0 & 0 & 0 & 0 & \frac{-h^2\nu}{6(1-\nu)} & \frac{h^2c_{12}A_2}{6A_1}
\end{pmatrix}.
\tag{52}
$$

Then, one can notably link as follows the generalized stress force and couple matrix introduced in Equation (43):

$$\overline{\mathbf{F}}^{\text{int}}_{/\mathcal{S}} \equiv \overline{\mathbb{C}}\,\overline{\mathbf{E}} \equiv \delta_{\overline{\mathbf{E}}}\overline{\mathscr{E}}^{\text{elas}}[\mathfrak{q}, \overline{\mathbf{E}}] \equiv \begin{cases} \delta_{\frac{\tilde{u}_{1\|1}}{A_2}}\overline{\mathscr{E}}^{\text{elas}} = N_{11}A_2 \\[4pt] \delta_{\frac{\tilde{u}_{2\|2}}{A_1}}\overline{\mathscr{E}}^{\text{elas}} = N_{22}A_1 \\[4pt] \delta_{\frac{\tilde{u}_{1\|2}}{A_1}}\overline{\mathscr{E}}^{\text{elas}} = N_{21}A_1 \\[4pt] \delta_{\frac{\tilde{u}_{2\|1}}{A_2}}\overline{\mathscr{E}}^{\text{elas}} = N_{12}A_2 \\[4pt] \delta_{\frac{\tilde{\gamma}_1}{A_2}}\overline{\mathscr{E}}^{\text{elas}} = Q_1A_2 \\[4pt] \delta_{\frac{\tilde{\gamma}_2}{A_1}}\overline{\mathscr{E}}^{\text{elas}} = Q_2A_1 \\[4pt] \delta_{\frac{\theta_{1\|1}}{A_2}}\overline{\mathscr{E}}^{\text{elas}} = M_{121}A_2 \\[4pt] \delta_{\frac{\theta_{2\|2}}{A_1}}\overline{\mathscr{E}}^{\text{elas}} = M_{212}A_1 \\[4pt] \delta_{\frac{\theta_{1\|2}}{A_1}}\overline{\mathscr{E}}^{\text{elas}} = M_{221}A_1 \\[4pt] \delta_{\frac{\theta_{2\|1}}{A_2}}\overline{\mathscr{E}}^{\text{elas}} = M_{112}A_2 \end{cases} \tag{53}$$

to the generalized deformation gradient column matrix $\overline{\mathbf{E}}$ introduced in Equation (23).

Importantly, it is also meaningful to introduce a *complementary* energy for the "*strain & curvature-change*" one $\overline{\mathscr{E}}^{\text{elas}}[\mathfrak{q}, \overline{\mathbf{E}}]$ in Equation (51), for which the "*stress resultant & stress couple*" *potential energy*

$$\overline{\mathscr{E}}^{\text{elas}}_c[\overline{\mathbf{F}}^{\text{int}}] \stackrel{\text{def}}{=} \frac{1}{2}\int_{\mathcal{S}}\left(\overline{\mathbf{F}}^{\text{int}}\right)^{\text{T}}\overline{\mathbb{C}}^{-1}\overline{\mathbf{F}}^{\text{int}}\,d\overline{\mathcal{S}} \equiv \overline{\mathscr{E}}^{\text{elas}}[\mathfrak{q}, \overline{\mathbf{E}}]\,, \tag{54}$$

which comes from inverting the relation in Equation (53) as follows:

$$\overline{\mathbf{E}} = \overline{\mathbb{C}}^{-1}\overline{\mathbf{F}}^{\text{int}}_{/\mathcal{S}} \equiv \delta_{\overline{\mathbf{F}}^{\text{int}}}\overline{\mathscr{E}}^{\text{elas}}_c[\overline{\mathbf{F}}^{\text{int}}] \text{ over } \overline{\mathcal{S}}\,. \tag{55}$$

$\overline{\mathscr{E}}^{\text{elas}}_c[\overline{\mathbf{F}}^{\text{int}}]$ is essential in the (analytical and numerical) variational analysis of elastic structures. Theoretically, it can also formally be obtained by Legendre–Fenchel's (or Young–Fenchel's) transformation:

$$\overline{\mathscr{E}}^{\text{elas}}_c[\overline{\mathbf{F}}^{\text{int}}] = \sup_{\text{k.a.}\,\overline{\mathbf{E}}}\left\{\int_{\mathcal{S}}\overline{\mathbf{E}}^{\text{T}}\overline{\mathbf{F}}^{\text{int}}\,d\overline{\mathcal{S}} - \overline{\mathscr{E}}^{\text{elas}}[\mathfrak{q}, \overline{\mathbf{E}}]\right\} \tag{56}$$

for which Equation (55) is, therefore, a necessary condition of local maximization (and a sufficient condition of the local invertibility of the mapping from $\overline{\mathbf{E}}$ to $\overline{\mathbf{F}}^{\text{int}}_{/\mathcal{S}}[\overline{\mathbf{E}}] \equiv \delta_{\overline{\mathbf{E}}}\overline{\mathscr{E}}^{\text{elas}}$ is effectively that the variation $\dfrac{\partial\overline{\mathbf{F}}^{\text{int}}_{/\mathcal{S}}}{\partial\overline{\mathbf{E}}^{\text{T}}} \equiv \overline{\mathbb{C}}$ is also invertible). Conversely, $\overline{\mathscr{E}}^{\text{elas}}_c[\overline{\mathbf{F}}^{\text{int}}] = \sup_{\text{k.a.}\,\mathfrak{q}}\overline{\mathscr{E}}^{\text{elas}}_{cc}[\mathfrak{q}, \overline{\mathbf{F}}^{\text{int}}]$ and Equation (55) can also be obtained from extremizing the following type of Hellinger–Prange–Reissner (HPR) functional (see, for instance, ([87], page 124, [88,89]):

$$\begin{aligned} \overline{\mathscr{E}}^{\text{elas}}_{cc}[\mathfrak{q}, \overline{\mathbf{F}}^{\text{int}}] &\stackrel{\text{def}}{=} \oint_{\mathcal{C}}\mathfrak{q}^{\text{T}}\overline{\mathbf{F}}^{\text{int}}\,d\bar{s}_\ell - \int_{\mathcal{S}}\mathfrak{q}^{\text{T}}\overline{\mathbb{D}}\,\overline{\mathbf{F}}^{\text{int}}\,d\overline{\mathcal{S}} - \overline{\mathscr{E}}^{\text{elas}}_c[\overline{\mathbf{F}}^{\text{int}}] \\ &\equiv \int_{\mathcal{S}}\overline{\mathbf{E}}^{\text{T}}\overline{\mathbf{F}}^{\text{int}}\,d\overline{\mathcal{S}} - \overline{\mathscr{E}}^{\text{elas}}_c[\overline{\mathbf{F}}^{\text{int}}]\,. \end{aligned} \tag{57}$$

Besides, when the real-time motion is taken for the virtual kinematic variations (i.e., when $\mathfrak{q} \equiv \dot{\mathfrak{q}}$), then Equations (51) and (54) provide, moreover, the (real) internal power due to the internal efforts of cohesion:

$$-\overline{\mathscr{P}}^{\text{int}}[\mathfrak{q}, \dot{\mathfrak{q}}] \equiv \dot{\overline{\mathscr{E}}}^{\text{elas}}[\mathfrak{q}, \overline{\mathbf{E}}] \quad (\equiv \dot{\overline{\mathscr{E}}}^{\text{elas}}_c[\overline{\mathbf{F}}^{\text{int}}] \text{ , as } \overline{\mathbf{F}}^{\text{int}}_{/\mathcal{S}} \equiv \overline{\mathbb{C}} \, \overline{\mathbf{E}}) . \tag{58}$$

As the last relevant remarks, it is worth mentioning that, if we limit our analysis to the classical first-order case in $h$ or when $\varrho = 0$ (so that $a_{IJ} = c_{IJ} = 1$ and $b_{IJ} = 0$ for $I, J = 1, 2$), then more compact formulas can also be used for the 2D shell constitutive law and potential energy. These can be split into membrane, bending, and shear contributions while using the mid-surface strains introduced in Equation (21). Such standard shortened formulations are, however, less convenient for PHs as (in general) $A_1 \neq A_2$ or are non-constant. To finish, one can also mention that the prior equations reduce to the Mindlin–Reissner moderately thick plate theory when the curvatures ($\min_{I=1,2}\{R_{n_I}\} = \infty$) are set to zero.

### 3.2.2. Effective External Force and Couple Resultants

We considered boundary problems that describe the linear elastodynamics of the shell undergoing the actions of different distributed forces in the bulk $\mathcal{V} \overset{\text{def}}{=} \overline{\mathcal{S}} \times [-h/2, h/2]$ and on a part of its surface boundary $\partial\mathcal{V}$. In principle, in shell analyses, *essential* (i.e., *kinematic*) *loading conditions* that notably involve some components of the displacement in Equation (24) can only be imposed on some part $\overline{\mathcal{C}}_{\boldsymbol{u}} \times [-h/2, h/2] \subseteq \mathcal{S}_\ell$, while the *natural* (i.e., *sthenic*) *loading* ones may arbitrary be imposed on any of portion of $\partial\mathcal{V}$. On $\mathcal{S}_\ell$, mixed (i.e., involving essential and natural) boundary conditions can also be applied; moreover, the prescribed components of displacements can still be associated with the reaction efforts, and we treat them hereafter as the other prescribed ones.

As in [90] (Chapter 9), the shell is subjected then to applied body forces of density $\boldsymbol{f} = f_I \boldsymbol{t}_I + f_3 \boldsymbol{n}$ per unit volume in the bulk of $\mathcal{V}$ and possibly boundary forces of density $\boldsymbol{F} = F_I \boldsymbol{t}_I + F_3 \boldsymbol{n}$ per unit surface on the boundary $\partial\mathcal{V}$; on the lateral surface $\mathcal{S}_\ell$, the components of that latter are more judiciously expressed by $\boldsymbol{F} = F_\tau \bar{\boldsymbol{\tau}} + F_\nu \bar{\boldsymbol{\nu}} + F_3 \boldsymbol{n}$ and $F_\nu = \bar{\boldsymbol{\nu}} \cdot \boldsymbol{F}$ with $F_\tau = \bar{\boldsymbol{\tau}} \cdot \boldsymbol{F}$. For simplicity, we assumed nevertheless that there is no concentrated load applied at the kinematical free edges and sharp corners of $\partial\mathcal{V}$ and, so, on the kinematical free vertices $\{\bar{s}^c_\ell\}$ of $\overline{\mathcal{C}} \setminus \overline{\mathcal{C}}_{\boldsymbol{u}}$.

**D'Alembert–Lagrange's generalized internal forces $\mathcal{Q}^{\text{ext}}_{\mathfrak{q}}$ of the NR-model.** As for the internal stress, the external loads can also be substituted by an effective system of force and couple loads considering the variational work (or virtual power) of the 3D loads $\boldsymbol{f}$ and $\boldsymbol{F}$ for any kinematically admissible field of variation $\underline{\boldsymbol{u}}$ as in Equation (26):

$$\mathscr{P}^{\text{ext}}[\boldsymbol{u}, \underline{\boldsymbol{u}}](t) \overset{\text{def}}{=} \int_{\mathcal{V}} \boldsymbol{f}(\boldsymbol{\alpha}, t) \cdot \underline{\boldsymbol{u}}(\boldsymbol{\alpha}) \, d\mathcal{V} + \int_{\partial\mathcal{V}} \boldsymbol{F}(\boldsymbol{\alpha}, t) \cdot \underline{\boldsymbol{u}}(\boldsymbol{\alpha}) \, d\partial\mathcal{V} . \tag{59}$$

The final contributions for Equation (28):

$$\overline{\mathscr{P}}^{\text{ext}}[\mathfrak{q}, \underline{\mathfrak{q}}](t) \equiv \overline{\mathscr{P}}^{\text{ext}}_{/\mathcal{S}}[\mathfrak{q}, \underline{\mathfrak{q}}](t) + \overline{\mathscr{P}}^{\text{ext}}_{/\overline{\mathcal{C}}}[\mathfrak{q}, \underline{\mathfrak{q}}](t) \tag{60a}$$

with

$$\begin{aligned}
\overline{\mathscr{P}}^{\text{ext}}_{/\mathcal{S}}[\mathfrak{q}, \underline{\mathfrak{q}}] &\overset{\text{def}}{=} \int_{\overline{\mathcal{S}}} [\bar{\boldsymbol{p}} \cdot \underline{\bar{\boldsymbol{u}}} + \bar{\boldsymbol{m}} \cdot \underline{\boldsymbol{\theta}}] d\overline{\mathcal{S}} \equiv \int_{\overline{\mathcal{S}}} \underline{\mathfrak{q}}^{\text{T}} \mathcal{Q}^{\text{ext}}_{\mathfrak{q}} d\overline{\mathcal{S}} \\
\overline{\mathscr{P}}^{\text{ext}}_{/\overline{\mathcal{C}}}[\mathfrak{q}, \underline{\mathfrak{q}}] &\overset{\text{def}}{=} \oint_{\overline{\mathcal{C}}} [\bar{\boldsymbol{\lambda}}(\bar{s}_\ell) \cdot \underline{\bar{\boldsymbol{u}}}(\bar{s}_\ell) + \bar{\boldsymbol{\mu}}(\bar{s}_\ell) \cdot \underline{\boldsymbol{\theta}}(\bar{s}_\ell)] d\bar{s}_\ell \equiv \oint_{\overline{\mathcal{C}}} \underline{\mathfrak{q}}^{\text{T}} \mathcal{Q}^{\text{ext}}_{\mathfrak{q}} d\bar{s}_\ell
\end{aligned} \tag{60b}$$

come then with all possible applied boundary force resultants and moment resultants for the shell, while over $\overline{\mathcal{S}}$

$$\mathcal{Q}^{\text{ext}}_{\bar{u}_i} = \bar{p}_i \quad , \quad \mathcal{Q}^{\text{ext}}_{\theta_I} = \bar{m}_I \text{ , for } (i, I) \in \{1, 2, 3\} \times \{1, 2\} \tag{60c}$$

and over $\overline{\mathcal{C}}$

$$\mathcal{Q}^{\text{ext}}_{\bar{u}_\tau} = \bar{\lambda}_\tau \quad , \quad \mathcal{Q}^{\text{ext}}_{\bar{u}_\nu} = \bar{\lambda}_\nu \quad , \quad \mathcal{Q}^{\text{ext}}_{\bar{u}_3} = \bar{\lambda}_3 \quad , \quad \mathcal{Q}^{\text{ext}}_{\theta_\tau} = \bar{\mu}_\tau \quad , \quad \mathcal{Q}^{\text{ext}}_{\theta_\nu} = \bar{\mu}_\nu . \tag{60d}$$

Again, on the boundary $\overline{\mathcal{C}}$, $\mathcal{Q}^{\text{ext}}_{\bar{u}_\tau}$, $\mathcal{Q}^{\text{ext}}_{\bar{u}_\nu}$, $\mathcal{Q}^{\text{ext}}_{\bar{u}_3}$, $\mathcal{Q}^{\text{ext}}_{\theta_\tau}$, and $\mathcal{Q}^{\text{ext}}_{\theta_\nu}$ are presumably unknown reaction forces or moments (per unit length $d\bar{s}_\ell$ of the curvilinear contour portion) where their related kinematic variables $\bar{u}_\tau$, $\bar{u}_\nu$, $\bar{u}_3$, $\theta_\tau$, and $\theta_\nu$ are prescribed, respectively. The external loads on the shell volume $\mathcal{V}$ and surfaces $\partial\mathcal{V}$ reduce to sthenically equivalent loads acting in the mid-surface $\overline{\mathcal{S}}$, as introduced previously. Thus, the efficient loads acting within the mid-surface $\overline{\mathcal{S}}$ of the shell $\mathcal{V}$ are as

$$\begin{aligned}
\bar{p} = \bar{p}_I t_I + \bar{p}_3 n \text{ with } \bar{p}_i(\bar{\boldsymbol{\alpha}}) \;\overset{\text{def}}{=}\; & \int_{-h/2}^{h/2} f_i(\boldsymbol{\alpha})\left(1 + 2\alpha_3\mathcal{H} + \alpha_3^2\mathcal{G}\right) d\alpha_3 \\
& + F_i(\bar{\boldsymbol{\alpha}}, -\frac{h}{2})\left(1 - 2h\mathcal{H} + h^2\mathcal{G}\right) \\
& + F_i(\bar{\boldsymbol{\alpha}}, \frac{h}{2})\left(1 + 2h\mathcal{H} + h^2\mathcal{G}\right) , \text{ over } \Omega
\end{aligned} \tag{61a}$$

$$\begin{aligned}
\overline{m} = \overline{m}_I t_I \text{ with } \overline{m}_I(\bar{\boldsymbol{\alpha}}) \;\overset{\text{def}}{=}\; & \epsilon_{IJ} \int_{-h/2}^{h/2} \alpha_3 F_J(\boldsymbol{\alpha})\left(1 + 2\alpha_3\mathcal{H} + \alpha_3^2\mathcal{G}\right) d\alpha_3 \\
& - \epsilon_{IJ}\frac{h}{2}F_J(\bar{\boldsymbol{\alpha}}, -\frac{h}{2})\left(1 - 2h\mathcal{H} + h^2\mathcal{G}\right) \\
& + \epsilon_{IJ}\frac{h}{2}F_J(\bar{\boldsymbol{\alpha}}, \frac{h}{2})\left(1 + 2h\mathcal{H} + h^2\mathcal{G}\right) , \text{ over } \Omega .
\end{aligned} \tag{61b}$$

These expressions involve S. Germain's mean curvature $\mathcal{H}$ and K.F. Gauss' curvature $\mathcal{G}$, which read as

$$\mathcal{H} \;\overset{\text{def}}{=}\; \frac{1}{2}\overline{\nabla} \cdot n \equiv \frac{1}{2}\text{tr}(\overline{\nabla}n) \equiv \frac{1}{2R_{n1}} + \frac{1}{2R_{n2}} \tag{61c}$$

$$\mathcal{G} \;\overset{\text{def}}{=}\; \det(\overline{\nabla}n) \equiv \frac{1}{R_{n1}R_{n2}} \equiv \mathcal{H}^2 - \frac{36\rho^2}{h^2} . \tag{61d}$$

Besides, the efficient loads acting on the mid-surface boundary contour $\overline{\mathcal{C}}$ (per unit length $d\bar{s}_\ell$) come (in slight contrast with [76], but comparable to the formulation in [90] Chapter 9, pp. 329–330) as

$$\begin{aligned}
\bar{\lambda} \;=\; & \bar{\lambda}_\tau \bar{\boldsymbol{\tau}} + \bar{\lambda}_\nu \bar{\boldsymbol{\nu}} + \bar{\lambda}_3 \boldsymbol{n} \\
& \text{with } \bar{\lambda}_\gamma(\bar{s}_\ell) \overset{\text{def}}{=} \int_{-h/2}^{h/2} F_\gamma(\bar{\boldsymbol{\alpha}}(\bar{s}_\ell), \alpha_3) B_\ell \, d\alpha_3 \text{ for } \gamma \in \{\tau, \nu, 3\}
\end{aligned} \tag{61e}$$

$$\begin{aligned}
\bar{\mu} \;=\; & \bar{\mu}_\tau \bar{\boldsymbol{\tau}} + \bar{\mu}_\nu \bar{\boldsymbol{\nu}} \\
& \text{with } \begin{cases} \bar{\mu}_\gamma(\bar{s}_\ell) \overset{\text{def}}{=} \int_{-h/2}^{h/2} [\delta_{\gamma\tau}F_\nu(\bar{\boldsymbol{\alpha}}(\bar{s}_\ell), \alpha_3) - \delta_{\gamma\nu}F_\tau(\bar{\boldsymbol{\alpha}}(\bar{s}_\ell), \alpha_3)]\alpha_3 B_\ell \, d\alpha_3 \\ \text{for } \gamma \in \{\tau, \nu\} \end{cases}
\end{aligned} \tag{61f}$$

3.2.3. D'Alembert–Lagrange's Generalized Acceleration (Inertial) Forces and Couple Resultants

As before, starting from the standard 3D definition of the AVP:

$$\mathscr{P}^{\text{acc}}[\boldsymbol{u}, \underaccent{\sim}{\boldsymbol{u}}](t) \overset{\text{def}}{=} \int_{\mathcal{V}} \rho\ddot{\boldsymbol{u}}(\boldsymbol{\alpha}, t) \cdot \underaccent{\sim}{\boldsymbol{u}}(\boldsymbol{\alpha}) \, d\mathcal{V} \tag{62}$$

where $\rho(\bar{\boldsymbol{\alpha}})$ represents the 3D mass density, we obtain then the 2D expression of the acceleration virtual power:

$$\overline{\mathscr{P}}^{\mathrm{acc}}[\mathsf{q},\underset{\sim}{\mathsf{q}}] \overset{\mathrm{def}}{=} \int_{\overline{S}}\int_{-h/2}^{h/2}\rho\Big[\big(\ddot{u}_I+\alpha_3\ddot{\beta}_I\big)\big(\underset{\sim}{\bar{u}}_I+\alpha_3\underset{\sim}{\beta}_I\big)+\ddot{u}_3\underset{\sim}{\bar{u}}_3\Big]\Big[1+2\alpha_3\mathcal{H}+\alpha_3^2\mathcal{G}\Big]d\alpha_3 d\overline{S}$$

$$\equiv \int_{\overline{S}}\rho h\bigg[\Big(1+\frac{h^2}{12}\mathcal{G}\Big)\ddot{u}_i\underset{\sim}{\bar{u}}_i+\frac{h^2}{6}\mathcal{H}\big(\ddot{u}_I\underset{\sim}{\beta}_I+\ddot{\beta}_I\underset{\sim}{\bar{u}}_I\big)$$

$$+\frac{h^2}{4}\Big(\frac{1}{3}+\frac{h^2}{20}\mathcal{G}\Big)\ddot{\beta}_I\underset{\sim}{\beta}_I\bigg]d\overline{S}$$

$$\equiv \int_{\overline{S}}\rho h\bigg\{\Big[\Big(1+\frac{h^2}{12}\mathcal{G}\Big)\ddot{u}_i+\epsilon_{iJ3}\frac{h^2\mathcal{H}}{6}\ddot{\theta}_J\Big]\underset{\sim}{\bar{u}}_i$$

$$+\Big[\epsilon_{JI}\frac{h^2\mathcal{H}}{6}\ddot{u}_J+\Big(\frac{h^2}{12}+\frac{h^4\mathcal{G}}{80}\Big)\ddot{\theta}_I\Big]\underset{\sim}{\theta}_I\bigg\}d\overline{S}\,. \tag{63}$$

Here, we used the fact that $\beta_I=\epsilon_{IJ}\theta_J\equiv\epsilon_{IJ3}\theta_J$, while introducing the Levi-Civita's 3D permutation (or alternating) symbol:

$$\epsilon_{ijk} \overset{\mathrm{def}}{=} \det\begin{pmatrix}\delta_{i1}&\delta_{i2}&\delta_{i3}\\\delta_{j1}&\delta_{j2}&\delta_{j3}\\\delta_{k1}&\delta_{k2}&\delta_{k3}\end{pmatrix}$$

$$\equiv \begin{cases} 1 & , \quad \text{if } (i,j,k) \text{ are an even permutation of } (1,2,3)\\ -1 & , \quad \text{if } (i,j,k) \text{ are an odd permutation of } (1,2,3)\\ 0 & , \quad \text{if any two indices are the same number}\end{cases}. \tag{64}$$

We subsequently inferred as the D'Alembert–Lagrange generalized acceleration forces in Equation (29) related on $\overline{S}$ to the *virtual kinematic variable* $\underset{\sim}{q}$:

$$\mathcal{Q}_{\bar{u}_i}^{\mathrm{acc}} = \rho h\bigg[\Big(1+\frac{h^2}{12}\mathcal{G}\Big)\ddot{u}_i+\epsilon_{iJ3}\frac{h^2}{6}\mathcal{H}\ddot{\theta}_J\bigg]\,,\ \text{for } i=1,2,3 \tag{65a}$$

$$\mathcal{Q}_{\theta_I}^{\mathrm{acc}} = \frac{\rho h^2}{12}\bigg[2\epsilon_{JI}\mathcal{H}\ddot{u}_J+\Big(1+\frac{3h^2}{20}\mathcal{G}\Big)\ddot{\theta}_I\bigg]\,,\ \text{for } I=1,2\,. \tag{65b}$$

On the other hand, there is no surface contribution for the AVP of the NR model, i.e.,

$$\overline{\mathscr{P}}^{\mathrm{acc}}[\mathsf{q},\underset{\sim}{\mathsf{q}}] = \overline{\mathscr{P}}_{/\overline{S}}^{\mathrm{acc}}[\mathsf{q},\underset{\sim}{\mathsf{q}}] \text{ as } \overline{\mathscr{P}}_{/\overline{C}}^{\mathrm{acc}}[\mathsf{q},\underset{\sim}{\mathsf{q}}]=0$$

and so

$$\mathcal{Q}_{q/\overline{C}}^{\mathrm{acc}}[\mathsf{q}]=0 \text{ for } q\in\{\bar{u}_\tau,\theta_\tau,\bar{u}_\nu,\theta_\nu,\bar{u}_3\} \text{ on } \overline{C}\,. \tag{66}$$

Lately, as the 3D model that is related to a well-known potential (of inertial forces) named kinetic co-energy, the 2D shell also derives from an analogous one that reads as

$$\overline{\mathscr{K}}[\mathsf{q},\dot{\mathsf{q}}] \equiv \frac{1}{2}\int_{\overline{S}}\rho h\bigg[\Big(1+\frac{h^2}{12}\mathcal{G}\Big)\dot{u}_i\dot{u}_i+\frac{h^2}{6}\mathcal{H}\dot{u}_I\dot{\beta}_I+\frac{h^2}{4}\Big(\frac{1}{3}+\frac{h^2}{20}\mathcal{G}\Big)\dot{\beta}_I\dot{\beta}_I\bigg]d\overline{S}\,. \tag{67}$$

**Matrix formulation of the acceleration power and the kinetic co-energy and energy.** The equivalent matrix-based forms that will be convenient for our PHS purposes read as follows:

$$\overline{\mathscr{P}}_{/\overline{S}}^{\mathrm{acc}}[\mathsf{q},\underset{\sim}{\mathsf{q}}] \equiv \int_{\overline{S}}\underset{\sim}{\mathsf{q}}^{\mathsf{T}}\dot{\pi}[\dot{\mathsf{q}}]\,d\overline{S}\text{, with}$$

$$\pi_{/\overline{S}}[\dot{\mathsf{q}}] \equiv \begin{Bmatrix}\pi_{\bar{u}_1}\\\pi_{\theta_1}\\\pi_{\bar{u}_2}\\\pi_{\theta_2}\\\pi_{\bar{u}_3}\end{Bmatrix}\overset{\mathrm{def}}{=}\delta_{\dot{\mathsf{q}}/\overline{S}}\overline{\mathscr{K}}[\mathsf{q},\dot{\mathsf{q}}]\,,\ \dot{\pi}_{/\overline{S}}[\dot{\mathsf{q}}]\equiv\mathcal{Q}_{\mathsf{q}/\overline{S}}^{\mathrm{acc}}[\mathsf{q}]=\mathbb{M}\ddot{\mathsf{q}}\,; \tag{68a}$$

$$\overline{\mathscr{P}}^{\mathrm{acc}}_{/\overline{C}}[\mathfrak{q},\underline{\mathfrak{q}}] \equiv \oint_{\overline{C}} \mathfrak{q}^{\mathrm{T}}\,\dot{\boldsymbol{\pi}}[\dot{\mathfrak{q}}]\,d\bar{s}_\ell\,,\ \text{with}$$

$$\boldsymbol{\pi}_{/\overline{C}}[\dot{\mathfrak{q}}] \equiv \begin{Bmatrix} \pi_{\bar{u}_\tau} \\ \pi_{\theta_\tau} \\ \pi_{\bar{u}_\nu} \\ \pi_{\theta_\nu} \\ \pi_{\bar{u}_3} \end{Bmatrix} \overset{\text{def}}{=} \delta_{\dot{\mathfrak{q}}_{/\overline{C}}}\overline{\mathscr{K}}[\mathfrak{q},\dot{\mathfrak{q}}]\,,\ \dot{\boldsymbol{\pi}}_{/\overline{C}}[\dot{\mathfrak{q}}] \equiv \mathcal{Q}^{\mathrm{acc}}_{\mathfrak{q}_{/\overline{C}}}[\mathfrak{q}] = \mathbf{0}_{5\times 1}\,;\tag{68b}$$

$$\overline{\mathscr{K}}[\mathfrak{q},\dot{\mathfrak{q}}] \equiv \frac{1}{2}\int_{\overline{S}}\dot{\mathfrak{q}}^{\mathrm{T}}\mathbb{M}\dot{\mathfrak{q}}\,d\overline{S} \equiv \overline{\mathscr{K}}_c[\boldsymbol{\pi}] \equiv \frac{1}{2}\int_{\overline{S}}\mathfrak{p}^{\mathrm{T}}\mathbb{M}^{-1}\mathfrak{p}\,d\overline{S}\,,\ \text{with}$$
$$\mathfrak{p} \overset{\text{def}}{=} \frac{1}{2}\frac{\partial}{\partial\dot{\mathfrak{q}}}(\dot{\mathfrak{q}}^{\mathrm{T}}\mathbb{M}\dot{\mathfrak{q}})\ \text{on}\ \overline{S}\cup\overline{C}\,.\tag{68c}$$

For more generality (which is necessary for inertially enhanced models), we distinguish here two different notions of momenta, which fortuitously turn out here to be equal up to time constant on $\overline{S}$, $\boldsymbol{\pi}_{/\overline{S}}[\dot{\mathfrak{q}}] \equiv \mathfrak{p}_{/\overline{S}} \equiv \mathbb{M}_{/\overline{S}}\dot{\mathfrak{q}}_{/\overline{S}}$: the first *Newtonian kinematic* (or *kinetic*) one $\mathfrak{p}$ provides the *linear momenta* defined in classical (Euler–Newton) sense:

$$\mathfrak{p} \equiv \begin{Bmatrix} \mathfrak{p}_{\bar{u}_1} \\ \mathfrak{p}_{\theta_1} \\ \mathfrak{p}_{\bar{u}_2} \\ \mathfrak{p}_{\theta_2} \\ \mathfrak{p}_{\bar{u}_3} \end{Bmatrix}\ \text{on}\ \overline{S}\ \text{and}\ \mathfrak{p} \equiv \begin{Bmatrix} \mathfrak{p}_{\bar{u}_\tau} \equiv \bar{\tau}_I\mathfrak{p}_{\bar{u}_I} \\ \mathfrak{p}_{\theta_\tau} \equiv \bar{\tau}_I\mathfrak{p}_{\theta_I} \\ \mathfrak{p}_{\bar{u}_\nu} \equiv \bar{\nu}_I\mathfrak{p}_{\bar{u}_I} \\ \mathfrak{p}_{\theta_\nu} \equiv \bar{\nu}_I\mathfrak{p}_{\theta_I} \\ \mathfrak{p}_{\bar{u}_3} \end{Bmatrix} \text{with} \begin{cases} \mathfrak{p}_{\bar{u}_\tau}\dot{\bar{u}}_\tau + \mathfrak{p}_{\bar{u}_\nu}\dot{\bar{u}}_\nu \equiv \mathfrak{p}_{\bar{u}_I}\dot{\bar{u}}_I \\ \mathfrak{p}_{\theta_\tau}\dot{\theta}_\tau + \mathfrak{p}_{\theta_\nu}\dot{\theta}_\nu \equiv \mathfrak{p}_{\theta_I}\dot{\theta}_I \end{cases}\ \text{on}\ \overline{C};\tag{69}$$

the second *Lagrangian* one $\boldsymbol{\pi}[\dot{\mathfrak{q}}]$ provides the *generalized/canonical momenta* that come from the (Euler–Lagrange) variational calculus as the conjugate to the generalized velocity $\dot{\mathfrak{q}}$. One observes that their counterparts on the boundary differ for the NR shell model, as

$$\boldsymbol{\pi}_{/\overline{C}}[\dot{\mathfrak{q}}] \equiv \mathbf{0}_{5\times 1} \neq \mathfrak{p}_{/\overline{C}} \equiv \mathbb{M}_{/\overline{C}}\dot{\mathfrak{q}}_{/\overline{C}}\,,\tag{70}$$

owing to the fact that, in general, on $\overline{C}$, the displacements introduced in Equation (24) are $\mathfrak{q}_{/\overline{C}} \neq 0$, and their time variations $\dot{\mathfrak{q}}_{/\overline{C}} \neq 0$. Besides, we also have the $(5\times 5)$ symmetric (and positive definite) matrices of the surface mass and inertia moment densities:

$$\mathbb{M}_{/\overline{S}} \overset{\text{def}}{=} \rho h \begin{array}{c} \begin{matrix} \dot{\bar{u}}_1 & \quad\dot{\theta}_1 & \quad\dot{\bar{u}}_2 & \quad\dot{\theta}_2 & \quad\dot{\bar{u}}_3 \end{matrix} \\ \begin{pmatrix} \frac{12+h^2\mathcal{G}}{12} & 0 & 0 & \frac{h^2\mathcal{H}}{6} & 0 \\ 0 & \frac{h^2}{12}+\frac{h^4\mathcal{G}}{80} & -\frac{h^2\mathcal{H}}{6} & 0 & 0 \\ 0 & -\frac{h^2\mathcal{H}}{6} & \frac{12+h^2\mathcal{G}}{12} & 0 & 0 \\ \frac{h^2\mathcal{H}}{6} & 0 & 0 & \frac{h^2}{12}+\frac{h^4\mathcal{G}}{80} & 0 \\ 0 & 0 & 0 & 0 & \frac{12+h^2\mathcal{G}}{12} \end{pmatrix} \end{array} \begin{matrix} \mathfrak{p}_{\bar{u}_1} \\ \mathfrak{p}_{\theta_1} \\ \mathfrak{p}_{\bar{u}_2} \\ \mathfrak{p}_{\theta_2} \\ \mathfrak{p}_{\bar{u}_3} \end{matrix}\tag{71a}$$

$$\mathbb{M}_{/\overline{C}} \overset{\text{def}}{=} \rho h \begin{array}{c} \begin{matrix} \dot{\bar{u}}_\tau & \quad\dot{\theta}_\tau & \quad\dot{\bar{u}}_\nu & \quad\dot{\theta}_\nu & \quad\dot{\bar{u}}_3 \end{matrix} \\ \begin{pmatrix} \frac{12+h^2\mathcal{G}}{12} & 0 & 0 & -\frac{h^2\mathcal{H}}{6} & 0 \\ 0 & \frac{h^2}{12}+\frac{h^4\mathcal{G}}{80} & \frac{h^2\mathcal{H}}{6} & 0 & 0 \\ 0 & \frac{h^2\mathcal{H}}{6} & \frac{12+h^2\mathcal{G}}{12} & 0 & 0 \\ -\frac{h^2\mathcal{H}}{6} & 0 & 0 & \frac{h^2}{12}+\frac{h^4\mathcal{G}}{80} & 0 \\ 0 & 0 & 0 & 0 & \frac{12+h^2\mathcal{G}}{12} \end{pmatrix} \end{array} \begin{matrix} \mathfrak{p}_{\bar{u}_\tau} \\ \mathfrak{p}_{\theta_\tau} \\ \mathfrak{p}_{\bar{u}_\nu} \\ \mathfrak{p}_{\theta_\nu} \\ \mathfrak{p}_{\bar{u}_3} \end{matrix}\,.\tag{71b}$$

As already expected and observed with the former simplifications that arise in the case of thin shells, where $h|\mathcal{H}| \ll 1$ and $h^2|\mathcal{G}| \ll 1$, the translational and rotational inertial contributions are also uncoupled at the first approximation order in $h/\max\limits_{I=1,2}\{R_{n_I}\}$.

As for the elastic energy, the complementary kinetic energy introduced in Equation (68) (with also, in fact, a necessary condition of maximization) also results from the Legendre–Fenchel transformation:

$$\overline{\mathscr{K}}_c[\boldsymbol{\pi}] \stackrel{\text{def}}{=} \sup_{\text{k.a. } \dot{\mathfrak{q}}} \left\{ \int_{\overline{\mathcal{S}}} \dot{\mathfrak{q}}^{\mathrm{T}} \boldsymbol{\pi} \, d\overline{\mathcal{S}} + \oint_{\overline{\mathcal{C}}} \dot{\mathfrak{q}}^{\mathrm{T}} \boldsymbol{\pi} \, d\bar{s}_\ell - \overline{\mathscr{K}}[\mathfrak{q}, \dot{\mathfrak{q}}] \right\}. \tag{72}$$

Moreover, when $\underline{\mathfrak{q}} \equiv \dot{\mathfrak{q}}$, then Equation (68) provides the (real) kinetic power due to the accelerations:

$$\overline{\mathscr{P}}^{\text{acc}}[\mathfrak{q}, \dot{\mathfrak{q}}] \equiv \dot{\overline{\mathscr{K}}}[\mathfrak{q}, \dot{\mathfrak{q}}] \quad (\equiv \dot{\overline{\mathscr{K}}}_c[\boldsymbol{\pi}], \text{ as } \boldsymbol{\pi}_{/\overline{\mathcal{S}}}[\dot{\mathfrak{q}}] \equiv \mathfrak{p}_{/\overline{\mathcal{S}}} \equiv \mathbb{M}_{/\overline{\mathcal{S}}} \dot{\mathfrak{q}}_{/\overline{\mathcal{S}}} \text{ and } \boldsymbol{\pi}_{/\overline{\mathcal{C}}}[\dot{\mathfrak{q}}] \equiv \mathbf{0}_{5\times 1}). \tag{73}$$

For the sake of simplicity, we shall consider hereafter the initial conditions for the displacement, velocity, and internal stress fields to be zero (i.e., rest configurations), in addition to having no impacting loads. Otherwise, in agreement with the physical principles of causality, the nonzero initial conditions for the displacement and velocity fields must be enforced variationally in terms of singularly time-localized inertial loading forces and couples following, for instance, the ideas in [91–94] and also the L. Schwartz' viewpoint adopted in [95,96]; with such a description, they can be considered as already included in the external virtual power $\overline{\mathscr{P}}^{\text{ext}}[\mathfrak{q}, \underline{\mathfrak{q}}]$.

## 4. Hamiltonian Variational Formalisms versus Canonical Ones

We turn now to the main variational purpose of this communication, which was to apply the PHS formalism to the linear elastodynamics of Naghdi–Reissner's (**NR**) shell model, by means of variational principles. As is also well known indeed [97], any classical mechanical system can be alternatively described as well from either a Lagrangian viewpoint or a Hamiltonian canonical one. The relations between these two possible formulations are fairly well established, and after having reminded briefly about the first viewpoint, we shall present now in this subsection the variational path leading to the Hamiltonian canonical ones and its currently developed PHS extension, with the purpose of clarification. Albeit that no new physics are necessarily introduced here in applying the Lagrangian/Hamiltonian mechanics and their PHS synthesis compared to the Newtonian mechanics, they provide a mathematically elegant, useful, and systematic way of presenting the system dynamic evolutions for analytical and numerical treatments (e.g., by the GNI methodology of interest). In addition, such variational principles can also be used either to derive or to solve other shell-like models [98,99].

### 4.1. Classical versus Port-Hamiltonian Canonical Equation Formalisms

In what follows, we denote $T \stackrel{\text{def}}{=} [t_i, t_f]$ as the time interval and $\partial T \stackrel{\text{def}}{=} \{t_i, t_f\}$ as its border. Then, in the spirit of Noether's variational viewpoint (and possibly, Hölder's viewpoint with both synchronous and asynchronous variations [100,101]), we let the final instant $t_f$ be an additional degree of freedom for the variational principles (some discussion on such nontrivial variations can be found, for instance, in [102]), which become Volterra's integral equations (or some kind of law of varying action [103]).

Besides, in this subsection, we focus only on a small number of alternatives for action integrals, in order to indicate the place of the PHS model within the Hamiltonian variational framework. Other formulations combining different expressions of energy functionals can be found, for instance, in [91,104].

#### 4.1.1. Classical Lagrangian Equations by Hamilton's Generalized Least-Action Principle in $\mathfrak{q}$-Configuration Space

As is well known, the dynamic equations in Equation (31) can also be recovered from the Hamilton(–Lagrange–D'Alembert) generalized variational principle ([5], Chapter 7, p. 207; Chapter 8) (which applies as well as to non-conservative, i.e., dissipative and

irreversible, processes and constrained systems as, e.g., in [99]; see also [45], Part 3, for the discrete Lagrange–d'Alembert principle):

$$\lim_{\epsilon \searrow 0^+} \frac{d\overline{\mathscr{A}}_L[\mathfrak{q} + \epsilon\, \underline{\mathfrak{q}}, t_f]}{d\epsilon} = -\int_{t_i}^{t_f} \mathscr{P}^{\text{ext}}[\mathfrak{q}, \underline{\mathfrak{q}}]\, d\hat{t} \tag{74a}$$

for all k.a. variations $\underline{\mathfrak{q}}$ and time interval $T \stackrel{\text{def}}{=} [t_i\,, t_f]$, while $\underline{\mathfrak{q}}_{/\partial T} = \mathbf{0}_{5\times 1}$ .

This re-states then Equation (31) as the necessary Euler–Lagrange condition of balance for the Gateaux first variation of the following *integral of action* $\overline{\mathscr{A}}_L[\mathfrak{q}, t_f]$ of the *Lagrangian* $\overline{\mathscr{L}}[\mathfrak{q}, \overline{\mathbf{E}}, \dot{\mathfrak{q}}]$ with the constraint in Equation (23):

$$\overline{\mathscr{A}}_L[\mathfrak{q}, t_f] \stackrel{\text{def}}{=} \int_{t_i}^{t_f} \overline{\mathscr{L}}[\mathfrak{q}, \overline{\mathbf{E}}, \dot{\mathfrak{q}}]d\hat{t}\,, \tag{74b}$$

$$\text{with } \overline{\mathscr{L}}[\mathfrak{q}, \overline{\mathbf{E}}, \dot{\mathfrak{q}}] \stackrel{\text{def}}{=} \overline{\mathscr{K}}[\mathfrak{q}, \dot{\mathfrak{q}}] - \overline{\mathscr{E}}^{\text{elas}}[\mathfrak{q}, \overline{\mathbf{E}}] \text{ and } \overline{\mathbf{E}}[\mathfrak{q}] \equiv \overline{\mathbb{B}}\,\mathfrak{q} \text{ over } \overline{\mathcal{S}}\,. \tag{74c}$$

The Lagrangian being here scleronomic (i.e., time is not present explicitly in $\overline{\mathscr{L}}$), the variations of the time $t_f$ (and so, $T$) provide, furthermore, in agreement with the power balance equation in Equation (30),

$$\overline{\mathscr{P}}^{\text{acc}}[\mathfrak{q}, \dot{\mathfrak{q}}] - \overline{\mathscr{P}}^{\text{int}}[\mathfrak{q}, \dot{\mathfrak{q}}] = \overline{\mathscr{P}}^{\text{ext}}[\mathfrak{q}, \dot{\mathfrak{q}}]\,, \text{ from } T \text{ with } \underline{\mathfrak{q}} \stackrel{\text{a.e.}}{\equiv} \dot{\mathfrak{q}} \text{ on } (\overline{\mathcal{S}} \cup \overline{\mathcal{C}}) \times (T \setminus \partial T)\,. \tag{75}$$

Here, by $\stackrel{\text{a.e.}}{\equiv}$, wherein a.e., stands for "almost everywhere", we mean an approximation is used by a convolution with a smooth positive mollifier over the open interval $T \setminus \partial T$. Accordingly, with Equations (51), (58), (68) and (73), we also have $\overline{\mathscr{P}}^{\text{acc}}[\mathfrak{q}, \dot{\mathfrak{q}}] - \overline{\mathscr{P}}^{\text{int}}[\mathfrak{q}, \dot{\mathfrak{q}}] \equiv \dot{\overline{\mathscr{K}}}[\mathfrak{q}, \dot{\mathfrak{q}}] + \dot{\overline{\mathscr{E}}}^{\text{elas}}[\mathfrak{q}, \overline{\mathbf{E}}]$, i.e., the power difference on the left-hand side of Equation (75) coincides here with the time derivative (rate of change) of the so-called "total" intrinsic mechanical energy $\overline{\mathscr{K}}[\mathfrak{q}, \dot{\mathfrak{q}}] + \overline{\mathscr{E}}^{\text{elas}}[\mathfrak{q}, \overline{\mathbf{E}}]$ stored (by inertia and elastic cohesion) in the 2D shell $\overline{\mathcal{S}}$.

### 4.1.2. Classical Canonical Hamiltonian Equations by Hamilton's Generalized Least-Action Principle in $[\mathfrak{q}, \pi]$-Phase Space

Naturally, following the Hamilton(–Lagrange–D'Alembert) generalized least-action variational principle, we recover then the governing canonical Hamiltonian equations and Hamiltonian mechanical energy rate for $t \in (T \setminus \partial T) \equiv [t_i\,, t_f]$ (see also [45], Part 3, for the discrete analog):

$$\dot{\mathfrak{q}}_{/\overline{\mathcal{S}}} = \delta_\pi \overline{\mathscr{H}}_{/\overline{\mathcal{S}}}[\mathfrak{q}, \pi]\,, \text{ from } \underline{\pi} \text{ on } \overline{\mathcal{S}} \tag{76a}$$

$$\dot{\pi}_{/\overline{\mathcal{S}}} = -\delta_\mathfrak{q} \overline{\mathscr{H}}_{/\overline{\mathcal{S}}}[\mathfrak{q}, \pi] + \mathcal{Q}^{\text{ext}}_{\mathfrak{q}/\overline{\mathcal{S}}}[\mathfrak{q}] \equiv -\delta_{\mathfrak{q}/\overline{\mathcal{S}}} \overline{\mathscr{E}}^{\text{elas}}[\mathfrak{q}, \overline{\mathbf{E}}] + \mathcal{Q}^{\text{ext}}_{\mathfrak{q}/\overline{\mathcal{S}}}[\mathfrak{q}]$$
$$\equiv \mathcal{Q}^{\text{int}}_{\mathfrak{q}/\overline{\mathcal{S}}}[\mathfrak{q}] + \mathcal{Q}^{\text{ext}}_{\mathfrak{q}/\overline{\mathcal{S}}}[\mathfrak{q}] \equiv \mathcal{Q}^{\text{acc}}_{\mathfrak{q}}[\mathfrak{q}]\,, \text{ from } \underline{\mathfrak{q}} \text{ on } \overline{\mathcal{S}} \tag{76b}$$

$$\dot{\pi}_{/\overline{\mathcal{C}}} = -\delta_\mathfrak{q} \overline{\mathscr{H}}_{/\overline{\mathcal{C}}}[\mathfrak{q}, \pi] + \mathcal{Q}^{\text{ext}}_{\mathfrak{q}/\overline{\mathcal{C}}}[\mathfrak{q}] \equiv -\delta_{\mathfrak{q}/\overline{\mathcal{C}}} \overline{\mathscr{E}}^{\text{elas}}[\mathfrak{q}, \overline{\mathbf{E}}] + \mathcal{Q}^{\text{ext}}_{\mathfrak{q}/\overline{\mathcal{C}}}[\mathfrak{q}]$$
$$\equiv \mathcal{Q}^{\text{int}}_{\mathfrak{q}/\overline{\mathcal{C}}}[\mathfrak{q}] + \mathcal{Q}^{\text{ext}}_{\mathfrak{q}/\overline{\mathcal{C}}}[\mathfrak{q}]\,, \text{ from } \underline{\mathfrak{q}} \text{ on } \overline{\mathcal{C}} \tag{76c}$$

$$\dot{\overline{\mathscr{H}}}[\mathfrak{q}, \pi] = \mathscr{P}^{\text{ext}}[\mathfrak{q}, \dot{\mathfrak{q}}]\,, \text{ from } T \text{ with } (\underline{\mathfrak{q}}, \underline{\pi}) \stackrel{\text{a.e.}}{\equiv} (\dot{\mathfrak{q}}, \dot{\pi}) \text{ on } (\overline{\mathcal{S}} \cup \overline{\mathcal{C}}) \times (T \setminus \partial T) \tag{76d}$$

as the necessary Euler–Lagrange condition of balance for the variation:

$$\lim_{\epsilon \searrow 0^+} \frac{d\overline{\mathscr{A}}_H[\mathfrak{q} + \epsilon\, \underline{\mathfrak{q}}, \pi + \epsilon\, \underline{\pi}, t_f]}{d\epsilon} = -\int_{t_i}^{t_f} \mathscr{P}^{\text{ext}}[\mathfrak{q}, \underline{\mathfrak{q}}]\, d\hat{t}\,, \tag{77a}$$

for all admissible variations $(\underline{\mathfrak{q}}, \underline{\pi}, T)$, while $T = [t_i\,, t_f]$ and $\underline{\mathfrak{q}}_{/\partial T} = \mathbf{0}_{5\times 1}$

of the following "integral of action":

$$\overline{\mathscr{A}}_{\mathrm{H}}[\mathfrak{q}, \boldsymbol{\pi}, t_f] \overset{\text{def}}{=} \int_{t_i}^{t_f} \left\{ \int_{\overline{\mathcal{S}}} \dot{\mathfrak{q}}^{\mathrm{T}} \boldsymbol{\pi} \, d\overline{\mathcal{S}} + \oint_{\overline{\mathcal{C}}} \dot{\mathfrak{q}}^{\mathrm{T}} \boldsymbol{\pi} \, d\bar{s}_\ell - \overline{\mathscr{H}}[\mathfrak{q}, \boldsymbol{\pi}] \right\} d\hat{\imath} \,. \tag{77b}$$

Here, we classically introduce the following Hamiltonian functional, which can also be interpreted as the Fenchel–Legendre transform of the Lagrangian $\overline{\mathscr{L}}[\mathfrak{q}, \overline{\mathbf{E}}, \dot{\mathfrak{q}}]$ of the NR 2D shell model:

$$\overline{\mathscr{H}}[\mathfrak{q}, \boldsymbol{\pi}] \overset{\text{def}}{=} \overline{\mathscr{K}}_c[\boldsymbol{\pi}] + \overline{\mathscr{E}}^{\mathrm{elas}}[\mathfrak{q}, \overline{\mathbf{E}}] \,. \tag{78}$$

This, of course, coincides here with the "total" intrinsic mechanical energy $\overline{\mathscr{K}}[\mathfrak{q}, \dot{\mathfrak{q}}] + \overline{\mathscr{E}}^{\mathrm{elas}}[\mathfrak{q}, \overline{\mathbf{E}}]$ stored (by inertia and elastic cohesion) in the 2D shell $\overline{\mathcal{S}}$ as $\boldsymbol{\pi}_{/\overline{\mathcal{S}}}[\dot{\mathfrak{q}}] \equiv \mathfrak{p}_{/\overline{\mathcal{S}}} \equiv \mathbb{M}_{/\overline{\mathcal{S}}} \dot{\mathfrak{q}}_{/\overline{\mathcal{S}}}$ and $\boldsymbol{\pi}_{/\overline{\mathcal{C}}}[\dot{\mathfrak{q}}] \equiv \mathbf{0}_{5 \times 1}$.

Equations (76) are notably given for sufficient regular Hamiltonian state fields, for which $(\mathfrak{q}, \overline{\mathbf{E}}, \boldsymbol{\pi})$ must satisfy the Schwarz' (or Clairaut's or Young's) condition of integrability/compatibility/consistency related to Equation (23):

$$\dot{\overline{\mathbf{E}}} \equiv \overline{\mathbb{B}} \dot{\mathfrak{q}}_{/\overline{\mathcal{S}}} \equiv \overline{\mathbb{B}} \delta_{\boldsymbol{\pi}} \overline{\mathscr{H}} \equiv \overline{\mathbb{B}} \delta_{\boldsymbol{\pi}} \overline{\mathscr{K}}_c \,, \text{ over } \overline{\mathcal{S}} \,. \tag{79}$$

(Let us mention in passing that in the case of shock waves (or a surface of discontinuity of first-order) where the velocity and strain fields are discontinuous, we have to consider the derivative in L. Schwartz's sense or introduced on the surface of derivative discontinuities Hadamard-Maxwell's compatibility conditions in addition to that one and the Rankine–Hugoniot's jump relations [87], Sections 72 and 73, pages 248–256; [105]. These conditions are required notably for nonsmooth mechanics problems [106].)

Subsequently, we can indifferently express the well-known theorem of the (total) Hamiltonian mechanical energy flow (or rate) in Equation (76) as

$$
\begin{aligned}
\dot{\overline{\mathscr{H}}}[\mathfrak{q}, \boldsymbol{\pi}] &\equiv \dot{\overline{\mathscr{K}}}_c[\boldsymbol{\pi}] + \dot{\overline{\mathscr{E}}}^{\mathrm{elas}}[\mathfrak{q}, \overline{\mathbf{E}}] \equiv \overline{\mathscr{P}}^{\mathrm{acc}}[\mathfrak{q}, \dot{\mathfrak{q}}] - \overline{\mathscr{P}}^{\mathrm{int}}[\mathfrak{q}, \dot{\mathfrak{q}}] \equiv \overline{\mathscr{P}}^{\mathrm{ext}}[\mathfrak{q}, \dot{\mathfrak{q}}] \\
&\equiv \int_{\overline{\mathcal{S}}} \left[ \dot{\mathfrak{q}}^{\mathrm{T}} \delta_{\mathfrak{q}} \overline{\mathscr{H}} + \dot{\boldsymbol{\pi}}^{\mathrm{T}} \delta_{\boldsymbol{\pi}} \overline{\mathscr{H}} \right] d\overline{\mathcal{S}} + \oint_{\overline{\mathcal{C}}} \left[ \dot{\mathfrak{q}}^{\mathrm{T}} \delta_{\mathfrak{q}} \overline{\mathscr{H}} + \dot{\boldsymbol{\pi}}^{\mathrm{T}} \delta_{\boldsymbol{\pi}} \overline{\mathscr{H}} \right] d\bar{s}_\ell \\
&\equiv \{ \overline{\mathscr{H}}[\mathfrak{q}, \boldsymbol{\pi}], \overline{\mathscr{H}}[\mathfrak{q}, \boldsymbol{\pi}] \}
\end{aligned}
\tag{80}
$$

where, in Equation (80), $\delta_{\boldsymbol{\pi}_{/\overline{\mathcal{C}}}} \overline{\mathscr{H}} = \mathbf{0}$ and the contribution due to the time variations of the strain $\overline{\mathbf{E}}[\mathfrak{q}]$ have been integrated into those related to the time variations of $\mathfrak{q}$. As, e.g., in [29,61,107–109] (see also [5]), the last equivalence in Equation (80) uses the variational Poisson brackets, which are defined for any pair of functionals $\mathscr{F}_1[\mathfrak{q}, \boldsymbol{\pi}]$ and $\mathscr{F}_2[\mathfrak{q}, \boldsymbol{\pi}]$ as

$$
\begin{aligned}
\{ \mathscr{F}_1, \mathscr{F}_2 \} \overset{\text{def}}{=} \ & \int_{\overline{\mathcal{S}}} \left[ (\delta_{\boldsymbol{\pi}} \mathscr{F}_2)^{\mathrm{T}} (\delta_{\mathfrak{q}} \mathscr{F}_1) - (\delta_{\mathfrak{q}} \mathscr{F}_2)^{\mathrm{T}} (\delta_{\boldsymbol{\pi}} \mathscr{F}_1) \right] d\overline{\mathcal{S}} \\
& + \oint_{\overline{\mathcal{C}}} \left[ (\delta_{\boldsymbol{\pi}} \mathscr{F}_2)^{\mathrm{T}} (\delta_{\mathfrak{q}} \mathscr{F}_1) - (\delta_{\mathfrak{q}} \mathscr{F}_2)^{\mathrm{T}} (\delta_{\boldsymbol{\pi}} \mathscr{F}_1) \right] d\bar{s}_\ell \,.
\end{aligned}
\tag{81}
$$

### 4.1.3. PHS Canonical Equations by Hamilton's Generalized Least-Action Principle in $[\mathfrak{q}, \overset{\star}{\overline{\mathbf{F}}}^{\mathrm{int}}, \boldsymbol{\pi}]$-Phase Space

Less commonly evoked (but implicitly implied) as concerns at least the PHS literature of distributed systems [8] is the complementary energy of the "total" intrinsic mechanical

energy $\overline{\mathscr{K}}[\mathfrak{q},\dot{\mathfrak{q}}]+\overline{\mathscr{E}}^{\mathrm{elas}}[\mathfrak{q},\overline{\mathbf{E}}]$ (and the related variational principles), which is related to the total Fenchel–Legendre–Young transform:

$$\overline{\mathscr{H}}_c[\overline{\mathbf{F}}^{\mathrm{int}},\boldsymbol{\pi}]\overset{\mathrm{def}}{=}\overline{\mathscr{K}}_c[\boldsymbol{\pi}]+\overline{\mathscr{E}}_c^{\mathrm{elas}}[\overline{\mathbf{F}}^{\mathrm{int}}]\tag{82a}$$

$$\equiv\int_{\overline{S}}\left[\dot{\mathfrak{q}}^{\mathrm{T}}\boldsymbol{\pi}+\overline{\mathbf{E}}^{\mathrm{T}}\,\overline{\mathbf{F}}^{\mathrm{int}}\right]d\overline{S}-\overline{\mathscr{K}}[\mathfrak{q},\dot{\mathfrak{q}}]-\overline{\mathscr{E}}^{\mathrm{elas}}[\mathfrak{q},\overline{\mathbf{E}}]\,,$$

$$\mathrm{as}\ \begin{cases}\mathfrak{p}_{/\overline{S}}\equiv\mathbb{M}_{/\overline{S}}\dot{\mathfrak{q}}_{/\overline{S}}\\\overline{\mathbf{F}}_{/\overline{S}}^{\mathrm{int}}\equiv\quad\overline{\mathbb{C}}\,\overline{\mathbf{E}}\end{cases}\tag{82b}$$

while accounting, therefore, simultaneously for both Equations (53) and (68). Then, as done previously, the former theorem of the (total) mechanical energy flow (or rate) in Equation (80) can also be expressed as

$$\begin{aligned}\dot{\overline{\mathscr{H}}}_c[\overline{\mathbf{F}}^{\mathrm{int}},\boldsymbol{\pi}]&\equiv\dot{\overline{\mathscr{K}}}_c[\boldsymbol{\pi}]+\dot{\overline{\mathscr{E}}}_c^{\mathrm{elas}}[\overline{\mathbf{F}}^{\mathrm{int}}]\equiv\overline{\mathscr{P}}^{\mathrm{acc}}[\mathfrak{q},\dot{\mathfrak{q}}]-\overline{\mathscr{P}}^{\mathrm{int}}[\mathfrak{q},\dot{\mathfrak{q}}]\equiv\overline{\mathscr{P}}^{\mathrm{ext}}[\mathfrak{q},\dot{\mathfrak{q}}]\\&\equiv\int_{\overline{S}}\left[\dot{\boldsymbol{\pi}}^{\mathrm{T}}\delta_{\boldsymbol{\pi}}\overline{\mathscr{H}}_c+\left(\dot{\overline{\mathbf{F}}}^{\mathrm{int}}\right)^{\mathrm{T}}\delta_{\overline{\mathbf{F}}^{\mathrm{int}}}\overline{\mathscr{H}}_c\right]d\overline{S}\\&\quad+\oint_{\overline{C}}\left[\dot{\boldsymbol{\pi}}^{\mathrm{T}}\delta_{\boldsymbol{\pi}}\overline{\mathscr{H}}_c+\left(\dot{\overline{\mathbf{F}}}^{\mathrm{int}}\right)^{\mathrm{T}}\delta_{\overline{\mathbf{F}}^{\mathrm{int}}}\overline{\mathscr{H}}_c\right]d\overline{s}_\ell\,.\end{aligned}\tag{83}$$

Here, $\delta_{\boldsymbol{\pi}_{/\overline{C}}}\overline{\mathscr{H}}_c=\delta_{\overline{\mathbf{F}}^{\mathrm{int}}_{/\overline{C}}}\overline{\mathscr{H}}_c=\mathbf{0}$ in the second line of equivalences. Bearing in mind then for later use the following identifications for the variations (over $\overline{S}$):

$$\delta_{\mathfrak{q}}\overline{\mathscr{H}}\equiv\delta_{\mathfrak{q}}\overline{\mathscr{E}}^{\mathrm{elas}}\equiv\overline{\mathcal{Q}}_{\mathfrak{q}}^{\mathrm{int}}\quad\mathrm{and}\quad\delta_{\overline{\mathbf{E}}}\overline{\mathscr{H}}\equiv\delta_{\overline{\mathbf{E}}}\overline{\mathscr{E}}^{\mathrm{elas}}\equiv\overline{\mathbf{F}}^{\mathrm{int}}\tag{84a}$$

$$\delta_{\boldsymbol{\pi}}\overline{\mathscr{H}}_c\equiv\delta_{\boldsymbol{\pi}}\overline{\mathscr{H}}\equiv\delta_{\boldsymbol{\pi}}\overline{\mathscr{K}}_c\equiv\mathbb{M}^{-1}\boldsymbol{\pi}\equiv\dot{\mathfrak{q}}\quad\mathrm{and}\quad\delta_{\overline{\mathbf{F}}^{\mathrm{int}}}\overline{\mathscr{H}}_c\equiv\delta_{\overline{\mathbf{F}}^{\mathrm{int}}}\overline{\mathscr{E}}_c^{\mathrm{elas}}\equiv\overline{\mathbf{E}}\,,\tag{84b}$$

we can claim, therefore, that the time variation of the internal (stored) energy (or energy flow) equals (in absence of dissipated power) the power supplied by the environment on the 2D shell for both ports, i.e., its bulk domain $\overline{S}$ and its boundary $\overline{C}$.

Now, the PHS formalism considers implicitly the necessary Euler–Lagrange conditions of balance for the following variation of the "integral of action" related to the extended Hellinger–Prange–Reissner variational one [97,104]:

$$\lim_{\epsilon\searrow0^+}\frac{d\overline{\mathscr{A}}_{\mathrm{HRPH}}[\mathfrak{q}+\epsilon\,\underline{\mathfrak{q}},\overline{\mathbf{F}}^{\mathrm{int}}+\epsilon\,\overset{\star}{\underline{\overline{\mathbf{F}}}}{}^{\mathrm{int}},\boldsymbol{\pi}+\epsilon\,\underline{\boldsymbol{\pi}},t_f]}{d\epsilon}=-\int_{t_i}^{t_f}\mathscr{P}^{\mathrm{ext}}[\mathfrak{q},\underline{\mathfrak{q}}]\,d\hat{t}\tag{85a}$$

for all admissible variations $(\underline{\mathfrak{q}},\underline{\boldsymbol{\pi}},\overset{\star}{\underline{\overline{\mathbf{F}}}}{}^{\mathrm{int}},T)$, while $T=[t_i,t_f]$ and $\underline{\mathfrak{q}}_{/\partial T}=\mathbf{0}_{5\times1}$, $\overset{\star}{\underline{\overline{\mathbf{F}}}}{}^{\mathrm{int}}_{/\partial T}=\mathbf{0}$.

Here, we used the HR potential $\overline{\mathscr{E}}_{cc}^{\mathrm{elas}}[\mathfrak{q},\overline{\mathbf{F}}^{\mathrm{int}}]$ in Equation (57) in the following integral of action and Hamiltonian energy:

$$\overline{\mathscr{A}}_{\mathrm{HRPH}}[\mathfrak{q},\overset{\star}{\overline{\mathbf{F}}}{}^{\mathrm{int}},\boldsymbol{\pi},t_f]\overset{\mathrm{def}}{=}\int_{t_i}^{t_f}\left[\int_{\overline{S}}\dot{\mathfrak{q}}^{\mathrm{T}}\,\boldsymbol{\pi}\,d\overline{S}+\oint_{\overline{C}}\dot{\mathfrak{q}}^{\mathrm{T}}\,\boldsymbol{\pi}\,d\overline{s}_\ell-\overline{\mathscr{H}}_{cc}[\mathfrak{q},\overline{\mathbf{F}}^{\mathrm{int}},\boldsymbol{\pi}]\right]d\hat{t}\tag{85b}$$

with

$$\begin{aligned}\overline{\mathscr{H}}_{cc}[\mathfrak{q},\overline{\mathbf{F}}^{\mathrm{int}},\boldsymbol{\pi}]&\overset{\mathrm{def}}{=}\overline{\mathscr{K}}_c[\boldsymbol{\pi}]+\overline{\mathscr{E}}_{cc}^{\mathrm{elas}}[\mathfrak{q},\overline{\mathbf{F}}^{\mathrm{int}}]\\&\equiv\overline{\mathscr{K}}_c[\boldsymbol{\pi}]-\int_{\overline{S}}\mathfrak{q}^{\mathrm{T}}\,\overline{\mathbb{D}}\,\overline{\mathbf{F}}^{\mathrm{int}}\,d\overline{S}+\oint_{\overline{C}}\mathfrak{q}^{\mathrm{T}}\,\overline{\mathbf{F}}^{\mathrm{int}}\,d\overline{s}_\ell-\overline{\mathscr{E}}_c^{\mathrm{elas}}[\overline{\mathbf{F}}^{\mathrm{int}}]\\&\equiv\overline{\mathscr{K}}_c[\boldsymbol{\pi}]+\int_{\overline{S}}\overline{\mathbf{E}}^{\mathrm{T}}\,\overline{\mathbf{F}}^{\mathrm{int}}\,d\overline{S}-\overline{\mathscr{E}}_c^{\mathrm{elas}}[\overline{\mathbf{F}}^{\mathrm{int}}]\ \mathrm{while}\ \overline{\mathbf{E}}[\mathfrak{q}]\equiv\overline{\mathbb{B}}\,\mathfrak{q}\,.\end{aligned}\tag{85c}$$

Inspired by thermodynamics with the Helmholtz *thermal displacement* [110], which is also sometimes called *thermacy* in dissipation-less thermoelasticity [111] (see also [112–114]),

we considered, as in [104], the impulse $\overset{\star}{\overline{\mathbf{F}}}^{\text{int}}$ of the generalized internal forces $\overline{\mathbf{F}}^{\text{int}} \equiv \dfrac{\partial \overset{\star}{\overline{\mathbf{F}}}^{\text{int}}}{\partial t}$ as an additional primal variable of variations. The impulse is defined up to an additive temporal constant according to:

$$\overset{\star}{f}(\cdot, t_i, t) \overset{\text{def}}{=} \overset{\star}{f}(\cdot, t_i, t_i) + \int_{t_i}^{t} f(\cdot, \hat{t}) d\hat{t}, \text{ for } t \in T \overset{\text{def}}{=} [t_i, t_f]. \tag{86}$$

Such a variational formulation allows pinpointing (in a systematic way and from an original energy variational viewpoint) the different contributions of the various energies in the PHS formulation with as a necessary condition of balance (while reminding notably about Equations (29), (40), and (43) for $t \in T \setminus \partial T$):

$$\dot{\mathfrak{q}}_{/\overline{\mathcal{S}}} = \delta_{\boldsymbol{\pi}} \overline{\mathcal{H}}_{cc/\overline{\mathcal{S}}}[\mathfrak{q}, \overline{\mathbf{F}}^{\text{int}}, \boldsymbol{\pi}], \text{ from } \boldsymbol{\pi} \text{ on } \overline{\mathcal{S}} \tag{87a}$$

$$\dot{\boldsymbol{\pi}}_{/\overline{\mathcal{S}}}[\dot{\mathfrak{q}}] = -\delta_{\mathfrak{q}} \overline{\mathcal{H}}_{cc/\overline{\mathcal{S}}}[\mathfrak{q}, \overline{\mathbf{F}}^{\text{int}}, \boldsymbol{\pi}] + \mathcal{Q}^{\text{ext}}_{\mathfrak{q}/\overline{\mathcal{S}}}[\mathfrak{q}]$$

$$\equiv \overline{\mathbb{D}} \overline{\mathbf{F}}^{\text{int}} + \mathcal{Q}^{\text{ext}}_{\mathfrak{q}/\overline{\mathcal{S}}}[\mathfrak{q}] \equiv \mathcal{Q}^{\text{int}}_{\mathfrak{q}/\overline{\mathcal{S}}}[\mathfrak{q}] + \mathcal{Q}^{\text{ext}}_{\mathfrak{q}/\overline{\mathcal{S}}}[\mathfrak{q}] \equiv \mathcal{Q}^{\text{acc}}_{\mathfrak{q}}[\mathfrak{q}], \text{ from } \mathfrak{q} \text{ on } \overline{\mathcal{S}} \tag{87b}$$

$$\dot{\boldsymbol{\pi}}_{/\overline{\mathcal{C}}}[\dot{\mathfrak{q}}] = -\delta_{\mathfrak{q}} \overline{\mathcal{H}}_{cc/\overline{\mathcal{C}}}[\mathfrak{q}, \overline{\mathbf{F}}^{\text{int}}, \boldsymbol{\pi}] + \mathcal{Q}^{\text{ext}}_{\mathfrak{q}/\overline{\mathcal{C}}}[\mathfrak{q}] \equiv \mathcal{Q}^{\text{int}}_{\mathfrak{q}/\overline{\mathcal{C}}}[\mathfrak{q}] + \mathcal{Q}^{\text{ext}}_{\mathfrak{q}/\overline{\mathcal{C}}}[\mathfrak{q}], \text{ from } \mathfrak{q} \text{ on } \overline{\mathcal{C}} \tag{87c}$$

$$\dot{\overline{\mathbf{E}}} \equiv \overline{\mathbb{B}} \dot{\mathfrak{q}} = \frac{\partial}{\partial t} \delta_{\overline{\mathbf{F}}^{\text{int}}} \overline{\mathcal{E}}^{\text{elas}}_{c/\overline{\mathcal{S}}}[\overline{\mathbf{F}}^{\text{int}}] \equiv \frac{\partial}{\partial t} \delta_{\overline{\mathbf{F}}^{\text{int}}} \overline{\mathcal{H}}_{c/\overline{\mathcal{S}}}[\overline{\mathbf{F}}^{\text{int}}, \boldsymbol{\pi}] \equiv \overline{\mathbb{C}}^{-1} \overset{\star}{\overline{\mathbf{F}}}^{\text{int}}_{/\overline{\mathcal{S}}}, \text{ from } \overset{\star}{\overline{\mathbf{F}}}^{\text{int}} \text{ on } \overline{\mathcal{S}} \tag{87d}$$

$$\dot{\overline{\mathcal{H}}}_{cc}[\mathfrak{q}, \overline{\mathbf{F}}^{\text{int}}, \boldsymbol{\pi}] = \mathscr{P}^{\text{ext}}[\mathfrak{q}, \dot{\mathfrak{q}}], \text{ from } T \text{ with}$$

$$(\dot{\mathfrak{q}}, \overset{\star}{\overline{\mathbf{F}}}^{\text{int}}, \dot{\boldsymbol{\pi}}) \overset{\text{a.e.}}{\equiv} (\dot{\mathfrak{q}}, \overline{\mathbf{F}}^{\text{int}}, \dot{\boldsymbol{\pi}}) \text{ on } (\overline{\mathcal{S}} \cup \overline{\mathcal{C}}) \times (T \setminus \partial T). \tag{87e}$$

In particular, this set of necessary conditions of stationarity includes, again, on the fifth lines in Equation (87), the condition of integrability related to Equations (23) and (79).

It is worth mentioning that some works, such as [91,92,115], also considered the Hellinger–Prange–Reissner variational functional in the variational principle with the following action integral, where $\overline{\mathbf{F}}^{\text{int}}$ is the additional primal variable of variations instead of $\overset{\star}{\overline{\mathbf{F}}}^{\text{int}}$:

$$\overline{\mathscr{A}}_{\text{HR}}[\mathfrak{q}, \overline{\mathbf{F}}^{\text{int}}, \boldsymbol{\pi}, t_f] \sim \overline{\mathscr{A}}_{\text{HRPH}}[\mathfrak{q}, \overset{\star}{\overline{\mathbf{F}}}^{\text{int}}, \boldsymbol{\pi}, t_f], \tag{88}$$

{(the symbol $\sim$ implying that, explicitly, their respective expression is literally the same). As a result, this choice of primary field variables does not yield the condition of integrability in Equation (79) as a variationally derived constraint.

At this point, we are ready to develop the PHS formulation of the NR shell model(s), the objective of which is to simultaneously satisfy the dynamic equations, as well as as the kinetic energy theorems in Equations (80) and (83). To proceed further towards our PHS formulation, let us first couch, for convenience, the following integers that represent, respectively, the number of lines in $\overline{\mathbf{E}}$ and the numbers of lines and columns of $\overline{\mathbb{B}}$:

$$\mathfrak{m} \overset{\text{def}}{=} \#\text{line}(\overline{\mathbb{B}}) = 10, \quad \mathfrak{n} \overset{\text{def}}{=} \#\text{column}(\overline{\mathbb{B}}) = 5. \tag{89}$$

We distinguish then the ensuing PHS for the two ports of the NR (five parameters) 2D shell model.

*The 2D shell bulk port.* Over $\overline{\mathcal{S}}$, a PHS distributed state $(\mathfrak{f}_{(\mathfrak{m}+\mathfrak{n})\times 1}, \mathfrak{e}_{(\mathfrak{m}+\mathfrak{n})\times 1})$ involves a PHS distributed effort $\mathfrak{e}_{(\mathfrak{m}+\mathfrak{n})\times 1}$, and the related PHS flow $\mathfrak{f}_{(\mathfrak{m}+\mathfrak{n})\times 1}$ of the NR shell model can be chosen then (while keeping in mind the identifications in Equation (84)) over $\overline{\mathcal{S}}$ as

$$\mathfrak{e}_{(m+n)\times 1} \overset{\text{def}}{=} \left\{ \begin{matrix} \dot{\mathfrak{q}}_{/\overline{\mathcal{S}}} \\ \overline{\mathbf{F}}^{\text{int}}_{/\overline{\mathcal{S}}} \end{matrix} \right\} \equiv \left\{ \begin{matrix} \delta_{\pi}\overline{\mathscr{H}}_c \\ \delta_{\overline{\mathbf{E}}}\overline{\mathscr{H}} \end{matrix} \right\} \tag{90a}$$

$$-\mathfrak{f}^{\text{int}}_{(m+n)\times 1} \overset{\text{def}}{=} \left\{ \begin{matrix} \overline{\mathcal{Q}}^{\text{int}}_{\mathfrak{q}/\overline{\mathcal{S}}} \\ \dot{\overline{\mathbf{E}}} \end{matrix} \right\} \equiv \left\{ \begin{matrix} \delta_{\mathfrak{q}}\overline{\mathscr{H}} \\ \frac{\partial}{\partial t}\delta_{\overline{\mathbf{F}}^{\text{int}}}\overline{\mathscr{H}}_c \end{matrix} \right\}, \text{ with } \dot{\overline{\mathbf{E}}} \equiv \overline{\mathbf{E}}[\dot{\mathfrak{q}}] \equiv \overline{\mathbb{B}}\,\dot{\mathfrak{q}}. \tag{90b}$$

Then, by following variational formulations like the foregoing ones and in agreement with Equations (23), (43), (43) and (87), the suitable PHS state variables arise related as:

$$\overset{-\mathfrak{f}^{\text{int}}_{(m+n)\times 1}}{\left\{ \begin{matrix} \mathcal{Q}^{\text{int}}_{\mathfrak{q}/\overline{\mathcal{S}}} \\ \dot{\overline{\mathbf{E}}} \end{matrix} \right\}} = \overset{\mathcal{J}_{(m+n)\times(m+n)}}{\begin{pmatrix} \mathbf{0}_{n\times n} & \overline{\mathbb{D}} \\ \overline{\mathbb{B}} & \mathbf{0}_{m\times m} \end{pmatrix}} \overset{\mathfrak{e}_{(m+n)\times 1}}{\left\{ \begin{matrix} \dot{\mathfrak{q}}_{/\overline{\mathcal{S}}} \\ \overline{\mathbf{F}}^{\text{int}}_{/\overline{\mathcal{S}}} \end{matrix} \right\}} \text{ over } \overline{\mathcal{S}} \tag{91}$$

$$\overset{\mathfrak{f}^{\text{acc}}_{(m+n)\times 1}}{\left\{ \begin{matrix} \dot{\boldsymbol{\pi}}_{/\overline{\mathcal{S}}} \\ \dot{\overline{\mathbf{E}}} \end{matrix} \right\}} = \overset{\mathcal{J}_{(m+n)\times(m+n)}}{\begin{pmatrix} \mathbf{0}_{n\times n} & \overline{\mathbb{D}} \\ \overline{\mathbb{B}} & \mathbf{0}_{m\times m} \end{pmatrix}} \overset{\mathfrak{e}_{(m+n)\times 1}}{\left\{ \begin{matrix} \dot{\mathfrak{q}}_{/\overline{\mathcal{S}}} \\ \overline{\mathbf{F}}^{\text{int}}_{/\overline{\mathcal{S}}} \end{matrix} \right\}} + \overset{\mathfrak{f}^{\text{ext}}_{(m+n)\times 1}}{\left\{ \begin{matrix} \mathcal{Q}^{\text{ext}}_{\mathfrak{q}/\overline{\mathcal{S}}} \\ \mathbf{0}_{m\times 1} \end{matrix} \right\}} \text{ over } \overline{\mathcal{S}}. \tag{92}$$

(Here, each colored entity on top of a matrix represents the matrix itself. These entities are not explicitly introduced to save space. The conventional minus sign is necessary in order to have consistency in the power flow from the shell.). The differential operator matrix $\mathcal{J}_{(m+n)\times(m+n)}(\bar{\boldsymbol{\alpha}})$ represents the PHS formally skew-symmetric differential operator in the sense specified in [8,116]. The PHS state variables associated with the "*shell-ports*" are $(\mathfrak{f}_{(m+n)\times 1}, \mathfrak{e}_{(m+n)\times 1}) = (\mathfrak{f}^{\text{acc}}_{(m+n)\times 1} + \mathfrak{f}^{\text{int}}_{(m+n)\times 1}, \mathfrak{e}_{(m+n)\times 1})$ and are interconnected with the energy storage $\overline{\mathscr{H}}_c \equiv \overline{\mathscr{H}}$ (i.e., the Hamiltonian function) of the shell system.

We can then identify the following "interior product" results (i.e., the power associated with the port) for the shell system with the power of the effective external efforts exerted on the 2D shell:

$$\begin{aligned} -\int_{\overline{\mathcal{S}}} \mathfrak{e}^{\text{T}}_{(m+n)\times 1}\,\mathfrak{f}^{\text{int}}_{(m+n)\times 1}d\overline{\mathcal{S}} &\equiv \int_{\overline{\mathcal{S}}}\left[\dot{\mathfrak{q}}^{\text{T}}\,\mathcal{Q}^{\text{int}}_{\mathfrak{q}} + \dot{\overline{\mathbf{E}}}^{\text{T}}\overline{\mathbf{F}}^{\text{int}}\right]d\overline{\mathcal{S}} \\ &\equiv -\oint_{\overline{\mathcal{C}}}\dot{\mathfrak{q}}^{\text{T}}\,\mathcal{Q}^{\text{int}}_{\mathfrak{q}}d\bar{s}_{\ell} \equiv -\overline{\mathscr{P}}^{\text{int}}_{/\overline{\mathcal{C}}}[\mathfrak{q},\dot{\mathfrak{q}}] \\ &\equiv \oint_{\overline{\mathcal{C}}}\dot{\mathfrak{q}}^{\text{T}}\left[\mathcal{Q}^{\text{ext}}_{\mathfrak{q}} - \dot{\boldsymbol{\pi}}_{/\overline{\mathcal{C}}}\right]d\bar{s}_{\ell} \\ &\equiv \overline{\mathscr{P}}^{\text{ext}}_{/\overline{\mathcal{C}}}[\mathfrak{q},\dot{\mathfrak{q}}] - \overline{\mathscr{P}}^{\text{acc}}_{/\overline{\mathcal{C}}}[\mathfrak{q},\dot{\mathfrak{q}}] \overset{\text{NR}}{=} \overline{\mathscr{P}}^{\text{ext}}_{/\overline{\mathcal{C}}}[\mathfrak{q},\dot{\mathfrak{q}}] \end{aligned} \tag{93}$$

as defined according to Equations (28) and (51), by integrating by parts the second term, while straightforwardly for the effective external efforts exerted on the 2D shell bulk domain $\overline{\mathcal{S}}$:

$$\int_{\overline{\mathcal{S}}} \mathfrak{e}^{\text{T}}_{(m+n)\times 1}\,\mathfrak{f}^{\text{ext}}_{(m+n)\times 1}d\overline{\mathcal{S}} \equiv \int_{\overline{\mathcal{S}}}\dot{\mathfrak{q}}^{\text{T}}\,\mathcal{Q}^{\text{ext}}_{\mathfrak{q}}d\overline{\mathcal{S}} \equiv \overline{\mathscr{P}}^{\text{ext}}_{/\overline{\mathcal{S}}}[\mathfrak{q},\dot{\mathfrak{q}}]. \tag{94}$$

Similarly, we have from the left-hand side of Equation (92) (and according to Equations (51) and (68)),

$$\begin{aligned} \int_{\overline{\mathcal{S}}} \mathfrak{e}^{\text{T}}_{(m+n)\times 1}\,\mathfrak{f}^{\text{acc}}_{(m+n)\times 1}d\overline{\mathcal{S}} &\equiv \int_{\overline{\mathcal{S}}}\left[\dot{\mathfrak{q}}^{\text{T}}\,\dot{\boldsymbol{\pi}}^{\text{T}} + \dot{\overline{\mathbf{E}}}^{\text{T}}\overline{\mathbf{F}}^{\text{int}}\right]d\overline{\mathcal{S}} \equiv \overline{\mathscr{P}}^{\text{acc}}_{/\overline{\mathcal{S}}}[\mathfrak{q},\dot{\mathfrak{q}}] - \overline{\mathscr{P}}^{\text{int}}[\mathfrak{q},\dot{\mathfrak{q}}] \\ &\equiv \overline{\mathscr{P}}^{\text{ext}}[\mathfrak{q},\dot{\mathfrak{q}}] - \overline{\mathscr{P}}^{\text{acc}}_{/\overline{\mathcal{C}}}[\mathfrak{q},\dot{\mathfrak{q}}] \overset{\text{NR}}{=} \overline{\mathscr{P}}^{\text{ext}}[\mathfrak{q},\dot{\mathfrak{q}}] \end{aligned} \tag{95a}$$

(the last simplification being tied to the NR shell modeling), while from the right-hand side of Equation (92) (and according to Equations (31), (51), and (60)),

$$
\begin{aligned}
\int_{\overline{\mathcal{S}}} \mathfrak{e}_{(\mathfrak{m}+\mathfrak{n})\times 1}^{\mathrm{T}} \left( \mathfrak{f}_{(\mathfrak{m}+\mathfrak{n})\times 1}^{\mathrm{ext}} - \mathfrak{f}_{(\mathfrak{m}+\mathfrak{n})\times 1}^{\mathrm{int}} \right) d\overline{\mathcal{S}} \;\; &\equiv \;\; \int_{\overline{\mathcal{S}}} \left[ \dot{\mathsf{q}}^{\mathrm{T}} \mathcal{Q}_{\mathsf{q}}^{\mathrm{ext}} + \dot{\mathsf{q}}^{\mathrm{T}} \, \overline{\mathbb{D}} \, \overline{\mathbf{F}}^{\mathrm{int}} + \dot{\overline{\mathbf{E}}}^{\mathrm{T}} \overline{\mathbf{F}}^{\mathrm{int}} \right] d\overline{\mathcal{S}} \\[4pt]
&\equiv \;\; \overline{\mathscr{P}}_{/\mathcal{S}}^{\mathrm{ext}}[\mathsf{q}, \dot{\mathsf{q}}] + \int_{\overline{\mathcal{S}}} \left[ \dot{\mathsf{q}}^{\mathrm{T}} \mathcal{Q}_{\mathsf{q}}^{\mathrm{int}} + \dot{\overline{\mathbf{E}}}^{\mathrm{T}} \overline{\mathbf{F}}^{\mathrm{int}} \right] d\overline{\mathcal{S}} \\[4pt]
&\equiv \;\; \overline{\mathscr{P}}_{/\mathcal{S}}^{\mathrm{ext}}[\mathsf{q}, \dot{\mathsf{q}}] - \oint_{\overline{\mathcal{C}}} \dot{\mathsf{q}}^{\mathrm{T}} \mathcal{Q}_{\mathsf{q}}^{\mathrm{int}} \, d\bar{s}_\ell \\[4pt]
&\equiv \;\; \overline{\mathscr{P}}_{/\mathcal{S}}^{\mathrm{ext}}[\mathsf{q}, \dot{\mathsf{q}}] + \oint_{\overline{\mathcal{C}}} \dot{\mathsf{q}}^{\mathrm{T}} \left[ \mathcal{Q}_{\mathsf{q}}^{\mathrm{ext}} - \dot{\boldsymbol{\pi}}_{/\overline{\mathcal{C}}} \right] d\bar{s}_\ell \\[4pt]
&\equiv \;\; \overline{\mathscr{P}}_{/\mathcal{S}}^{\mathrm{ext}}[\mathsf{q}, \dot{\mathsf{q}}] + \overline{\mathscr{P}}_{/\overline{\mathcal{C}}}^{\mathrm{ext}}[\mathsf{q}, \dot{\mathsf{q}}] - \overline{\mathscr{P}}_{/\overline{\mathcal{C}}}^{\mathrm{acc}}[\mathsf{q}, \dot{\mathsf{q}}] \\[4pt]
&\equiv \;\; \overline{\mathscr{P}}^{\mathrm{ext}}[\mathsf{q}, \dot{\mathsf{q}}] - \overline{\mathscr{P}}_{/\overline{\mathcal{C}}}^{\mathrm{acc}}[\mathsf{q}, \dot{\mathsf{q}}] \;\overset{\mathrm{NR}}{=}\; \overline{\mathscr{P}}^{\mathrm{ext}}[\mathsf{q}, \dot{\mathsf{q}}] . \qquad \text{(95b)}
\end{aligned}
$$

*The 2D shell boundary port.* It is usual as well to define PHS state variables (i.e., boundary flow and effort or, again, co-energy variables) on the boundary $\overline{\mathcal{C}}$. Over $\overline{\mathcal{C}}$, a PHS distributed state $(\mathfrak{f}_{2\mathfrak{n}\times 1}, \mathfrak{e}_{2\mathfrak{n}\times 1})$ involves a PHS distributed effort $\mathfrak{e}_{2\mathfrak{n}\times 1}$, and the related PHS flow $\mathfrak{f}_{2\mathfrak{n}\times 1}$ of the NR shell model can be chosen then as

$$
\mathfrak{e}_{2\mathfrak{n}\times 1} \;\overset{\mathrm{def}}{=}\; \left\{ \begin{matrix} \dot{\mathsf{q}}_{/\overline{\mathcal{C}}} \\ \overline{\mathbf{F}}_{/\overline{\mathcal{C}}}^{\mathrm{int}} \end{matrix} \right\} \;\not\equiv\; \left\{ \begin{matrix} \delta_{\boldsymbol{\pi}_{/\overline{\mathcal{C}}}} \overline{\mathscr{H}}_c \\ \delta_{\mathsf{q}_{/\overline{\mathcal{C}}}} \overline{\mathscr{H}} \end{matrix} \right\} \quad \text{and} \quad -\mathfrak{f}_{2\mathfrak{n}\times 1}^{\mathrm{int}} \;\overset{\mathrm{def}}{=}\; \left\{ \begin{matrix} \overline{\mathcal{Q}}_{\mathsf{q}/\overline{\mathcal{C}}}^{\mathrm{int}} \\ \dot{\mathsf{q}}_{/\overline{\mathcal{C}}} \end{matrix} \right\} \;\not\equiv\; \left\{ \begin{matrix} \delta_{\mathsf{q}_{/\overline{\mathcal{C}}}} \overline{\mathscr{H}} \\ \frac{\partial}{\partial t} \delta_{\overline{\mathbf{F}}_{/\overline{\mathcal{C}}}^{\mathrm{int}}} \overline{\mathscr{H}}_c \end{matrix} \right\} \qquad \text{(96)}
$$

(whereas $\overline{\mathbf{F}}_{/\overline{\mathcal{C}}}^{\mathrm{int}} \equiv \delta_{\mathsf{q}_{/\overline{\mathcal{C}}}} \overline{\mathscr{H}}$ and $\overline{\mathcal{Q}}_{\mathsf{q}/\overline{\mathcal{C}}}^{\mathrm{int}} \equiv \delta_{\mathsf{q}_{/\overline{\mathcal{C}}}} \overline{\mathscr{H}}$). Therefore, by following variational formulations like the foregoing ones and in agreement with Equations (24), (44), and (87), the PHS state variables are related as

$$
\underset{\color{magenta}{-\mathfrak{f}_{2\mathfrak{n}\times 1}^{\mathrm{int}}}}{\left\{ \begin{matrix} \mathcal{Q}_{\mathsf{q}/\overline{\mathcal{C}}}^{\mathrm{int}} \\ \dot{\mathsf{q}}_{/\overline{\mathcal{C}}} \end{matrix} \right\}} \;=\; \underset{\color{magenta}{\mathcal{J}_{2\mathfrak{n}\times 2\mathfrak{n}}}}{\begin{pmatrix} \mathbf{0}_{\mathfrak{n}\times\mathfrak{n}} & -\mathbb{I}_{\mathfrak{n}\times\mathfrak{n}} \\ \mathbb{I}_{\mathfrak{n}\times\mathfrak{n}} & \mathbf{0}_{\mathfrak{n}\times\mathfrak{n}} \end{pmatrix}} \;\underset{\color{magenta}{\mathfrak{e}_{2\mathfrak{n}\times 1}}}{\left\{ \begin{matrix} \dot{\mathsf{q}}_{/\overline{\mathcal{C}}} \\ \overline{\mathbf{F}}_{/\overline{\mathcal{C}}}^{\mathrm{int}} \end{matrix} \right\}} \quad \text{over } \overline{\mathcal{C}} \qquad \text{(97)}
$$

$$
\underset{\color{magenta}{\mathfrak{f}_{2\mathfrak{n}\times 1}^{\mathrm{acc}}}}{\left\{ \begin{matrix} \dot{\boldsymbol{\pi}}_{/\overline{\mathcal{C}}} \\ \dot{\mathsf{q}}_{/\overline{\mathcal{C}}} \end{matrix} \right\}} \;=\; \underset{\color{magenta}{\mathcal{J}_{2\mathfrak{n}\times 2\mathfrak{n}}}}{\begin{pmatrix} \mathbf{0}_{\mathfrak{n}\times\mathfrak{n}} & -\mathbb{I}_{\mathfrak{n}\times\mathfrak{n}} \\ \mathbb{I}_{\mathfrak{n}\times\mathfrak{n}} & \mathbf{0}_{\mathfrak{n}\times\mathfrak{n}} \end{pmatrix}} \;\underset{\color{magenta}{\mathfrak{e}_{2\mathfrak{n}\times 1}}}{\left\{ \begin{matrix} \dot{\mathsf{q}}_{/\overline{\mathcal{C}}} \\ \overline{\mathbf{F}}_{/\overline{\mathcal{C}}}^{\mathrm{int}} \end{matrix} \right\}} \;+\; \underset{\color{magenta}{\mathfrak{f}_{2\mathfrak{n}\times 1}^{\mathrm{ext}}}}{\left\{ \begin{matrix} \mathcal{Q}_{\mathsf{q}/\overline{\mathcal{C}}}^{\mathrm{ext}} \\ \mathbf{0}_{\mathfrak{n}\times 1} \end{matrix} \right\}} \quad \text{over } \overline{\mathcal{C}}. \qquad \text{(98)}
$$

This expression for the NR model involves the identity matrix $\mathbb{I}_{\mathfrak{n}\times\mathfrak{n}}$ of rank $\mathfrak{n}$. Thus, $\mathcal{J}_{2\mathfrak{n}\times 2\mathfrak{n}}$ turns over $\overline{\mathcal{C}}$ like the classical (unit) constant symplectic matrix [68] of the usual canonical Hamiltonian systems. As a result,

$$
\oint_{\overline{\mathcal{C}}} \mathfrak{e}_{2\mathfrak{n}\times 1}^{\mathrm{T}} \mathfrak{f}_{2\mathfrak{n}\times 1}^{\mathrm{ext}} \, d\bar{s}_\ell \equiv \oint_{\overline{\mathcal{C}}} \dot{\mathsf{q}}^{\mathrm{T}} \mathcal{Q}_{\mathsf{q}}^{\mathrm{ext}} d\bar{s}_\ell \equiv \overline{\mathscr{P}}_{/\overline{\mathcal{C}}}^{\mathrm{ext}}[\mathsf{q}, \dot{\mathsf{q}}] \qquad \text{(99)}
$$

$$
\begin{aligned}
\oint_{\overline{\mathcal{C}}} \mathfrak{e}_{2\mathfrak{n}\times 1}^{\mathrm{T}} \mathfrak{f}_{2\mathfrak{n}\times 1}^{\mathrm{acc}} \, d\bar{s}_\ell \;\; &\equiv \;\; \oint_{\overline{\mathcal{C}}} \left\{ \dot{\mathsf{q}}^{\mathrm{T}} \dot{\boldsymbol{\pi}}_{/\overline{\mathcal{C}}} + \left[ \overline{\mathbf{F}}_{/\overline{\mathcal{C}}}^{\mathrm{int}} \right]^{\mathrm{T}} \dot{\mathsf{q}} \right\} d\bar{s}_\ell \equiv \oint_{\overline{\mathcal{C}}} \dot{\mathsf{q}}_{/\overline{\mathcal{C}}}^{\mathrm{T}} \left[ \mathcal{Q}_{\mathsf{q}}^{\mathrm{acc}} - \mathcal{Q}_{\mathsf{q}}^{\mathrm{int}} \right] d\bar{s}_\ell \\[4pt]
&\equiv \;\; \overline{\mathscr{P}}_{/\overline{\mathcal{C}}}^{\mathrm{acc}}[\mathsf{q}, \dot{\mathsf{q}}] - \overline{\mathscr{P}}_{/\overline{\mathcal{C}}}^{\mathrm{int}}[\mathsf{q}, \dot{\mathsf{q}}] = -\overline{\mathscr{P}}_{/\overline{\mathcal{C}}}^{\mathrm{int}}[\mathsf{q}, \dot{\mathsf{q}}] \qquad \text{(100)}
\end{aligned}
$$

(in which $\overline{\mathscr{P}}_{/\overline{\mathcal{C}}}^{\mathrm{int}}[\mathsf{q}, \dot{\mathsf{q}}] \equiv 0$ for the simple NR shell model), so that

$$\oint_{\overline{C}} \mathfrak{e}_{2n\times1}^{T} \mathfrak{f}_{2n\times1}^{int} \, d\bar{s}_{\ell} \equiv -\oint_{\overline{C}} \left\{ \dot{\mathfrak{q}}^{T} \mathcal{Q}_{\mathfrak{q}}^{int} + \left[ \overline{\mathbf{F}}_{/\overline{C}}^{int} \right]^{T} \dot{\mathfrak{q}} \right\} d\bar{s}_{\ell}$$

$$\equiv \oint_{\overline{C}} \mathfrak{e}_{2n\times1}^{T} \left[ \mathfrak{f}_{2n\times1}^{ext} - \mathfrak{f}_{2n\times1}^{acc} \right] d\bar{s}_{\ell}$$

$$\equiv \overline{\mathscr{P}}_{/\overline{C}}^{ext}[\mathfrak{q},\dot{\mathfrak{q}}] + \overline{\mathscr{P}}_{/\overline{C}}^{int}[\mathfrak{q},\dot{\mathfrak{q}}] - \overline{\mathscr{P}}_{/\overline{C}}^{acc}[\mathfrak{q},\dot{\mathfrak{q}}]$$

$$= \overline{\mathscr{P}}_{/\overline{C}}^{ext}[\mathfrak{q},\dot{\mathfrak{q}}] + \overline{\mathscr{P}}_{/\overline{C}}^{int}[\mathfrak{q},\dot{\mathfrak{q}}] . \tag{101}$$

Finally, according to the PHS expectations,

$$\int_{\overline{S}} \mathfrak{e}_{(m+n)\times1}^{T} \left[ \mathfrak{f}_{(m+n)\times1}^{acc} + \mathfrak{f}_{(m+n)\times1}^{int} \right] d\overline{S} \equiv \overline{\mathscr{P}}^{ext}[\mathfrak{q},\dot{\mathfrak{q}}] - \overline{\mathscr{P}}_{/\overline{C}}^{ext}[\mathfrak{q},\dot{\mathfrak{q}}] \equiv \overline{\mathscr{P}}_{/\overline{S}}^{ext}[\mathfrak{q},\dot{\mathfrak{q}}] \tag{102a}$$

$$\oint_{\overline{C}} \mathfrak{e}_{2n\times1}^{T} \left[ \mathfrak{f}_{2n\times1}^{acc} + \mathfrak{f}_{2n\times1}^{int} \right] d\bar{s}_{\ell} \equiv \overline{\mathscr{P}}_{/\overline{C}}^{ext}[\mathfrak{q},\dot{\mathfrak{q}}] \tag{102b}$$

and so, the sum of Equation (102) yields the full (kinetic) energy conservation theorem:

$$\int_{\overline{S}} \mathfrak{e}_{(m+n)\times1}^{T} \left[ \mathfrak{f}_{(m+n)\times1}^{acc} + \mathfrak{f}_{(m+n)\times1}^{int} \right] d\overline{S} + \oint_{\overline{C}} \mathfrak{e}_{2n\times1}^{T} \left[ \mathfrak{f}_{2n\times1}^{acc} + \mathfrak{f}_{2n\times1}^{int} \right] d\bar{s}_{\ell} \equiv \overline{\mathscr{P}}^{ext}[\mathfrak{q},\dot{\mathfrak{q}}] . \tag{102c}$$

### 4.2. Some Remarks on These Variational Principles and Their Numerical Treatments

For completeness, we comment below on some of the salient points of these equivalent variational formulations. Then, we provide some references on their numerical processing.

#### 4.2.1. Advantages and Limitations of the Foregoing Variational Formalisms

**Limitations**. The multi-primary field principles presented previously are valid beyond the framework of the classical theory of linear elastodynamics under some conditions. In particular, to express the complementary (elastic and kinetic) energy potential densities, it is necessary that the *momentum velocity* constitutive law and *stress–strain* constitutive law can be inverted explicitly (even by using the appropriate implicit Legendre–Fenchel transforms in Equations (56) and 72)). When this requirement fails, the foregoing variational formalism can employ as well more multi-primary fields as in the (Chien–)Hu–Washizu (**HW**) and Hamilton-Pontryagin variational formalisms [65,88,89,97,104,117–125]. In such a case, the knowledge gained from the last development and the cited works suggests that the appropriate integral of action would depend on the generalized internal force field $\overline{\mathbf{F}}^{int}$ (or its impulse $\overset{\star}{\overline{\mathbf{F}}}{}^{int}$), the generalized strain field $\overline{\mathbf{E}}$, the displacement field $\mathfrak{q}$, the velocity field $v \sim \dot{\mathfrak{q}}$, and the generalized momentum field $\boldsymbol{\pi}$. Thus, in order to obtain the canonical equations in Equation (76) from the integral of action of this new multi-field formulation, the generalized internal force field $\overline{\mathbf{F}}^{int}$ and the generalized momentum $\boldsymbol{\pi}$ must enter like Lagrange's multipliers [119,120,122]; the first field enforces, therefore, the strain–displacement relation in Equation (23), while the second field enforces the velocity–displacement relation $\mathfrak{q} = \overset{\star}{v}$ (defined in accordance with Equation (86)), wherein the displacement field $\mathfrak{q}$ is considered as the impulse of the velocity field $v = \dot{\mathfrak{q}}$. This yields, then, for the integral of action of the Hamilton–Pontryagin–Hu–Washizu type,

$$\overline{\mathscr{A}}_{HPHW^{*}}[\mathfrak{q}, \overset{\star}{\overline{\mathbf{F}}}{}^{int}, \boldsymbol{\pi}, v, \overline{\mathbf{E}}, t_{f}] \overset{def}{=} \int_{t_{i}}^{t_{f}} \left\{ \mathscr{L}[\overset{\star}{v}, \overline{\mathbf{E}}, v] \right.$$

$$+ \int_{\overline{S}} \left[ (\dot{\mathfrak{q}} - v)^{T} \boldsymbol{\pi} - (\overline{\mathbb{B}}\mathfrak{q} - \overline{\mathbf{E}})^{T} \overline{\mathbf{F}}^{int} \right] d\overline{S} \tag{103}$$

$$\left. + \oint_{\overline{C}} \left[ (\dot{\mathfrak{q}} - v)^{T} \boldsymbol{\pi} \right] d\bar{s}_{\ell} \right\} d\hat{t}$$

or $\overline{\mathscr{A}}_{\text{HPHW}}[\mathsf{q}, \overline{\mathbf{F}}^{\text{int}}, \boldsymbol{\pi}, v, \overline{\mathbf{E}}, t_f] \sim \overline{\mathscr{A}}_{\text{HPWH}^*}[\mathsf{q}, \overset{\star}{\overline{\mathbf{F}}}^{\text{int}}, \boldsymbol{\pi}, v, \overline{\mathbf{E}}, t_f]$ (with the adequate variational conditions at $\{t_i, t_f\}$). These integrals of action involve the Lagrangian $\overline{\mathscr{L}}[\overset{\star}{v}, \overline{\mathbf{E}}, v] \overset{\text{def}}{=} \overline{\mathscr{K}}[\overset{\star}{v}, v] - \overline{\mathscr{E}}^{\text{elas}}[\overset{\star}{v}, \overline{\mathbf{E}}]$ introduced in Equation (74); as before, the potential energy $\overline{\mathscr{E}}^{\text{elas}}[\overset{\star}{v}, \overline{\mathbf{E}}]$ and the kinetic energy $\overline{\mathscr{K}}[\overset{\star}{v}, v]$ are, respectively, defined according to Equations (51) and (68) in the linear elastodynamic case. Nevertheless, as mentioned in [117], this type of Hamilton–Pontryagin–Hu–Washizu integral of action is also suitable for nonlinear elastic theory with finite displacements. Besides, the expression of the integral of action can also be said to variationally follow the involutive process of the double-Fenchel–Legendre–Young transforms with respect to the space–time differential operators, yielding so the following non-classical Hamiltonian and Lagrangian functionals:

$$\overline{\mathscr{H}}_{\text{ncHPHW}}[\overset{\star}{v}, \overline{\mathbf{F}}^{\text{int}}, \boldsymbol{\pi}, v, \overline{\mathbf{E}}] \overset{\text{def}}{=} \int_{\overline{\mathcal{S}}} \left[ v^{\mathsf{T}} \boldsymbol{\pi} - \overline{\mathbf{E}}^{\mathsf{T}} \overline{\mathbf{F}}^{\text{int}} \right] d\overline{\mathcal{S}} + \oint_{\overline{\mathcal{C}}} \left[ v^{\mathsf{T}} \boldsymbol{\pi} \right] d\overline{s}_\ell - \overline{\mathscr{L}}[\overset{\star}{v}, \overline{\mathbf{E}}, v] \quad (104)$$

$$\begin{aligned} \overline{\mathscr{L}}_{\text{HPHW}}[\mathsf{q}, \overset{\star}{v}, \overline{\mathbf{F}}^{\text{int}}, \boldsymbol{\pi}, v, \overline{\mathbf{E}}] \overset{\text{def}}{=} \ & \int_{\overline{\mathcal{S}}} \left[ \dot{\mathsf{q}}^{\mathsf{T}} \boldsymbol{\pi} - \left( \overline{\mathbb{B}} \mathsf{q} \right)^{\mathsf{T}} \overline{\mathbf{F}}^{\text{int}} \right] d\overline{\mathcal{S}} \\ & + \oint_{\overline{\mathcal{C}}} \left[ \dot{\mathsf{q}}^{\mathsf{T}} \boldsymbol{\pi} \right] d\overline{s}_\ell - \overline{\mathscr{H}}_{\text{ncHPHW}}[\overset{\star}{v}, \overline{\mathbf{F}}^{\text{int}}, \boldsymbol{\pi}, v, \overline{\mathbf{E}}] \, . \end{aligned} \quad (105)$$

These functionals make it possible to write new canonical equations, including the aforementioned jump conditions and compatiblity ones, which will be explored in a different communication.

**Advantages**. All the previous variational principles are theoretically equivalent from a continuum viewpoint. However the Hamilton–(Hellinger–Prange–Reissner and Pontryagin–Hu–Washizu) multi-field variational principles formulated for the PHS formalism have certain advantages over the other ones from the numerical point of view. Notably, on the one hand, they involve several primary variables to be approximated simultaneously (for instance, by a mixed or hybrid finite-element method) [88,89,118,126,127] by enabling shifting the regularity assumptions between the involved functional spaces of the variables. On the other hand, such multi-field variational principles are known as well to improve the computational determination of the solution variables. Although the resulting saddle-point problems are more complicated to analyze (since the computational procedure needed to solve such a multifield formulation may require more calculations than that of the single field one), more robust numerical methods can be constructed for them. Moreover, a major incentive in developing such multi-primary field (as well as hybrid) variational formulations for shell analysis is to overcome the well-known shear-locking problem in the limit case of the thin shells in finite-element analyses. Works on linear elastodynamics that used quite similar Hamilton–(Hellinger–Prange–Reissner and/or Pontryagin–Hu–Washizu) functionals are, for instance, [51,88,89,91,92,104,127] or, again, [54,98,128] for a shell application.

### 4.2.2. Time-Differential versus Time-Variational GNI Treatments

**Strong time integration.** In many numerical strategies of resolution involving space–time variables, a spatial discretization (e.g., finite-element method) is customarily performed first to obtain a system of ordinary differential equations in the time variable. The classical Hamiltonian canonical equations in Equation (76) and the PHS canonical ones in Equation (87) are often solved following this time-differential numerical methodology.

This strategy does not require the use of the impulse $\overset{\star}{\overline{\mathbf{F}}}^{\text{int}}$ of the generalized internal forces $\overline{\mathbf{F}}^{\text{int}} \equiv \dfrac{\partial \overset{\star}{\overline{\mathbf{F}}}^{\text{int}}}{\partial t}$, but the latter itself. This classical partially variational strategy of the PHS formalism can be considered as a strong time-integration approach [91]. After discretization in space, the symplectic integration schemes (which are designed for numerical solutions of

canonical equations) can be used. Nevertheless, in order to benefit from some advantages of the mixed variational principles, the resolution must be naturally and systematically based on mixed finite elements for the discretization on the space domain at least, but not on the time domain.

This computational strategy was applied in [30] called the *partitioned finite-element method* (PFEM) in [129]. It was also applied to linear Reissner–Mindlin and Kirchhoff–Love plates in [14,15,130] (which can be considered as different natural limit cases of the NR shell model, the Kirchhoff–Love plate limit being numerically troublesome due to the occurrence of the *shear-locking phenomenon*). It has also been applied to the nonlinear shallow water equation in 2D [13] and extended to dissipative systems in [131]; the implementation details can be found in [132], and the numerical analysis of the convergence of the PFEM was presented in [133].

**Partial- versus full-variational, weak time integrations.** Besides, discrete-time versions of the foregoing Hamiltonian-like variational principles, i.e., which approximate the integrals of action in Equations (74), (77) and (85), are also of great interest. In general, these weak discrete approaches follow consistently the chosen variational principle.

Thus, the works in [46–48,134] (see also [135]) developed integrators that are based on a discretization of the Lagrange–d'Alembert principle (with action integrals like $\overline{\mathscr{A}}_{\mathrm{L}}[\mathfrak{q}, t_f]$ in Equation (74)), as well as on a variational formulation of dissipation. They mentioned satisfactory results. As illustrated by the numerical experiments in [39,46,136], by adopting a space–time view of variational integrators of the integrals of action, one can have integrators that preserve the energy, momentum, and symplectic structure.

Illustrating the problems of linear elastodynamics that are solved with such a time-variational strategy, multi-field variational principles in which displacement, velocity, and momentum fields are taken to be independent (with action integrals like $\overline{\mathscr{A}}_{\mathrm{H}}[\mathfrak{q}, \pi, t_f]$ in Equation (77)) are, for instance, in [91,92,119] and treated with space–time finite elements in [137].

An analogous discrete methodology was also developed for the other variational formulations with similar success, while using their primary fields (including the time variable). For instance, the work on linear shell elastodynamics in [91] (see also [92] for beam elastodynamics and [138] for (2D) plane elastodynamics/elasticity) provided such a weak time-variational integration strategy both for the extended Hamilton–(Hellinger–Prange–Reissner and Pontryagin–Hu–Washizu) variational formulations. In particular, their action integrals were, rather, of the type:

$$\overline{\mathscr{A}}_{\mathrm{LH}}[\mathfrak{q}, \overline{\mathbf{F}}^{\mathrm{int}}, t_f] \stackrel{\text{def}}{=} \int_{t_i}^{t_f} \left[ \overline{\mathscr{K}}[\mathfrak{q}, \dot{\mathfrak{q}}] - \overline{\mathscr{E}}_{cc}^{\mathrm{elas}}[\mathfrak{q}, \overline{\mathbf{F}}^{\mathrm{int}}] \right] d\hat{t} \,, \tag{106}$$

which slightly differ from the PHS action integral $\overline{\mathscr{A}}_{\mathrm{HRPH}}[\mathfrak{q}, \overset{\star}{\overline{\mathbf{F}}}{}^{\mathrm{int}}, \pi, t_f]$ in Equation (85), but being currently in use to cope with structural (shear, volume, etc.) locking phenomena. This is supposed to give better performance than the simple Hamiltonian approach (with action integrals like $\overline{\mathscr{A}}_{\mathrm{H}}[\mathfrak{q}, \pi, t_f]$ in Equation (77)).

The integral of action in Equation (106) considers the generalized internal force field (i.e., $\overline{\mathbf{F}}^{\mathrm{int}}$) as a primary variable, and not its impulse (i.e., $\overset{\star}{\overline{\mathbf{F}}}{}^{\mathrm{int}}$). In comparison, our former numerical work with the PHS strategy may have used an action integral $\overline{\mathscr{A}}_{\mathrm{HR}}[\mathfrak{q}, \overline{\mathbf{F}}^{\mathrm{int}}, \pi, t_f]$ in Equation (88). Discrete-time PHS modeling following the weak ideas was also applied quite recently in [30,41,42]. However, variationally, this choice of a space–time weak integration formulation cannot include the kinematic compatibility conditions (nor the Rankine–Hugoniot-like jump ones) as natural variational conditions. A weak and fully variational formulation, including the use of the impulse $\overset{\star}{\overline{\mathbf{F}}}{}^{\mathrm{int}}$ of the generalized internal

forces $\overline{\mathbf{F}}^{\text{int}} \equiv \dfrac{\partial \, \overset{\star}{\overline{\mathbf{F}}}\vphantom{F}^{\text{int}}}{\partial t}$, was used in thermoelasticity by [113,114,139]. Their action integrals of the type:

$$\overline{\mathscr{A}}_{\text{LH}^*}[\mathfrak{q}, \overset{\star}{\overline{\mathbf{F}}}\vphantom{F}^{\text{int}}, t_f] \sim \overline{\mathscr{A}}_{\text{LH}}[\mathfrak{q}, \overline{\mathbf{F}}^{\text{int}}, t_f] \qquad (107)$$

still slightly differ from the PHS action integral $\overline{\mathscr{A}}_{\text{PH}}[\mathfrak{q}, \overset{\star}{\overline{\mathbf{F}}}\vphantom{F}^{\text{int}}, \boldsymbol{\pi}, t_f]$ in Equation (85). This yields space–time weak forms for the conservation laws, where the proper kinematic conditions of compatibility (and the Rankine–Hugoniot jump conditions) are natural variational conditions of the integral of actions. Nevertheless, we do not know yet about the benchmarking numerical experiments, albeit some numerical simulations have been performed (e.g., as in [114]).

As a specificity of that weak and fully variational formulation, one must allow (in accordance with the causal virtual power principle) variations for $\mathfrak{q}_{/\partial T}(t_i) \neq \mathbf{0}_{5\times 1}$, $\mathfrak{p}_{/\partial T}(t_i) \neq \mathbf{0}_{5\times 1}$ (for the Hamiltonian case or $\dot{\mathfrak{q}}_{/\partial T}(t_i) \neq \mathbf{0}_{5\times 1}$ for the Lagrangian case), and $\overset{\star}{\mathbf{F}}{}^{\text{int}}_{/\partial T}(t_i) \neq \mathbf{0}$ to variationally recover the nonzero initial conditions for the displacement and velocity fields enforced by singularly time-localized inertial forces [91,95,96]. The proper accounting of the contribution of the internal forces, as well as of the nonzero initial kinematic conditions has a certain importance from the point of view of numerical computation. Interestingly, the aforementioned work on linear elastodynamics in [91] (see also [92]) provided some interesting numerical comparisons regarding the time integration with the extended Hamilton–(Hellinger–Prange–Reissner and Pontryagin–Hu–Washizu) variational formulations. As many others also concluded regarding the stability and high-frequency behavior, some algorithms generated by variational formulations with weakly enforced initial conditions (which we suggested before to treat as a contribution to the external virtual power due to initial inertia of the material points) are unconditionally stable, higher-order accurate, and effectively dissipative in the spurious high-frequency modes. Some algorithms derived from variational formulations with initial conditions enforced a priori are unconditionally stable, but less accurate and endowed with a less effective algorithmic dissipation. Moreover, they mentioned, with reference to the global performance of space–time discretization, that the time integration methods obtainable by the variational form principle do not show, generally, any remarkable improvement of the performances, but merit attention and developments.

## 5. Conclusions and Future Directions

In modeling, finding the most-convenient representation of the equations of motion of the studied physical system is an important task, which may have an impact on the resolution methods and the control algorithms. The identification of the Hamiltonian structure of the dynamical model is important, as it yields information about conserved quantities (such as energy function, momentum, etc.) in the system that, from a fundamental point of view, should preferably be respected in coupling simulations. Our goal was to contribute to such a quest for improved analysis procedures.

**Main objective.** More precisely, the intent of this work was to present the basic PHS formulation of Naghdi–Reissner's moderately thick 2D shell model. By their nature, 2D shell equations are approximations to the 3D ones, which result from making specific assumptions determining the range of their applicability. In order to lay down thusly the appropriate PHS expressions for a future numerical work on shell–fluid coupling, we followed a standard variational approach to degenerate the 3D shell continuum description to 2D shell ones exhibiting their effective behaviors. The equations of motion were obtained according to the principle of virtual powers, and the possible intrinsic (but well-known) coupling of kinematic modes due to the often-neglected shell geometric considerations was emphasized. These are relevant for modeling doubly curved shells for example.

**Main findings of our investigation.** The originality of our contribution was to establish then a formal link between an existing mixed (i.e., multi-primary field) variational principle (a complete Hamilton–Hellinger–Prange–Reissner (**HHPR**) one, with a generalized internal force impulse field) and the PHS formulation. This interpretation may seem "new" to some research communities less familiar with (computational) continuum mechanics and interested in multiphysics coupling. Although the variational model is known in some publications, for completeness, we included a brief discussion of the numerical integration procedures related to the HHPR principle and its canonical PHS equations. This discussion also briefly sketched out a possible extension to a Hamilton–Pontryagin–Hu–Washizu type of variational principle for the PHS formulation.

We believe our contribution allows a better understanding of this PHS formalism from the perspective of continuum mechanics and its computational analysis. It also sheds new light on the limit cases developed recently in [14,15,24,130] for the thin or thick plate theories. The formal variational link, the short discussion on variational integrations, and our PHS formulation of the NR-shell model will be useful for our extensions to other theoretical and numerical applications.

**Future extensions of our PHS formulation.** Below, we collect some extensions that we intend to develop in the near future:

- *PHS formulations completed for more complex constitutive material laws and shell geometry*. The full formulation of the PHS of the NR shell is nevertheless far from being exposed here. For simplicity, our mathematical treatment of this continuous shell medium was intentionally limited to transversely homogeneous and tangentially isotropic linearly elastic behavior. The model was expressed in the *principal orthogonal curvilinear coordinate system*, which would apply as well to many other shell mechanic models with both constant and variable thicknesses. Moreover, our derivation did not consider as well the singularities and/or discontinuities of any kind (whether fixed or mobile), which can occur, for example, for the cases of shell junctions or impacts. The required regularity can—and must—be relaxed in the variational formalisms so that they include jump conditions for the cases of shell junctions or impacts for instances.
  For more general engineering applications coupled with various physical modeling, our work can therefore be modified for a transversely isotropic material [77] and in non-principal curvilinear coordinate forms by formulating it in invariant tensor notation. This may seem very complex at first sight, but is very efficient for a large variety of more complex (composite) shell structure shapes appearing in many industries, including the automotive, aircraft, and civil engineering industries. Giving the formulations of some of these extensions will be one of our future objectives.
- *Multiphysics coupling*. The use of the PHS formalism in multiphysics coupling analyses constitutes for us another main interest. In particular, the generic variational framework used to derive the PHS formulation for the NR shell model is broadly applicable as well to more complex physical models. By unifying the expressions of the coupled systems of conservationlaws, the so-derived multiphysics PHS models can further be used for controller design in a wide variety of applications such as inflatable space structures, launcher tank vibration damping, payload vibration protection with smart materials, and many other related applications.
  Interestingly, the PHS formalism can be used quite readily and directly in certain vibro-acoustical or thermo-mechanical applications, based, for instance, on the thermo-elasticity theory in [110,112–114,139]. With the proper definition of the operators, which we shall also present elsewhere, the PHS formulation can cover these coupling cases with the infinitesimal deformation cases of linear elastic, moderately thick shells and linear elastic thin shells (with or without rotatory inertia), as well as either deep or shallow shells (including the degenerate case of plates) [70,71]. An interesting coupling can also be with some enhanced shell models, such as those in [54,59,98]. These models include more complex features than the NR shell model, such as Cosserat's additional microrotations in [54,140] and the micromorphic-like stress field features

across the shell thickness in [59,98]. Their micromorphic shell models were obtained variationally, by applying the Hellinger–Prange–Reissner functional directly to the 3D elastic shell structure with the polynomial approximations of the stress field. As a result, the shell model [98] proved to be very accurate in the evaluation of stresses, since 3D equilibrium equations and boundary conditions at the faces of the shell are (or can be approximatively) satisfied.

As regards the fluid–shell structure interaction problems, the application of our variational description of the PHS equations turns out to be also very close to the one in [115]. Nevertheless, in that work, as in others, some features required for the PHS formulation of the shell were missing.

- *Numerical simulations*. Once the PHS formulation has been obtained at the continuous level, a very interesting topic is the structure-preserving discretization of the system. Indeed, obtaining a finite-dimensional PHS from of an infinite-dimensional one in a systematic way is very useful for simulation and control purposes. We discussed in Section 4.2 the possible numerical solution strategies (and mentioned some of their illustrations) that can be applied to our mixed variational formulation of NR shell elastodynamics by the (weak or strong) PHS formalisms ([91,92]). Investigations of the weak time PHS formalism have been applied quite recently in [30,41,42], but have not yet been applied to shells. Therefore, regarding the benefit of the current symplectic computational developments, benchmarking comparisons with the other GNI strategies should be also performed and will be presented for beams, plates, and shells. Simulations including the multiphysics enrichments mentioned above will also be carried out.

**Author Contributions:** Conceptualization, M.C., I.F.N. and D.M.; Methodology, M.C. and Y.G.; Validation, M.C., I.F.N. and Y.G.; Formal analysis, M.C. and I.F.N.; Investigation, M.C., I.F.N. and D.M.; Resources, M.C., I.F.N. and D.M.; Data curation, not applicable; Writing—original draft, M.C. and I.F.N.; Writing—review & editing, M.C., Y.G. and D.M.; Supervision, Y.G. and D.M.; Project administration, Y.G. and D.M. All authors have read and agreed to the published version of the manuscript.

**Funding:** The development of the port-Hamiltonian approach for these multidomain distributed-parameter systems was started first in an ANR (French National Research Agency) research project, which was named Hamiltonian Methods for the Control of Multidomain Distributed Parameter Systems (HAMECMOPSYS) (No. ANR-11-BS03-0002), pursued in a second ANR-DFG research project named INterconnected in FInite-Dimensional systems for HEterogenous Media (INFIDHEM) (No. ANR-16-CE92-0028). I.F.N.'s research internship was supported by ISAE-Supaero/DMSM. The authors gratefully acknowledge these projects and ISAE-Supaero/DMSM for the financial support.

**Institutional Review Board Statement:** Not applicable.

**Informed Consent Statement:** Not applicable.

**Data Availability Statement:** Not applicable.

**Acknowledgments:** The authors thank the anonymous reviewers for their expertise and helpful comments, and the journal for its assistance in proofreading and checking the document. They have greatly enabled us to improve this communication.

**Conflicts of Interest:** The authors declare no conflict of interest.

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
