# Peer review of "Port-Hamiltonian Formulations of Some Elastodynamics Theories of Isotropic and Linearly Elastic Shells: Naghdi–Reissner’s Moderately Thick Shells"

_applsci, doi:10.3390/app13042608_

Round 1

Reviewer 1 Report

The current manuscript is long and unclear. Most of the results are hidden behind notation and/or redefinition of many variables/functions. Some comments on Lagrangian/Hamiltonian formalism are either wrong or misleading. A tremendous revision would be needed to be publishable in my opinion. 

Author Response

First, we thank the reviewers for their helpful comments. This allowed us to notice some language errors. We hope to have corrected most of them. Moreover, we understand that this contribution is really long. The reason is that we wanted the manuscript to be clear and sufficiently self-contained.

We have tried to reorganize the manuscript so that your message is clearer and the added values well- identified. Several bibliographic references have been added to justify the relevance of the work. As well, we have added a discussion section on the applicative usefulness of our analysis.

This work proposes a new application of the port-Hamiltonian system (PHS) formalism in the framework of Structural and Continuum Mechanics (CSM). The continuous model considered is that of Nagdhi-Reissner’s (NR) shells, for which there are non-trivial elastic and inertial coupling effects due to geometric curvatures. This analytical study constitutes a very useful preliminary work for the numerical analysis of the Fluid/Shell- Structure interactions (FSI), of the control from the edge of the shell, and which will be developed more specif- ically within the framework of symplectic geometry and Hamiltonian variational formalism. While keeping in mind our future applications to FSI coupling (the central theme of our work) developed elsewhere, we thought (almost like [35,43]) important to re-derive the equations of the two-dimensional (2D) linear elastic shell mod-els from the three-dimensional (3D) ones in order to highlight their main assumptions and keep trace of the loads imposed on the generic 3D linear elastic solid structure. For our aim and for the sake of transparency, this contribution tries to make the formulations of both the linear NR’s shell theory and the variational ones ac-cessible to a wide variety of readers, at the expense of conciseness. This choice yields indeed a ‘rather explicit’ (but consequently long) presentation of the NR shell model, but it allows to sufficiently unify and clarify (for readers unfamiliar with shell theories) the notations of formulations before their numerical treatment.

An originality of this communication is the formal identification of the extended variational Hamiltonian principle corresponding to the PHS formalism in (see Section 4). This constitutes a major contribution to research on this PHS formalism because it indicates a procedure to systematically generate its systems of canonical equations for CSM problems. Quite unexpectedly, the identified variational principle corresponds to a complete Hamilton-Hellinger-Prange-Reissner (HHPR) principle, but with the impulse of the generalized internal force field as one of the primary variables in addition to the displacement and momentum fields. Such a variational model is known in some communities, notably in [77, 116] which provided various variational principles for linear elastodynamics and Thermoelasticity, etc. We include a subsection developing a brief discussion of numerical integration procedures related to the HHPR principle and its canonical PHS equations.

Finally, we think these findings and insights are useful for our work, but also to other close ones (for instance in fluid/shell-structure interaction problems [2], etc.) where some features (like the enforcing of continuity con- ditions) required for the PHS formulation of the shell were missing.

Reviewer 2 Report

This work aims at applying port-Hamiltonian system (PHS) approach to the Naghdi-Reissner’s five-kinematic-field shell model in linear elasticity, while including often neglected higher-order intrinsic geometric coupling effects, and preparing so the theoretical background required for the coupling (or interconnection) with an acoustic fluid model, and the different types of interactions which can arise among them. More precisely, the intent of this work is to present the basic PHS formulation of Nagdhi-Reissner’s moderately thick 2D-shell model, in order to lay down thus the appropriate PHS expressions for a future numerical work on shell-fluid coupling. However, the accuracy of the obtained model remains to be discussed, and the innovation of this paper is not highlighted, so the conclusions cannot reflect the practical application value of the project. Furthermore, some problems about the expression and the sentence exist in this paper. Thus, the manuscript is not recommended for publication, and some suggestions are advised for further revision:

(1)    Please clearly explain the novelty of your manuscript with the state-of-the-art of the specific research direction, and give a detailed conclusions focusing on the new numerical methods.

(2)    At present, the accuracy of this model needs to be discussed. It is suggested to add verification examples in this paper to compare with previous works.

(3)    The analysis of PHS is limited to the shell modeling type, and neglects to explain the advantages and disadvantages of the proposed method.

(4)    The words in the paper should be further checked. For example, the subscript of gI is I, and the incomplete punctuation after the formula.

(5)    The manuscript should be polished by careful editing and paying attention to English spelling, format, and sentence structure.

Author Response

(The authors gave the same response as above.)

Reviewer 3 Report

First of all, I apologize for the delay in revising the manuscript which I read it very carefully and with interest. I am pleaseantly surprised of the scientific level of the work: well written, organized, and with a high degree of mathematical detail to justify the assumptions of the work. It is scientiifically sounded and the work is worth accepting in its present form , I recommend it for publicartion in Applied Sciences.

the port-Hamiltonian systems approach provides an elegant and efficient  framework to model, couple and control various complex multi-physics systems such as mechanical, electrical, thermal ones, while taking into account the effects of energy flows among the different linked PH systems. The aim of thw work was to contribute in finding the most convenient representation of the equations of motion of the studied physical system by use of improved analysis procedures. The work goes in technical details to present the basic PHS formulation of Nagdhi-Reissner’s moderately thick 2D-shell model to obtain appropriate expressions for shell-fluid coupling. The mathematical treatment of the continuous shell medium was limited to transversely-homogeneous, and tangentially isotropic, linearly elastic behavior. The model was expressed in the principal orthogonal curvilinear coordinate system, which would apply as well to many other shell mechanic models with both constant and variable thicknesses. Such a formulation may be useful for coupling with various physical modeling such as  thermo-mechanical or vibro-acoutical applications. BEsides, it is argued that the approach presented here could be modified for transversely isotropic materials and in non-principal curvilinear coordinate forms by formulating in invariant tensor notation, for more complex shell structure materials to be applied in the automotive, aircraft and civil engineering industries.

I reccomend the authors to obtain expert help in english writing to improve some sentences and parts of the text.

Author Response

First, we thank the reviewers for their helpful comments. This allowed us to notice some language errors. We hope to have corrected most of them. Moreover, we understand that this contribution is really long. The reason is that we wanted the manuscript to be clear and sufficiently self-contained.

Round 2

Reviewer 2 Report

The figures need be presented with high resolution, take Figure 1 as an example, please redraw the relevant figures to make them clear. 

Author Response

First, we thank the reviewers for their helpful comments. This allowed us to notice some language errors. We hope to have corrected most of them.

Response to Reviewer 1 Comments

Point 1: The current manuscript is long and unclear. 

Response 1:  We understand that this contribution is really long. The reason is that we wanted the manuscript to be clear and sufficiently self-contained.

In summary, this work proposes a new application of the port-Hamiltonian system (PHS) formalism  in the framework of Structural and Continuum Mechanics (CSM). The continuous model considered is that of Naghdi-Reissner's (NR) shells, for which there are non-trivial elastic and inertial coupling effects due to geometric curvatures.

While keeping in mind our future applications to FSI coupling (the central theme of our work) developed elsewhere, we thought important to re-derive the equations of the two-dimensional (2D) linear elastic shell models from the three-dimensional (3D) ones in order to highlight their main assumptions and keep trace of the loads imposed on the generic 3D linear elastic solid structure. For our aim and for the sake of transparency, this contribution tries to make the formulations of both the linear NR's shell theory and the variational ones accessible to a wide variety of readers, at the expense of conciseness.

This choice yields indeed a `rather explicit' (but consequently long) presentation of the NR shell model, but it allows to sufficiently unify and clarify (for readers unfamiliar with shell theories) the notations of formulations before their numerical treatment.

Point 2: Most of the results are hidden behind notation and/or redefinition of many variables/functions.

Response 2:  We have tried to reorganize the manuscript so that our message is clearer and the added values well-identified.

Several bibliographic references have been added to justify the relevance of the work.

As well, we have added a discussion section on the applicative usefulness of our analysis.

Point 3: Some comments on Lagrangian/Hamiltonian formalism are either wrong or misleading.

Response 3:  The mentioned flaw related to the Lagrangian/Hamilton formalisms has not be found. So to justify our approach, the new submitted version cites few more references agreeing with our presentation.

Point 4: A tremendous revision would be needed to be publishable in my opinion.

Response 4: We also recognized this need, which allowed us to improve this communication. Thank you.

Response to Reviewer 2 Comments

This work aims at applying port-HamiHonian systell (PHS) approach to the Naghdi-Reissner's five-kinematic-field shell model in linear elasticity. while including often neglected higher-order intrinsic, geometric coupling effects, and preparing so the theoretical background required for the coupling (or interconnection) with an acoustic fluid model, and the different types of interactions which can arise among them. More precisely, the intent of this work is to present the basic PHS formulation of Nagdhi-Reissner's moderatety thick 2D-shell model, in order to lay down thus the appropriate PHS expressions for a future numerical work on shell-fluid coupling.

However. the accuracy of the obtained model remains to be discussed. and the innovation of this paper is not highlighted, so the conclusions cannot reflect the practical application value of the proiect. Furthermore. some problems about the expression and the sentence exist in this paper.

Thus. the manuscript is not recommended for publication, and some suggestions are advised for further revision'

Point 1: Please clearly explain the novelty of your manuscript with the state-of-the-art of the specific research direction and give a detailed conclusions focusing on the new numerical methods. 

Response 1: We have added some comments accordingly in the new submitted document. This analytical study constitutes a very useful preliminary work for the numerical analysis of the Fluid/Shell-Structure interactions (FSI), of the control from the edge of the shell, and which will be developed more specifically within the computational framework of symplectic geometry and Hamiltonian variational formalism.

An originality of this communication is the formal identification of the extended variational Hamiltonian principle corresponding to the PHS formalism (see Section 4.1). This constitutes a major contribution to research on this PHS formalism because it indicates a procedure to systematically generate its systems of canonical equations for problems in Structural and Continuum Mechanics (CSM).

Quite unexpectedly, the identified variational principle corresponds to a complete Hamilton-Hellinger-Prange-Reissner (HHPR, related to PHS hereafter in the text) principle, but with the impulse of the generalized internal force field as one of the primary variables in addition to the displacement and momentum fields.

We late discovered that such a variational model was known in some communities (mainly those dealing with variational principles and/or numerical methods), which have provided various variational principles for elastodynamics and thermoelasticity (linear and nonlinear), etc. but it is not yet widespread elsewhere. Our work therefore indicates the existence of different numerical resolution strategies (weak or strong) for PHS analyses, which generally (but not exclusively, as found recently) use the strong numerical formulation in time.

Finally, we think these findings and insights are useful for our work, but also to other close ones (for instance in fluid/shell-structure interaction problems, etc.) where some features (like the enforcing of continuity conditions) required for the PHS formulation of the shell were missing.

Point 2: At present, the accuracy of this model needs to be discussed. It is suggested to add verification examples in this paper to compare with previous works. 

Response 2: this paper did not intend to address the numerical aspects. We had intended and still intend to postpone this task with numerical benchmarking illustrations on the NR shells to a separate communication. However, we have now included a new subsection developing a brief discussion of numerical integration procedures related to the HHPR principle and its canonical PHS equations.

Point 3: The analysis of PHS is limited to the shell modeling type and neglects to explain the advantages and disadvantages of the proposed method. 

Response 3: in the new manuscript, our message is clearer and the added values well-identified. Our goal has been re-explained in the introduction section and re-emphasized again in the conclusion section. Several bibliographic references have been added to justify the relevance of the work. As well, we have added a discussion (at different places) on the applicative usefulness of our analysis (in different research domains). We have also discussed about the advantages and disadvantages of the PHS (mixed variational) formulations in section 4.2. The variational formulation is quite general and might not have been restricted to the NR shell model, according to our formal discussion in Section 4.1.

Point 4: The words in the paper should be further checked. For example. the subscript of gI, is I,and the incomplete punctuation after the formula

Response 4: We totally agree that some words should have been further checked. Nevertheless, we could not locate the mentioned problem for $g_l$. Please, could you be more precise?

Point 5: The manuscript should be polished by careful editing and paying attention to English spelling, format, and sentence structure.

Response 5: We sincerely hope to have fixed most of them, in this short time. Thank you very much for your opinion, your questions, and your suggestions which enrich and allow us to strengthen the submitted article.

Response to Reviewer 3 Comments

First of all , I apologize for the delay in revising the manuscript which I read it very carefully and with interest. I am pleaseantly surprised of the scientific: level of the work: well written, organized, and with a high degree of mathemalical detail to justify the assumplions of the work. It is scientifically sounded and the work is worth accepting in its present form , I recommend it for publication in Applied Sciences.

The port-Hamiltonian systems approach provides an elegant and efficient framework to model, couple and control various complex multi-physics systems such as mechanical. electrical, thermal ones, while taking into account the effects of energy flows among the different linked PH systems. The aim of this work was to contribute in finding the most convenient representation of the equations of motion of the studied physical system by use of improved analysis procedures.

The work goes in techn ical details to present the basic PHS formulation of Nagdhi-Reissner's moderately thick 2D-shell model to obtain appropriate expressions for shell-fluid coupling. The mathematical treatment of the continuous shell medium was limited to transversely-homogeneous, and tangentially isotropic, linearly etastic behavior. The model was expressed in the principal orthogonal curvilinear coordinate system , which would apply as well to many other shell mechanic models with both constant and variable thicknesses. Such a formulation may be useful for coupling with various physical modeling such as thermo-mechanical or vibro·acoustical applications. Besides, it is argued that the approach presented here could be modified for transversely isotropic materials aoo in non-principal curvilinear coordinate forms by formulating in invariant tensor notation, for more complex shell structure materials to be applied in the automotive. aircraft and civil engineering industries.

Point 1: I recommend the authors to obtain expert help in english writing to improve some sentences and parts of the text.

Response 1:  Thank you very much for your review and analysis. We sincerely hope to have corrected most of the language defects on our own.
